# Tissue memory relies on stem cell priming in distal undamaged areas

Chiara Levra Levron [1,2,7], Mika Watanabe [1,2,7], Valentina Proserpio [1,2,3,7], Gabriele Piacenti [1,2], Andrea Lauria [1,2,3], Stefan Kaltenbach[4], Annalaura Tamburrini[1,2,3], Takuma Nohara[5], Francesca Anselmi[1,2], Carlotta Duval[1,2], Luca Elettrico[1,2], Daniela Donna[1,2], Laura Conti[2,6], Denis Baev[3], Ken Natsuga [5], Tzachi Hagai[4], Salvatore Oliviero [1,2,3] & Giacomo Donati [1,2] ✉

Epithelial cells that participated in wound repair elicit a more efficient response to future injuries, which is believed to be locally restricted. Here we show that cell adaptation resulting from a localized tissue damage has a wide spatial impact at a scale not previously appreciated. We demonstrate that a specific stem cell population, distant from the original injury, originates long-lasting wound memory progenitors residing in their own niche. Notably, these distal memory cells have not taken part in the first healing but become intrinsically pre-activated through priming. This cell state, maintained at the chromatin and transcriptional level, leads to an enhanced wound repair that is partially recapitulated through epigenetic perturbation. Importantly wound memory has long-term harmful consequences, exacerbating tumourigenesis. Overall, we show that sub-organ-scale adaptation to injury relies on spatially organized memory-dedicated progenitors, characterized by an actionable cell state that establishes an epigenetic field cancerization and predisposes to tumour onset.

Forming the outer layer of organs, epithelia have predominantly a barrier function and are able to sense and adapt to environmental changes. The homeostatic integrity of these tissues is maintained through a continuous turnover ensured by stem cells (SCs)[1–3], compartmentalized as in those so-called transition zones, present in the epithelia of oesophagus, eye, anus, lung, stomach and cervix[4,5]. Upon tissue damage, each epithelial lineage resident nearby the injury acquires cell plasticity that allows cells to migrate towards the wound site, thus contributing to the re-epithelialization[6–11].

In the past 6 years it emerged that epithelial cells adapt to a local stressful event, such as wound, through the establishment of a chromatin memory to respond faster to an eventual similar challenge[12,13].

Nevertheless, the lineage specificity of wound memories, through a direct comparison of different epidermal cell populations, has not yet been elucidated[14].

The pioneer work by the Fuchs laboratory showed that the SCs located in close proximity to the injured tissue can be trained, suggesting a locally restricted potential of wound memory[12,13]. However, besides the positive effect of memory on regenerative potential[15], negative consequences such as cancer[16] might be related to it. In this context, it would be of translational interest to understand the spatial distribution and extent of wound memory in epithelial cells located distally from the repaired area and to characterize the full impact on the epithelium, long term.

[1]Department of Life Sciences and Systems Biology, University of Turin, Torino, Italy. [2]Molecular Biotechnology Center 'Guido Tarone', University of Turin, Torino, Italy. [3]Italian Institute for Genomic Medicine, Candiolo (TO), Italy. [4]Shmunis School of Biomedicine and Cancer Research, George S Wise Faculty of Life Sciences, Tel Aviv University, Tel Aviv, Israel. [5]Department of Dermatology, Faculty of Medicine and Graduate School of Medicine, Hokkaido University, Sapporo, Japan. [6]Department of Molecular Biotechnology and Health Sciences, University of Turin, Torino, Italy. [7]These authors contributed equally: Chiara Levra Levron, Mika Watanabe, Valentina Proserpio. ✉e-mail: giacomo.donati@unito.it

It has been demonstrated that, in parallel to the immune system, also epithelial cells exhibit trained wound memory of an injury. After a wound event, the chromatin memory is kept transcriptionally dormant, but it allows a quick re-activation in the event of an eventual further lesion[12,13,15]. Differently, another adaptation mechanism of immune cells, named priming, describes an activation state that never turns off even when the stimulus ceases[17]. Currently, it is unknown if other adaptation programmes, such as priming, are opted by epithelial cells[14].

In this Article, in the context of two consecutive skin injuries, we combined lineage tracing with single-cell analysis to comprehensively understand the spatial extent of wound memory and the full spectrum of the adaptive responses of epithelial cells (that is, trained wound memory versus wound priming). We show that specific SCs give rise to wound-primed progenitors that exist in a wide undamaged area distant from the damaged zone, while remaining within their original epidermal niche. Mechanistically, we demonstrate that transcriptional derepression is functional for memory onset. Finally, the memory that is established at the transcriptional and chromatin level in a wide area surrounding the wound represents an epigenetic field cancerization event[18,19] favouring tumour onset. Altogether, our unexpected results drastically change the assumption relative to the spatial distribution of memory cells by highlighting the existence of memory progenitors located far from the injury, in undamaged areas.

## Results

### Tissue injuries educate SCs in distal undamaged areas

Recently, the existence of an epigenetic wound memory has been assessed for hair follicle (HF) SCs[15], but the precise lineage identity, the spatial distribution and the spectrum of adaptation programmes acquired by the memory cells are still unknown (Fig. 1a).

We focused on three well-characterized compartmentalized cell populations of the HF: Lrig1+ SCs localize at the HF junctional zone (JZ) and maintain the sebocytes, sebaceous ducts and infundibulum (INFU); Gata6+ are committed-to-differentiation and differentiated duct cells of the upper pilosebaceous unit; and Lgr5+ SCs localize at the lower HF[11,20,21].

To understand if their progenies elicit a wound-induced memory, we genetically labelled (GL) Lrig1+, Lgr5+ and Gata6+ HF cells (Extended Data Fig. 1a,b) in adult mice. We used a two-consecutive-injury model in tail skin (Fig. 1b) where, in the presence of a minimal tissue contraction the second injury heals faster than the first one (Extended Data Fig. 1c–f). Briefly, at time 0w we performed a first full thickness wound. Eight weeks after (8w pw1) when a new homeostasis was reset (Extended Data Figs. 1f–s and 2a–e), we induced a second identical and overlapping injury (Fig. 1b and Extended Data Fig. 2f). This procedure allows the removal of the HF-derived interfollicular epidermal (IFE) SCs that were the focus of Gonzales et al.'s work[15]. In these settings, we investigated the memory of SCs that remain localized in their original HF niche, without contributing to the repair of the IFE.

Although the contribution of Lrig1 GL cells is quantitatively higher than Lgr5 GL cells, both progenies show enhanced re-epithelialization ability during the second healing, while differentiating Gata6 GL cells do not (Fig. 1c and Extended Data Fig. 2g).

The spatial extent of the wound memory is unknown, although the cell contribution to skin full thickness wound repair is spatially restricted to less than 1 mm away from the injury in a 1 mm ear wound context[22], as well as the communication between damaged HFs[23]. Consistently, the wound-engaged HFs (defined as the HFs in which GL tdTomato+ cells move from their homeostatic HF niche into IFE) are mainly localized in the close surroundings of the injury at 1w pw1. However, at 1w pw2 this phenotype exists up to 7 mm away from the injury site, exclusively for Lrig1 progeny (Fig. 1d–g and Extended Data Fig. 2h–k). Horizontal whole-mount confirmed that Lrig1 GL cells in wound distal areas remain localized in their niche (upper HF) at 1w pw1 until 8w pw1, while they exit into the IFE as basal and suprabasal only at 1w pw2 (Fig. 1g). This wound-elicited education of Lrig1 GL cells in distal HFs was confirmed in an additional setting where a second injury was performed distally from the previously healed area (zone B) (Fig. 1d,h and Extended Data Fig. 2l,m).

Thus, we show that, as a consequence of a lesion, different epithelial SC lineages acquire memory if located in wound proximity. However, exclusively the Lrig1+ SC progeny is wound educated within the HFs located up to ~7 mm from the injury. We will refer to this phenomenon as distal memory.

### Distal memory elicits an enhanced migration

Since Lrig1 GL cells, but not Lgr5 GL ones, show adaptive behaviour to injury in distal areas, we compared the expression profile of their sorted progenies. The two hair follicle stem cell (HFSC) lineages have specific wound-associated transcriptional programmes (Extended Data Fig. 3a). We defined the memory genes as those genes deregulated during the first healing and whose deregulation was of greater magnitude after the second injury. Lrig1 GL cells have more memory genes with respect to Lgr5 GL cells (Fig. 2a and Supplementary Table 1). Their Gene Ontology (GO) analysis suggests 'Cell polarity/ Migration', a major cell phenotype in wound healing[22,24], but not proliferation, as a feature of the wound-educated Lrig1 GL cells (Fig. 2b and Extended Data Fig. 3b).

To validate the enhanced migratory potential of distal memory cells, we collected skin biopsies at 8w pw1 from distal memory (Distal) or naïve, without memory, (Ctrl) areas and we performed migration assays[13]. Ex vivo migration assay confirmed the higher migratory ability of Distal Lrig1 GL cells when compared with Ctrl as well as an increased cell polarization[25] (Fig. 2c,d and Extended Data Fig. 3c–g). This is also confirmed in vitro, in absence of their niche stimuli (Fig. 2e–g and Extended Data Fig. 3h–k). Thus, distal memory elicits enhanced repair capabilities that, once established, are maintained in the absence of the physiological microenvironment.

At 8w pw1 the memory genes display: (A) a constitutive expression pattern where they remain deregulated or (B) an inducible trend

---

**Fig. 1 | Wound memory in HF lineages and its spatial extension.**
**a**, Left: repairing HFs in wound proximity and the HF-derived newly formed epidermis, focus of the study by Gonzales et al.[15]. Right: cells in distal HFs and their adaptation programme to wound are the focus of this study. IFESCs: interfollicular stem cells. **b**, Two-consecutive-skin-injury model: −1w, GL; 0w, homeostasis and first injury; 1w pw1, 1 week post first wound; 2w pw1, 2 weeks post first wound; 8w pw1, 8 weeks post first wound, and second injury; 1w pw2, 1 week post second wound. **c**, Localizations of Lrig1+, Lgr5+ and Gata6+ cells (left). Epidermal whole-mounts of GL tdTomato+ cells (red channel) exiting from HFs (right). HG: hair germ; INFU: infundibulum. **d–g**, Whole-mounts of Lrig1 GL engaged HFs (red channel). Capital letters define four zones: A, up to 1 mm from wound; B, 1 to 3 mm; C from 3 to 5 mm; D from 5 to 7 mm. Dashed red lines highlight representative engaged HF triplets (**d**). Number of engaged HFs.

*n* = 6 (0w and 8w pw1), *n* = 9 (1w pw1 and 1w pw2) (**e**). Representative pictures of HF engagement in the four zones (A, B, C and D) at 1w pw1 and 1w pw2 (**f**). Images showing the localization of Lrig1 GL cells in zone D. Asterisks represent Lrig1 GL fibroblasts. White arrow indicates GL cells exiting into IFE (**g**). **h**, Top: epidermal whole mounts showing the closure of a second overlapping wound (1w pw2_Over) (left) or of a distal injury (1w pw2_Distal), made in zone B (right). Bottom: quantification of distance from wound centre. *n* = 4 (1w pw2_Over), *n* = 5 (1w pw1) and *n* = 6 (1w pw2_Distal). Yellow arrows indicate wound position (**f** and **g**). Dashed circles indicate wound perimeter at 8w pw1 (light blue), 0w (orange); lines underline the migration front of GL cells at 1w pw2 (purple) or 1w pw1 (green) (**c** and **h**). *P*-value from a two-tailed *t*-test. Data are mean ± s.d. Scale bars: 1 mm (**c**, **d** and **h**); 100 μm (**f** and **g**).

where, after wound resolution, the 0w expression is restored (Fig. 2h and Extended Data Fig. 3l,m). Since in Lrig1 GL cells almost 90% of the memory genes belong to the 'A' type, we hypothesized that the constitutive expression of memory genes at 8w pw1 might be due to the existence of priming, as wound adaptation programme, in a subpopulation of Lrig1 SC progeny.

## Wound priming of Lrig1 SC progeny in distal HFs

To better dissect the transcriptional basis of wound memory, considering cell heterogeneity in HF niches[26], we performed single-cell RNA sequencing (scRNA-seq) of Lrig1 GL cells at 0w, 1w pw1, 8w pw1 and 1w pw2 (Fig. 3a and Extended Data Fig. 4a). The combination of unsupervised clustering analysis, cluster marker identification, data

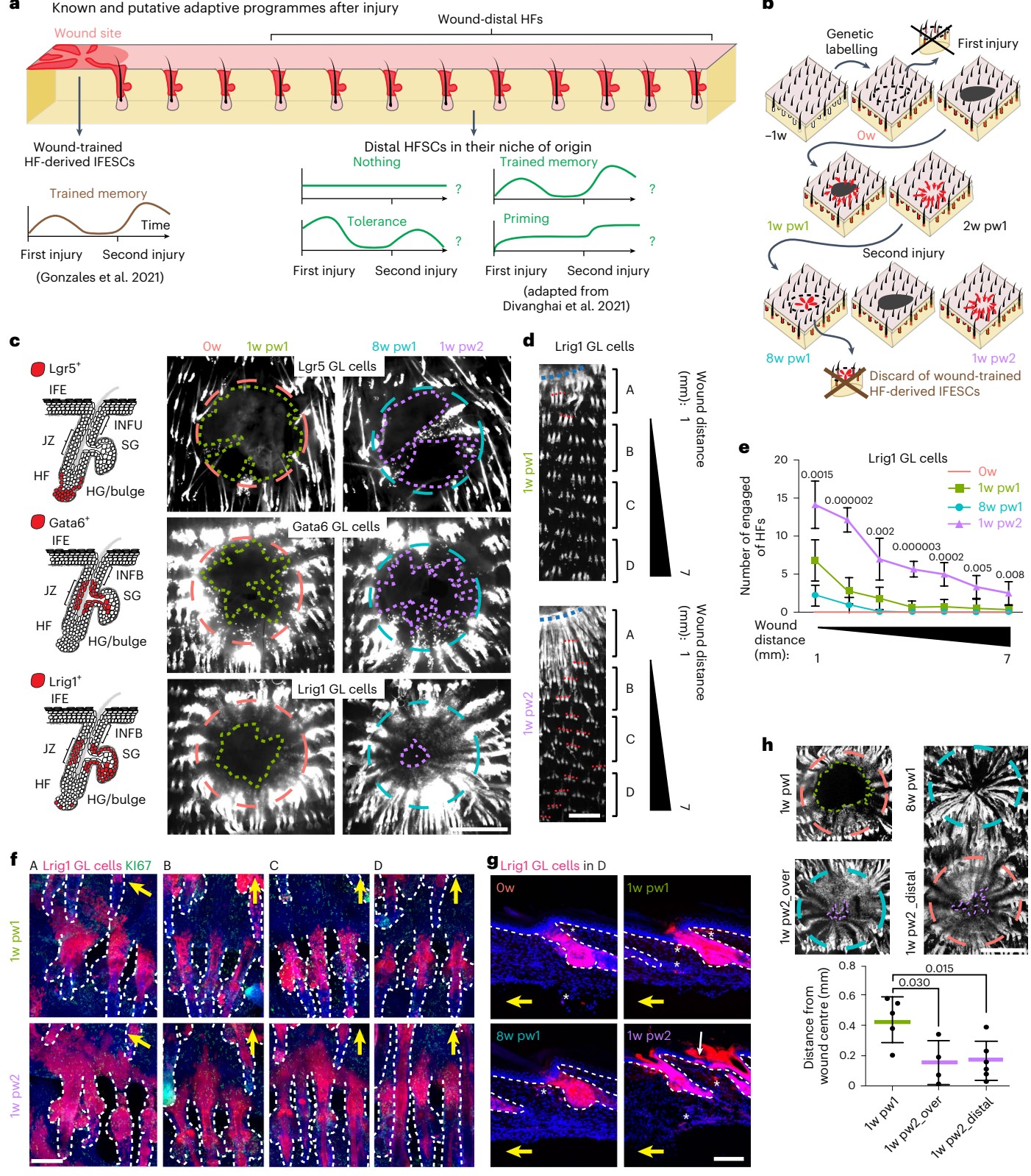

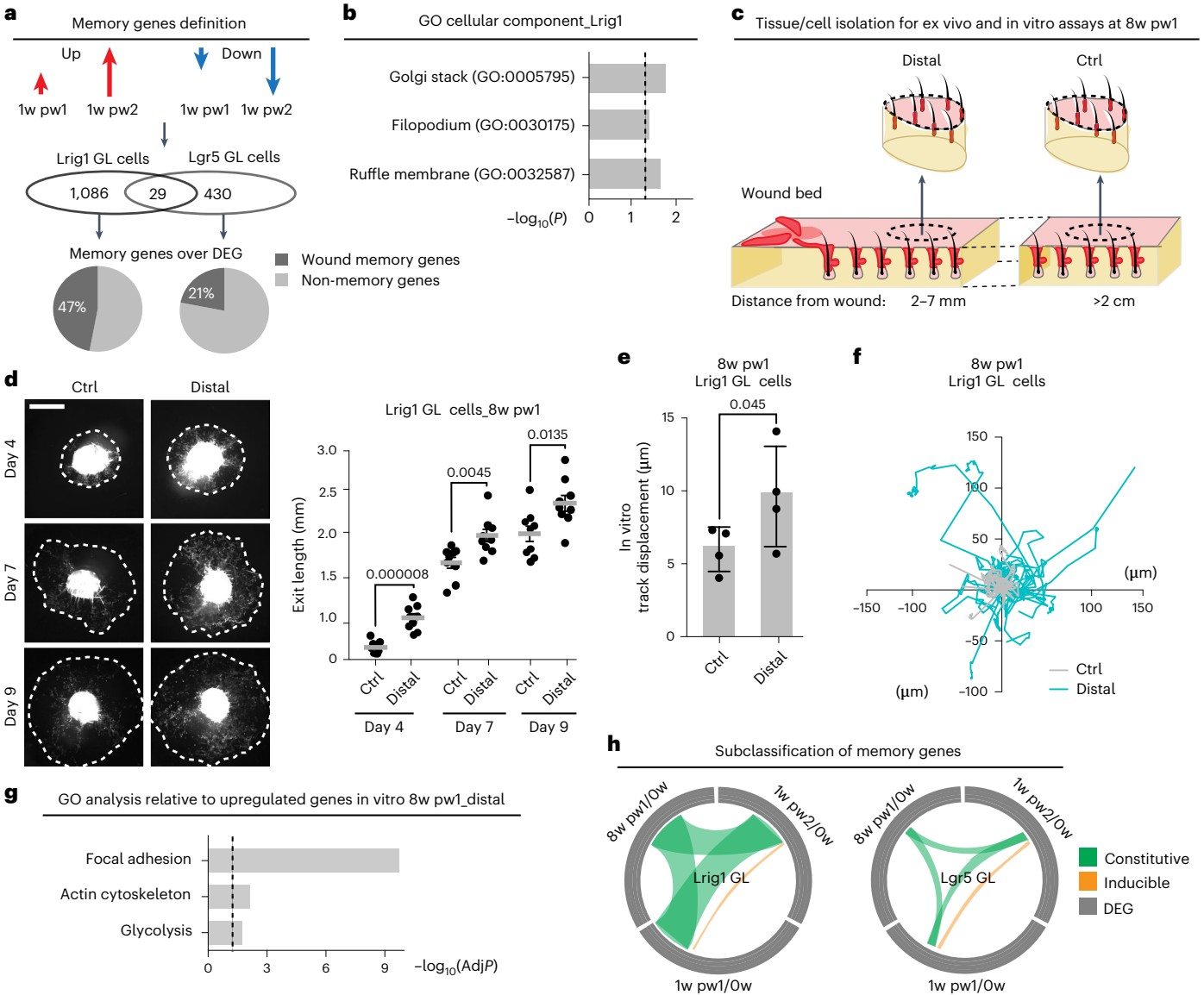

**Fig. 2 | Transcriptome of Lrig1 GL cells predicts a new primed subpopulation arising after wound resolution. a**, Memory genes (logFC (1w pw2) > logFC (1w pw1)) and Venn diagram reporting the number and the percentage of inferred memory genes over the total DEGs across the time course in Lgr5 and Lrig1 GL cells. $n = 3$ mice. **b**, Enriched GO terms for memory genes in Lrig1 GL cells, as $-\log_{10}$ of $p$-value. Dashed line indicates significance. $n = 3$ mice. **c**, Ex vivo and in vitro experimental settings at 8w pw1: Lrig1 GL skin biopsies from wound-educated Distal region and from Ctrl area, outside from memory zone are compared. For in vitro assays the biopsies were dissociated, and cells were plated. **d**, Images of ex vivo migration of Lrig1 GL cells (left) and quantification of the migration (exit length) (right). Dashed lines mark the migration front

of epidermal cells. Scale bar: 2 mm. Data are mean ± s.e.m. $n = 9$ skin explants. **e,f**, Time lapse migration assay in vitro. Track displacement (μm) (**e**) and representative normalized start position graph (**f**) of cultured Lrig1 GL cells. Data are mean ± s.d. $n = 4$ mice. **g**, RNA-seq of cultured Lrig1 GL cells from wound-educated distal region (Distal 3–7 mm from wound bed ($n = 6$)) and Ctrl (distance >2 cm from wound bed ($n = 8$)). The GO analysis relative to upregulated genes in Distal is reported, as $-\log_{10}$ of adjusted $p$-value (AdjP). Dashed line underlines significance. **h**, Circular ideogram plot (CIRCOS) of shared DEGs (grey) between 1w pw1, 8w pw1 and 1w pw2 (green) or between 1w pw1 and 1w pw2 only (yellow), in Lgr5 and Lrig1 GL skins. $P$-value from a two-tailed $t$-test.

integration with Joost et al.[26] dataset, together with pseudotime analysis distinguishes cells according to lineage identity and differentiation stage (Fig. 3b–d, Extended Data Fig. 4b–f and Supplementary Table 2) is summarized in Fig. 3c. As previously suggested[20], our scRNA-seq data integrated with marker genes from Dekoninck et al.[27] confirm that Lrig1 GL cells contribute to the healing specifically as interscale lineage, one of the two IFE differentiation programmes[28] (Extended Data Fig. 4g).

Trajectory D is the most interesting in terms of wound-induced plasticity. Indeed, Lrig1+ SCs acquire plasticity while leaving the Niche of Origin-JZ Cluster 11 and move along the trajectory in the Transition Cluster 7 towards differentiated IFE (cluster 0), where they contribute to the repair (Fig. 3e). In the Transition Cluster, cells reside in the INFU at

the starting point of the trajectory and then move towards IFE, as suggested by the expression of the INFU marker Postn[26] (Extended Data Fig. 4h). Comparing the two homeostatic stages (0w and 8w pw1), we notice an unexpected increase in the number of Lrig1 GL cells in the INFU at 8w pw1 (Fig. 3e,f). Since the cells from the infundibular cluster at 8w pw1 express intermediate levels of the Transition Cluster genes, between 0w and healing phases (1w pw1 and 1w pw2), with the second induction being greater than 1w pw1 (Fig. 3g), we infer the existence of priming adaptive programme, as described for immune cells[17]. Indeed, in the newly established homeostasis (8w pw1) the genes associated with 'Cell activation' and the wound-activation marker[29] Krt6 are primed (Extended Data Fig. 4h–j).

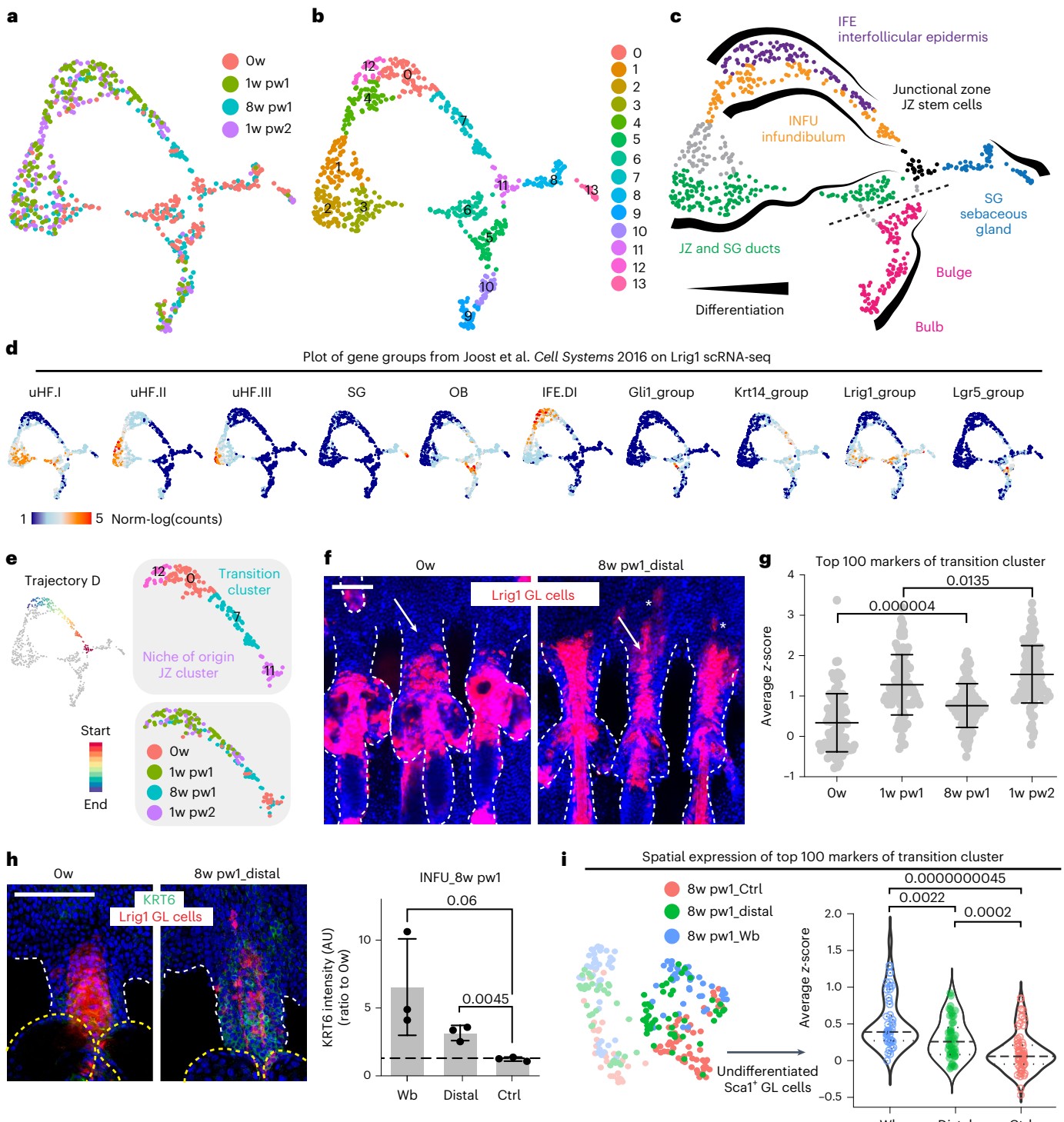

**Fig. 3 | Identification of wound-primed cells in the INFU with a pre-activated transcriptional programme. a**, UMAP of scRNA-seq data of Lrig1 GL cells at 0w (red), 1w pw1(green), 8w pw (light blue) and 1w pw2 (purple). **b**, Unsupervised clustering of single-cell transcriptomic data. **c**, Summary illustrating epidermal lineages and differentiation in Lrig1 GL single-cell data. Cells are coloured according to their epidermal lineages. Dashed line identifies the homeostatic compartment boundary between upper and lower HF[11]. **d**, Expression plot of gene set from Joost et al. study[26]. **e**, Pseudotime analysis. Trajectory D is coloured by timepoints and clusters. **f**, Epidermal whole mounts of Lrig1 GL tdTomato[+] cells occupancy showing cell localization in the INFU (white arrows) in the distal HFs at 8w pw1 (at 5 mm from wound site). Asterisks mark differentiated IFE cells. **g**, Plot of the average z-score of the top 100 markers of Transition Cluster 7 showing the intermediate transcriptional state at 8w pw1. **h**, INFU

whole-mount pictures of Krt6 at 0w and 8w pw1 at 5 mm from wound site (left) and quantification at 8w pw1 (right), in wound bed (Wb), distal memory region (Distal 3–7 mm from Wb) or from a naïve region (Ctrl >2 cm from Wb). Ratio to 0w is plotted. Dashed lines indicate 0w average. Data are mean ± s.d. *n* = 3 mice. **i**, Spatially resolved scRNA-seq analysis of upper-HF Lrig1 GL cells from wound bed (Wb), distal memory region (Distal 3–7 mm from Wb) and naïve region (Ctrl >2 cm from Wb) was performed at 8w pw1 and analysed in Extended Data Fig. 5. Plot of average z-score expression of the 100 markers of the Transition Cluster identified in Fig. 3g in the cells from each group. *n* = 42 (Wb), *n* = 48 (Distal), *n* = 59 (Ctrl) cells. *P*-value from a two-tailed *t*-test. Scale bars: 100 μm (**f** and **h**). Data are median with 25th and 75th percentiles if not differently indicated. scRNA-seq data (**a**–**e**, **g** and **i**) are the integration of two independent experiments, each of them based on four biological replicates.

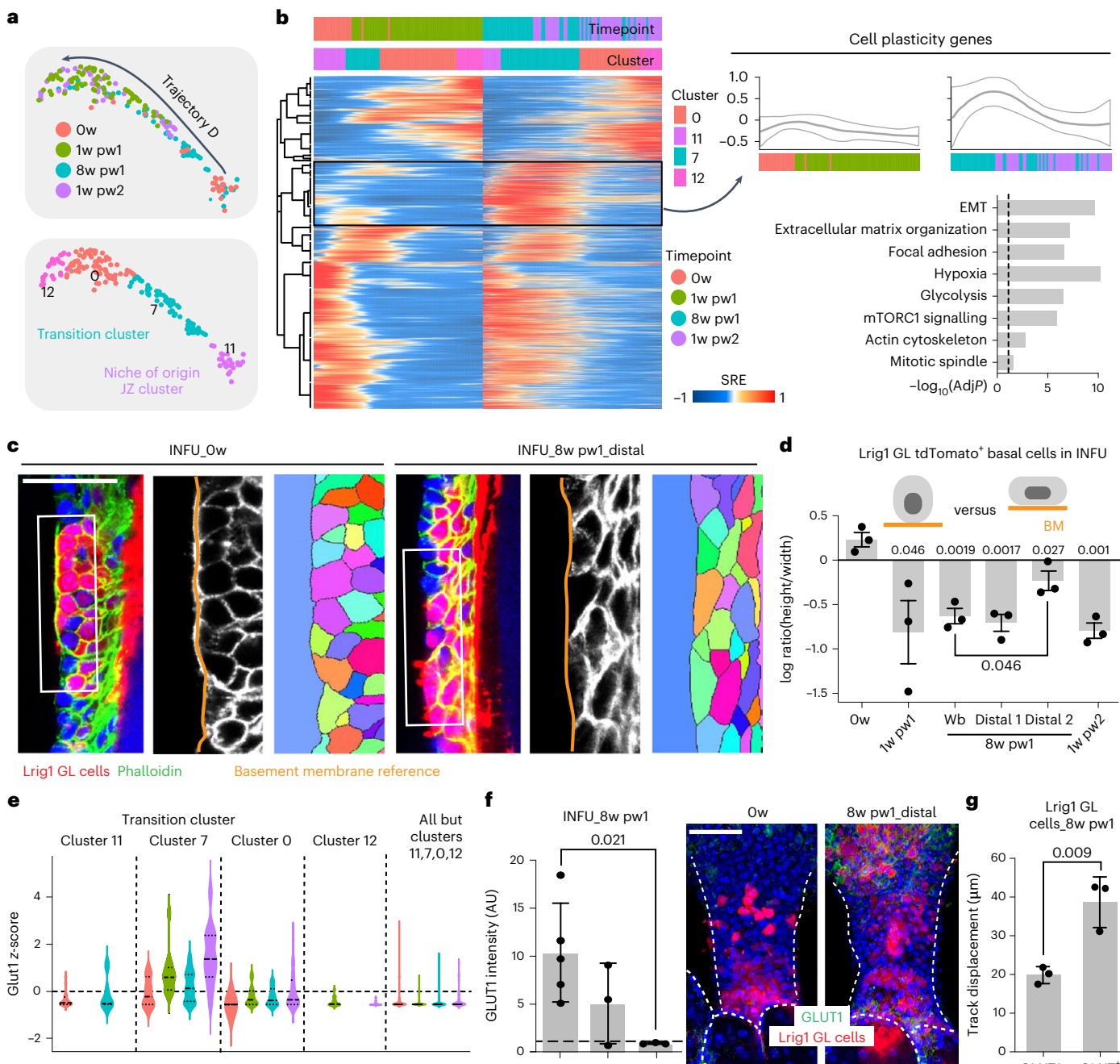

**Fig. 4 | Spatial distribution and cell state characterization of priming.**
**a**, UMAP of pseudotime trajectory D with timepoint (top) and clusters (bottom).
**b**, Comparison of first and second healing. Left: smoothed relative expression (SRE) of deregulated genes in trajectory D. Clusters and timepoints are indicated above, and transiently induced cell plasticity genes (Supplementary Table 3) are highlighted (black box). Right: average $z$-score expression of cell plasticity genes (top) and their GO analysis (bottom) as $-\log_{10}$ of the adjusted $p$-value (AdjP). Dashed line underlines significance. **c**, Single stack of epidermal whole-mount stained with phalloidin (green) and relative cell segmentation (right) at 0w and distal 8w pw1 at 5 mm from wound bed. **d**, Quantification of cell shape (phalloidin) as ratio between height and width in Lrig1 GL tdTomato[+] cells in the INFU. 8w pw1 sample was analysed in wound bed (Wb) and distal areas (Distal 1: 3–5 mm and Distal 2: 5–7 mm from wound bed). $n = 3$ mice. **e**, Violin plot of Glut1 expression in cells from clusters 0, 7, 11 and 12 and all the other clusters. Cells are divided by timepoint. **f**, Quantification of Glut1 level in the INFU at 8w pw1, in Wb, Distal and Ctrl (>2 cm from Wb) (ratio to 0w) (left) and INFU whole-mount pictures (right) are shown. $n = 3$ (Wb), $n = 5$ (Distal and Ctrl). **g**, Time lapse migration assay in vitro. Mean track displacement (µm) of cultured 8w pw1 Glut1[+] versus Glut1[−] Lrig1 GL tdTomato[+] sorted cells. $n = 3$ mice. $P$-value from a two-tailed $t$-test. Data are mean ± s.d. Scale bars: 50 µm (**c** and **f**). scRNA-seq data (**a**, **b** and **e**) are the integration of two independent experiments, each of them based on four biological replicates.

We showed that Lrig1[+] SC progeny residing in HFs distally located from the injury acquire distal memory (Fig. 1d–h). The histological analysis of the spatial expression of Krt6 confirms the spatial extent of Lrig1 GL memory cells resident in INFU (Fig. 3h), supporting distal priming. To further prove this, Lrig1 GL cells at 8w pw1 were isolated from either wound bed, distal memory or far away naïve areas and analysed by scRNA-seq (Extended Data Fig. 5a–j). To profile only the upper-HF memory cells, we sorted Ly6a/Sca-1[+] Lrig1 GL[30] (Extended Data Fig. 5a). Strikingly, the expression of the Transition Cluster marker genes (Fig. 3g) follows the same spatial gradient as Krt6 (Fig. 3h and

Extended Data Fig. 5i), proving that Lrig1 GL distal memory cells adapt to wound through priming (Fig. 3i).

Overall, these results indicate that, consequently to an injury, wound-distant Lrig1+ SC progeny occupies the INFU where it remains transcriptionally pre-activated even when the damage has been resolved and a new homeostasis has been reset. We show that priming is opted by epithelial cells as a memory of tissue repair in sites distant from the wound.

## Characterization of distal priming

To dissect the transcriptional basis of wound priming, we characterized the expression profile of the cells in trajectory D, separating the first healing process (0w to 1w pw1) from the second one (8w pw1 to 1w pw2). The data highlight the existence of cell plasticity genes that are specifically induced at higher levels during the second healing in the Transition Cluster (Fig. 4a,b and Supplementary Table 3) and associated with migration-related GO terms, as well as glycolysis, hypoxia and mTORC1 signalling terms (Fig. 4b (right) and Extended Data Fig. 5i,j).

After gathering the cells of the Transition Cluster on the basis of timepoints and clusters, we confirm within each GO term the intermediate transcriptional state of Lrig1 GL cells at 8w pw1 that lies between 0w and the healing phases (Extended Data Fig. 6a,b). Consistent with the GO terms analysis, the infundibular Lrig1 GL cells display a more elongated shape and increased phospho-S6 levels at 8w pw1, compared with 0w. Both these features are spatially organized with a gradient pattern in distal HFs (Fig. 4c,d and Extended Data Fig. 6c).

Concerning the enriched metabolism-related GO terms, we assessed the glycolytic/hypoxic state evaluating the levels of Glut1 (Slc2a1), a marker of highly glycolytic cells[31]. As predicted, Glut1 is upregulated specifically in the actively repairing cells of the Transition Cluster and strongly induced at 1w pw2 compared with 1w pw1 (Fig. 4e and Extended Data Fig. 6d–g). MitoTracker staining validates the Oxidative Phosphorylation GO term (Extended Data Fig. 6h).

From a spatial point of view, Glut1, as a marker of primed cells at 8w pw1, displays a gradient pattern towards distal memory areas (Fig. 4f and Extended Data Fig. 5i), matching Krt6 and cell shape data and corroborating the spatial extent of memory.

Finally, isolated distal memory cells (Glut1+tdTomato+) have an enhanced migratory potential with respect to naïve counterpart (Glut1−tdTomato+), validating the functional implications of distal priming on cell fitness (Fig. 4g and Extended Data Fig. 6i).

Overall, wound priming relies on high metabolism and enhanced migration rate. Strikingly, the integration of scRNA-seq with the histological validation of the molecular signature of priming shows an unexpected large spatial extent of wound memory in infundibular Lrig1 GL cells.

## Epithelial priming is lineage specific

Our previous data show that Lgr5 progeny acquire memory after injury with an individual transcriptional programme (Figs. 1c and 2a). To dissect the wound adaptation of Lgr5+ bulge SC progeny, we performed scRNA-seq. Importantly, no Lgr5 GL cells occupy the INFU when homeostasis is re-established at 8w pw1 (Extended Data Fig. 7a–d), demonstrating that only activated Lrig1+ SCs give rise to the infundibular primed cells. The Transition Cluster contains actively repairing and proliferating Lgr5 GL cells (1w pw1 and 1w pw2), positive for Glut1 and Krt6a (Extended Data Fig. 7e,f). Targeted analysis of the GO terms enriched in Lrig1 GL cells reveals that Lgr5 GL cells at 8w pw1 are transcriptionally comparable to cells at 0w (Extended Data Fig. 7g), suggesting the absence of a substantial cell priming.

To directly compare the two lineages, we performed pseudotime analysis on Lgr5 GL cells. The Niche of Origin-Bulge Cluster 5 is used as trajectory D starting point (Extended Data Fig. 7h). We find that: (1) priming of Lgr5 SCs in their niche of origin is extremely limited compared with Lrig1+ SCs (cell subset 'A'), as bulge-derived progeny adapts through a trained memory strategy[15]; (2) compared with first healing, the enhanced transcription at 1w pw2, is lower for Lgr5 GL cells with respect to Lrig1 GL cells, consistently with their lower contribution to re-epithelialization (cell subset 'B'); (3) only Lrig1 GL cells have a primed adaptive memory (cell subset 'C'). Interestingly, although the GO terms enriched for cell plasticity genes are similar between the two lineages, the individual genes are mostly lineage specific (Extended Data Fig. 7i,j). Thus, in the context of skin full-thickness wound, the Lgr5+ SCs display a trained adaptation to wound, as shown in superficial wound[15]. However, priming is peculiar to the Lrig1 SC progeny.

## Chromatin landscape of distal priming

To evaluate the consequences of wound adaptation at chromatin level we performed assay for transposase-accessible chromatin with sequencing (ATAC–seq) of Lrig1 GL cells from the distal memory areas (Fig. 5a–f). We observe a global chromatin opening at 1w pw1 and 1w pw2, as well as at 8w pw1, compared with 0w (Fig. 5a). Most chromatin opening events at 8w pw1 are shared by both 1w pw1 and 1w pw2 conditions, with 1,665 wound memory peaks (Fig. 5b–d). GO analysis of the genes associated with memory peaks reveals an enrichment of the same GO terms (Fig. 5e) identified for the primed transcripts (Fig. 4b).

To evaluate if cell-intrinsic priming relies on chromatin changes, we performed an additional ATAC–seq on cultured Lrig1 GL cells, sorted from distal memory areas and far away control areas at 8w pw1 (Fig. 5g–j). After 7 days in culture, the genes associated with the gained peaks of in vitro distal memory cells are consistent with in vivo memory (1,665 peaks) and they belong to the same GO categories (Fig. 5h,i). In addition, DNA motif analysis shows that the accessible genomic loci of in vivo and cultured memory cells share the top transcription factor consensus (Fig. 5f,j). Strikingly, the results of GO enrichment analysis from in vitro ATAC–seq data are confirmed by the gene set enrichment analysis (GSEA) of in vitro RNA-seq data (Fig. 5k).

We conclude that the transcriptional priming is supported by an increase in chromatin accessibility in genes associated to metabolism and migration, acquired after the first injury, and maintained ever since. Importantly, all these features are intrinsic to distal Lrig1 GL

**Fig. 5 | Chromatin remodelling of Lrig1 GL distal priming. a–f**, ATAC–seq of sorted Lrig1 GL upper-HF cells (Sca-1+) at the four timepoints in distal areas (Distal 3–7 mm from wound bed). Venn diagrams of ATAC–seq peaks (logFC > 0.5, *P* < 0.05). Peaks from distal area (Distal) at 1w pw1, 8w pw1 and 1w pw2 respect to 0w (gained peaks in red) (**a**) and shared peaks (red) (**b**). Density plots (**c**) depict ATAC–seq signals ±1 kb of 1,665 distal memory peaks (up) and heat map of the signal score of individual peaks (down). Snapshots of genomic loci of representative distal memory peaks (**d**). GO analysis (**e**) of genes associated with gained peaks at 1w pw1, 8w pw1 and 1w pw2 versus 0w. −log₁₀ of adjusted *p*-value (AdjP) is represented by colour scale and gene number by dot size. Transcription factor (TF) motifs (**f**) enriched in the ATAC–seq peaks gained at 8w pw1_Distal versus 0w, based on de novo motif discovery. **g–j**, ATAC–seq of Lrig1 GL cultured cells from distal memory (Distal 3–7 mm from wound bed) and control regions (Ctrl distance >2 cm from wound) at 8w pw1. Density plots (**g**) depict ATAC–seq signals ±1 kb of gained peaks (up) and heat map of the signal score of individual peaks (down) in Distal versus Ctrl cells. Venn diagram (**h**) of genes associated to gained peaks in 8w pw1_Distal in vitro and associated with distal memory peaks in vivo. Random permutation *p*-value (*P*) is shown. GO analysis (**i**) of genes associated with gained peaks Distal versus Ctrl as −log₁₀(AdjP). Dashed line indicates significance. TF motifs (**j**) enriched in the ATAC–seq peaks gained in 8w pw1_Distal versus 8w pw1_Ctrl, based on de novo motif discovery. **k**, RNA-seq of cultured Lrig1 GL cells from Distal and Ctrl. GSEA ranking using the differential gene expression. NES, normalized expression score. Motifs with a match score of >0.8 to a known motif are ranked (**f** and **j**). ATAC–seq data (**a–j**) are the integration of two independent experiments, each of them based on two biological replicates.

memory cells and exist even in the absence of the in vivo physiological microenvironment.

## Long-term maintenance of primed progenitors

It has been shown that in epidermal cells wound memory can last up to 80 days[15]. To evaluate how long distal priming is preserved during ageing, we analysed the epidermis of Lrig1 GL mice 10 months after the first injury (Fig. 6a). The closure advantage at 1w pw2(40) with respect to 1w pw1(40), is still evident (Fig. 6b,c). Importantly aged wound-distal Lrig1 GL cells show both the HF engagement phenotype (Fig. 6d) and an ex vivo enhanced migratory ability when compared with the untrained counterpart (Fig. 6e and Extended Data Fig. 7k).

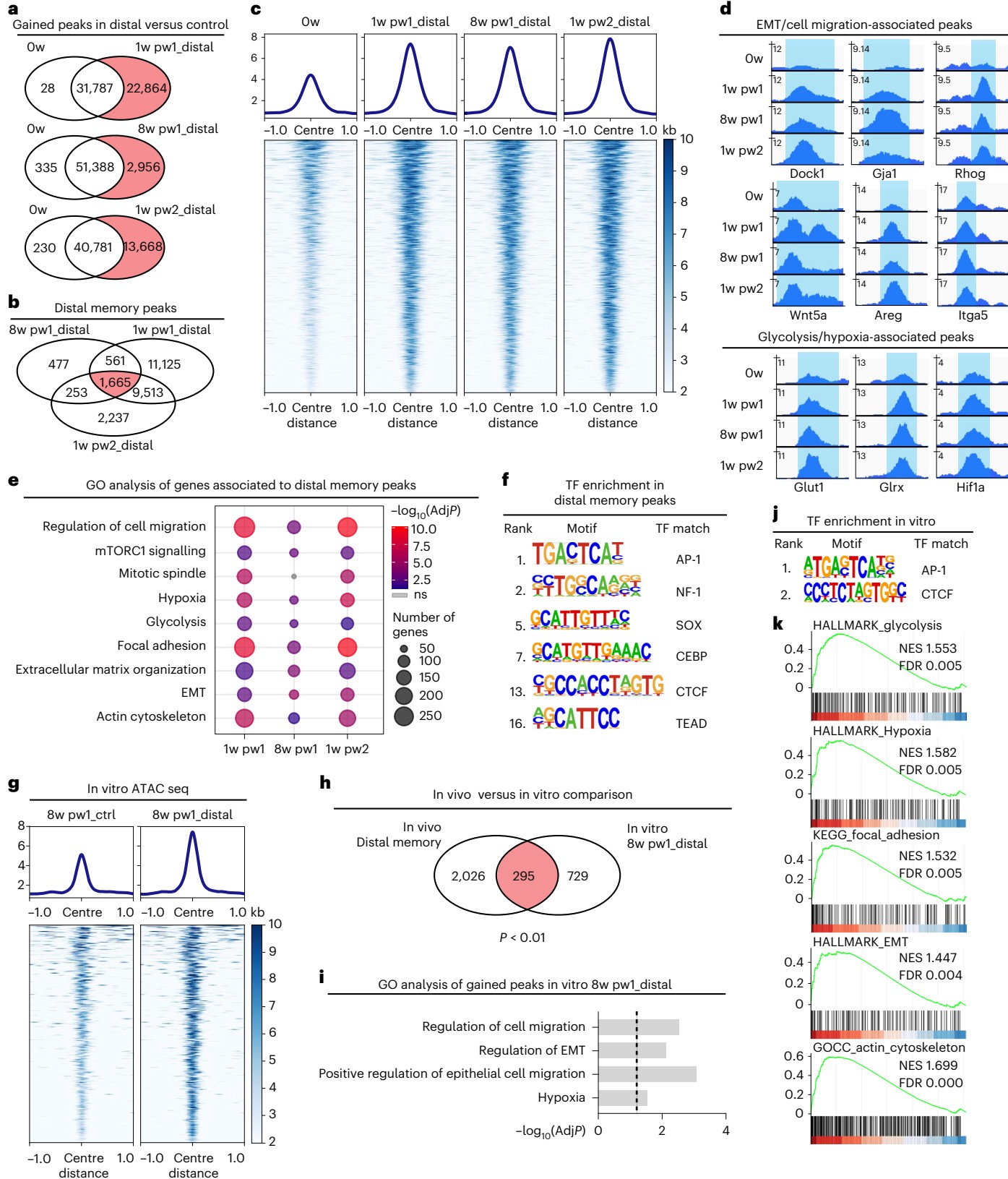

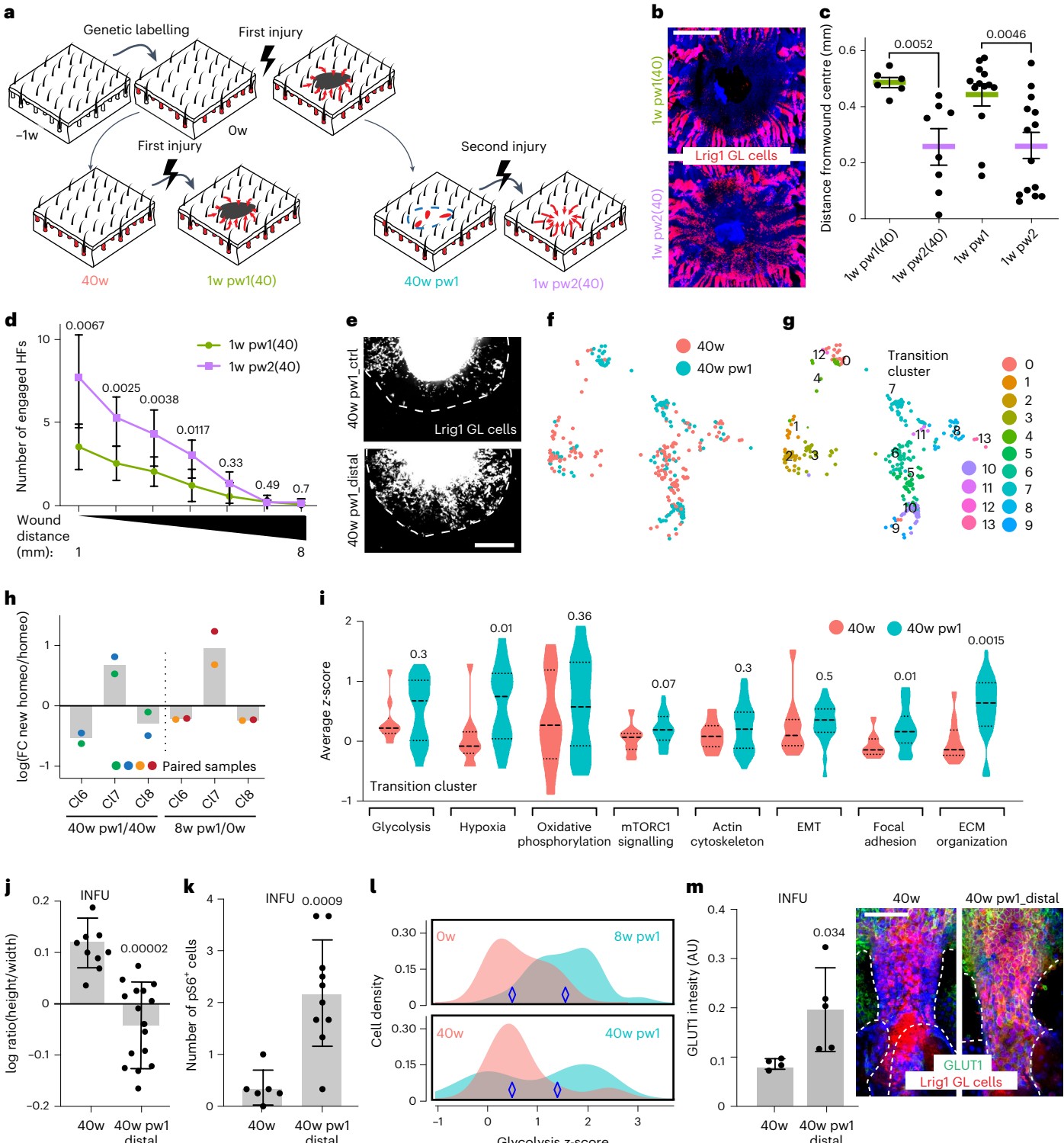

**Fig. 6 | Wound-distal primed cells are preserved in ageing. a,** Settings to evaluate long-term priming in Lrig1 GL skin: (−1w) genetic labelling; (0w) injury or not; 40 weeks after labelling, wound on both unwounded (40w) and wounded (40w pw1) mice; 1 week after injury, 1w pw1(40) or 1w pw2(40). **b,c,** Epidermal whole mounts showing Lrig1 GL tdTomato⁺ cells at wound site at 1w pw1(40) and 1w pw2(40) (**b**) and quantification of the distance from wound centre, compared with young timepoints (**c**). n = 6 (1w pw1(40)), n = 8 (1w pw2(40)), n = 13 (1w pw1), n = 14 (1w pw2). **d,** Number of engaged HFs. n = 6 (1w pw1(40)), n = 7 (1w pw2(40)). **e,** Ex vivo migration assay. Pictures of Lrig1 GL tdTomato⁺ cells exiting from Distal or Ctrl skin explants collected as Fig. 2c. **f,g,** scRNA-seq of Lrig1 GL cells at 40w and 40w pw1. UMAP of cells coloured by timepoints (**f**) or clusters (**g**). **h,** Comparison of cell lineages derived from Lrig1 SCs in two old (40w and 40w pw1) and two young (0w and 8w pw1) homeostasis, as ratio between them.

**i,** Violin plot of the average expression of memory GO terms (from Fig. 4b) comparing 40w and 40w pw1. Data are median with 25th and 75th percentiles. n = 10 (40w), n = 32 (40w pw1) cells. **j,k,** Quantification of cell shape, as height–width ratio (n = 9 (40w), n = 16 (40w pw1_Distal)) (**j**) and pS6 (n = 6 (40w), n = 10 (40w pw1_Distal)) (**k**) in INFU at 40w and at 40w pw1 in distal memory areas (Distal 3–7 mm from wound site). **l,** Density plot for glycolysis gene signature in Lrig1 GL cells from Transition Cluster in old (40w and 40w pw1) and young (0w and 8w pw1) homeostasis. **m,** Glut1 quantification (left) in the INFU whole mounts (right) at 40w and at 40w pw1 Distal. n = 4 (40w), n = 5 (40w_Distal). scRNA-seq data (**f–i**) are the integration of two independent experiments, each of them based on four biological replicates. P-value from a two-tailed t-test. Data are mean ± s.d., if not differently indicated. Scale bars: 1 mm (**b** and **e**); 50 μm (**m**).

Furthermore, the scRNA-seq profiling of aged homeostatic skin (40w) was compared with the new homeostasis at 40w pw1 (Fig. 6f,g and Extended Data Fig. 7l,m). Strikingly, at 40w pw1 the infundibular primed cells are maintained in the Transition Cluster, with respect to the other differentiated cells derived from Lrig1+ SCs (Fig. 6h). However, despite a partial memory loss (Extended Data Fig. 7n,o), the GO terms enriched in the Transition Cluster in young 8w pw1 (Fig. 4b) are still overexpressed ~1 year after wound when compared with 40w (Fig. 6i). Histological validation comparing wound-distal infundibular cells at 40w pw1 with cells at 40w, shows cell elongation and increased levels of pS6, Glut1 and MitoTracker (Fig. 6j–m and Extended Data Fig. 7p,q).

Thus, we demonstrate that distal priming is largely stable, functional and maintained in time by memory progenitors that are preserved in the aged INFU.

## H2AK119ub reduction mediates distal priming

To identify epigenetic regulators of priming, we pharmacologically targeted in vivo five histone-modifying enzymes. Pre-treatment with PRT4165 leads to a more efficient healing at 1w pw2 (Extended Data Fig. 8a–c) and, more importantly, increases the engagement of HFs located distally from the injury (Extended Data Fig. 8c,d). This drug inhibits the activity of Ring1a/Ring1b, components of the Polycomb repressive complex 1 (PRC1), responsible for the monoubiquitination of lysine 119 on histone H2A (H2AK119ub)[32–34]. In homeostasis, the JZ and the INFU already express lower levels of this histone modification, when compared with other epidermal compartments (Extended Data Fig. 8e). The H2AK119ub repressive mark decreases in INFU at 1w pw1 and 1w pw2, and it is not restored at the original 0w levels at 8w pw1 or 40w pw1 (Fig. 7a and Extended Data Fig. 8f,g). Since the decrease in H2AK119ub is also evident in the INFU of wound-distant HFs, following the spatial distribution of distal memory, we hypothesized H2AK119ub to be a key functional component in distal priming through a transcriptional de-repression mechanism. We pre-treated Lrig1 GL epidermis with PRT4165 (PRC1i) and we induced an injury (Fig. 7b). PRC1i elicits an enhanced healing rate and wound-distal HFs are moderately engaged (Fig. 7c and Extended Data Fig. 8h,i).

To assess whether PRC1i promotes the derepression of genes associated with wound priming, we performed a scRNA-seq on vehicle or PRC1i-treated Lrig1 GL mice. Even in the absence of an injury, PRC1i activates several pathways previously identified in the physiological priming of Lrig1 GL cells in the Niche of Origin-JZ Cluster (Fig. 7d,e and Extended Data Fig. 9a–d).

To assess whether PRC1i can trigger similar chromatin changes to a wound, we performed ATAC–seq. Strikingly, the genes associated with PRC1i-specific chromatin opening are consistent with the ones associated with the 1,665 memory peaks (Fig. 5b) and they belong to the same GO categories (Fig. 7f–j). In addition, the transcription factor binding sequences that we identified are mostly the same of the physiological 1,665 memory peaks (Fig. 7k). Therefore, we conclude

that the PRC1i is able to mimic wound effect and memory onset at the transcriptional and chromatin level.

Three days post treatment, these chromatin and transcriptional changes trigger Lrig1 SC progeny to move from the JZ niche into INFU towards IFE, through a cell mechanism that involves enhanced migration and metabolism (Fig. 7l–n and Extended Data Fig. 9e–j), thus recapitulating features of priming.

Differently to Lrig1 GL cells, PRC1i does not lead to either distal HFs engagement or enhanced wound closure in Lgr5 or Gata6 GL skin. However, Lgr5 GL cells only exit their niche of origin towards the upper bulge, in accordance with previous observations[35] (Extended Data Fig. 9k–m).

In conclusion, H2AK119ub in the INFU decreases physiologically after the first injury and the original level is not resorted at 8w pw1 new homeostasis, following the distal spatial distribution of priming. The physiological reduction of H2AK119ub is functional to memory onset, mediating chromatin remodelling and leading to the de-repression of primed genes in distal memory cells derived from Lrig1+ SCs.

## Wound memory promotes tumour onset with a spatial gradient

Wound memory has a beneficial impact on long-term tissue fitness related to skin repair. However, since wound healing and cancer share many hallmarks[36], and the epigenetic landscape of cell-of-origin can define epidermal tumour subtypes with differential features of epithelial-to-mesenchymal transition (EMT)[37], we hypothesized that wound priming might impact on tumourigenesis, in line with recent observations[16,38]. In addition, because it has a long-range spatial distribution, the detrimental effect might follow this spatial trend, building on the epigenetic field cancerization phenomenon that predisposes to tumour onset[18,19].

We observe that the memory of an antecedent lesion exacerbates the Lrig1 GL cells response to an oncogenic stimulus (Extended Data Fig. 10a–d), potentially triggering cancer.

To verify these hypotheses, we induced carcinoma formation in mice through UVB[39] (Extended Data Fig. 10e), to avoid papilloma formation[40]. The comparison between wounded (Wd), UVB-treated (TS) and wounded and UVB-treated (Wd&TS) tail skin shows the onset of epidermal dysplasia, typical of actinic keratosis or early squamous cell carcinoma in situ (eSCC)[41] specifically in Wd&TS (Fig. 8a and Extended Data Fig. 10f,g). eSCCs derive mainly from Lrig1 GL primed cells (Extended Data Fig. 10h). Strikingly, eSCCs follow a spatial gradient from wound, where tumour incidence is highest, towards distal memory regions (Fig. 8a). In this pre-cancerous context, we also observe a spatial distribution of the anti-correlation of H2AK119ub and γ-H2A.X, as a marker of DNA damage[42]. Indeed, the accumulation of DNA damage is found where H2AK119ub is reduced, in the distal memory areas (Fig. 8b). This scenario is reminiscent of a field cancerization (also termed as field change or cancer field effect) phenomenon in which cells in wide areas

---

**Fig. 7 | Transcriptional de-repression is a functional component of wound memory. a,** Epidermal whole mounts of H2AK119ub in INFU at 0w and 8w pw1, 1w pw1 and 1w pw2, in distal memory area (5 mm from wound bed). **b,** Setting for PRT4165 treatment in Lrig1 GL mice: day 0, genetic labelling through 4-Hydroxytamoxifen (4OHT); day 7-9-11, PRT4165/vehicle treatments; day 11, 6 h after the last treatment scRNA-seq, ATAC–seq or a full-thickness injury; day 18 (1w pw1), histology. **c,** Epidermal whole mounts of Lrig1 GL tdTomato+-treated cells. **d,e,** scRNA-seq data from Lrig1 GL-treated cells. **d,** UMAP of cells coloured by cluster. **e,** Heat map of known memory GO terms. Gene expression is shown per cluster as ratio between PRT4165- and vehicle-treated cells. **f–k,** ATAC–seq of sorted Lrig1 GL upper-HF cells (Sca-1+) from PRT4165- and vehicle-treated skin. **f,** Genomic loci gained after PRT4165 treatment. **g,** Density plots of ATAC–seq signals ±1 kb of peaks (logFC > 0.5, P < 0.05) (top) and heat map of the signal score of individual peaks (bottom). **h,** Venn diagram showing the ATAC–seq peaks gained in PRT4165-treated versus vehicle-treated (red). **i,j,** Analysis of

genes associated with gained peaks in PRT4165-treated versus vehicle-treated. **i,** Intersection between these genes and the genes associated to distal memory peaks. Random permutation p-value (P) is shown. **j,** GO analysis, plotted as −log10 of adjusted p-value (AdjP). Dashed line underlines significance. **k,** Transcription factor (TF) motifs enrichment for PRT4165 gained peaks (to vehicle), based on de novo motif discovery. Ranking of motifs with a match score of >0.8 to a known motif. **l,** 3D surface plots of Lrig1 GL cells from HF whole mounts. Dashed line represents the IFE–HF boundary while arrows the IFE exit. **m,** Quantification (left) and images (right) of ex vivo assay of treated Lrig1 GL skin (red channel). n = 10 explants. **n,** Glut1 quantification (left) and INFU whole mounts (right) of treated epidermis. n = 6 mice. P-value from a two-tailed t-test, if not differently indicated. Data are mean ± s.d. Scale bars: 50 μm (**a, l, m** and **n**); 1 mm (**c**). scRNA-seq (**d** and **e**) and ATAC–seq (**f–k**) data are the integration of two independent experiments, each of them based on four and two biological replicates, respectively.

within a tissue are affected by carcinogenic alterations both genetic and epigenetic[18,19]. To verify if epigenetic field change is linked to priming, we performed ATAC-seq of Lrig1 GL cells, upon chronic UVB irradiation, in presence or absence of a previous healed wound (Fig. 8c–h). As for wound priming a global chromatin opening is preserved in Lrig1 GL cells resident in wound distal (Distal) in comparison with naïve (Ctrl) areas upon irradiation (Fig. 8c–e). The genes associated to the gained peaks in Distal versus Ctrl areas are consistent with the memory ones

(1,665 memory peaks), and belong to the same GO categories (Fig. 8f,g and Extended Data Fig. 10i); even the DNA motif analysis shows shared transcription factor binding sequences (Fig. 8h). Thus, an epigenetic field cancerization event occurs after wound repair in distal memory areas, consequently to H2AK119ub reduction and wound priming, and it is maintained during the pre-cancerous stages.

To further investigate the link of the epigenetic field effect and wound memory, we induced tumour formation in back skin, where fully

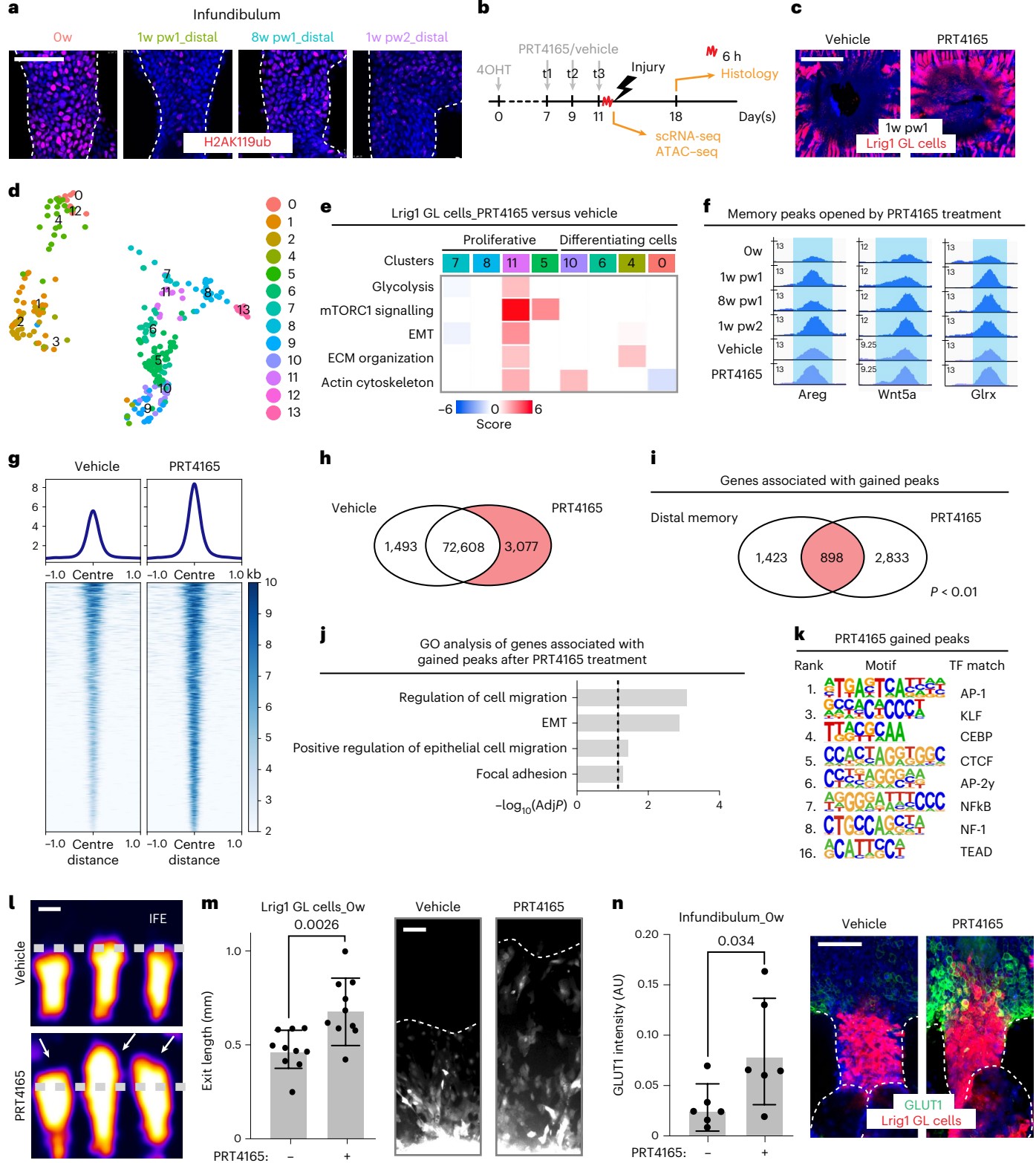

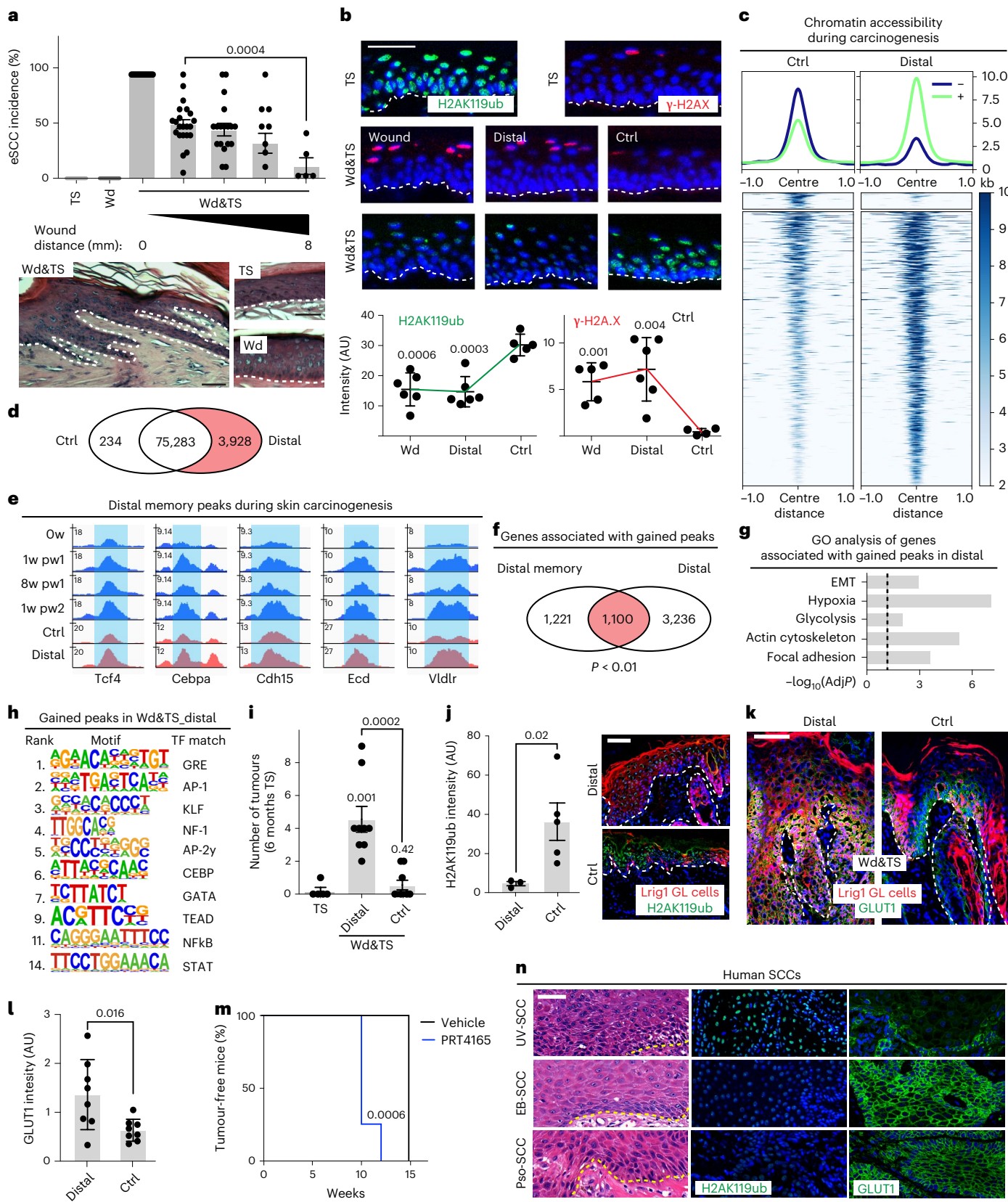

developed SCCs were expected[43]. Wd&TS mice have faster SCC onset, characterized by a more severe phenotype and an increased percentage of Lrig1 GL cells within the tumour mass with respect to TS mice (Fig. 8i and Extended Data Fig. 10j–s). Remarkably, all these phenotypes follow the gradient spatial distribution of distal memory cells.

The level of H2AK119ub, whose reduction was functional in priming, is lower in Wd&TS tumours when compared with TS counterpart and negatively correlates with the memory marker Glut1 (Fig. 8j–l). Moreover, the treatment with PRC1i accelerates DNA damage accumulation and tumour onset in hairless mice (Fig. 8m and Extended

**Fig. 8 | Wound priming initiates field cancerization. a,** eSCC onset in wounded (Wd), UVB-treated only (TS) or wounded and UVB-treated (Wd&TS) tail skin. Incidence (top) and representative H&E (bottom). Data are mean ± s.e.m. $n = 5$ (TS, Wd), $n = 22$ (Wd&TS). **b,** Skin section stained with H2AK119ub or γ-H2A.X (top) and quantifications (bottom) in wound bed (Wb), distal memory (Distal 3–7 mm from Wb) and far control (Ctrl >2 cm from Wb) areas. $n = 4$ (Ctrl), $n = 6$ (Distal, Wb). **c–h,** ATAC–seq of sorted Lrig1 GL upper-HF cells (Sca-1⁺), upon UVB treatment from Distal and Ctrl. **c,** Density plots depict ATAC–seq signals ±1 kb of deregulated peaks (logFC > 0.5, $P < 0.05$) (top) and heat map of the signal score of individual peaks (bottom). **d,** Venn diagram of peaks in Distal versus Ctrl. Distal gained peaks in red. **e,** Snapshots of genomic loci associated to gained peaks in Distal UVB-treated skin. **f,** Overlap (red) between the genes associated to gained peaks in Distal UVB-treated skin (respect to Ctrl) and to Distal memory peaks. Random permutation $p$-value ($P$) is shown. **g,** GO analysis of genes associated to gained peaks in Distal versus Ctrl, measured as $-\log_{10}$ of $p$-value ($P$). Dashed line underlines significance. **h,** Transcription factor (TF) motifs enriched in gained peaks Distal versus Ctrl, based on de novo motif discovery. Motifs with a match score of >0.8 to a known motif are ranked. **i,** Number of tumours. $n = 5$ (TS), $n = 9$ (Wd&TS). **j,** Quantification (left) and skin section staining (right) of H2AK119ub. $n = 3$ (Ctrl), $n = 5$ (Distal). **k,l,** Glut1 expression in Distal or Ctrl area of Wd&TS samples. Skin section images (**k**) and quantification (**l**). $n = 8$ Wd&TS. **m,** Tumour incidence in vehicle versus PRT4165-treated hairless mice. $n = 7$ (vehicle), $n = 4$ (PRT4165). Statistic: Mantel–Cox test. **n,** H2AK119ub and Glut1 staining on human SCC sections. $P$-value from a two-tailed $t$-test, if not differently indicated. Data are mean ± s.d., if not differently indicated. Scale bars: 50 μm (**a**, **b**, **j**, **k** and **n**). ATAC–seq (**c–h**) data are the integration of two independent experiments, each of them based on two biological replicates.

---

Data Fig. 10t), thus recapitulating the physiological anticorrelation between DNA damage and H2AK119ub loss observed in the context of UVB-treated skin (Fig. 8b). Overall, long-term wound priming promotes skin tumourigenesis following its spatial gradient distribution, in accordance with the wound-induced epigenetic field effect scenario.

Finally, we evaluated the relevance of our findings in the human SCC context, comparing cutaneous SCCs of different causal origins: epidermolysis bullosa-derived SCCs (EB-SCCs); SCCs from psoriatic skin (Pso-SCCs), since these patients have higher risk of tumourigenesis, possibly through a mechanism that involves wound and inflammatory memory[44,45]; SCCs derived from simple UV exposure during life (UV-SCCs). Consistently with our murine observations, Pso-SCCs and EB-SCCs, have a decreased H2AK119ub and an enhanced Glut1 expression when compared to UV-SCCs (Fig. 8n and Extended Data Fig. 10u), pointing out a memory–tumourigenesis link in humans.

Altogether, we showed that wound memory establishes an epigenetic field change that relies on H2AK119ub reduction, thus enhancing the incidence of squamous cell carcinomas (SCCs).

## Discussion

Innate immune cells adapt to a stressful event, keep an epigenetic memory of it and respond faster to a second assault[17,46,47]. The spectrum of epithelial cell responses and their adaptation mechanisms to a stressful event has started to be understood and trained wound memory has been reported for bulge hair follicle stem cells[15]. We demonstrated that only an epidermal SC progeny is wound primed during healing and maintained in the newly established homeostasis. Indeed, the transcriptional profile of these memory cells, which is intermediate between repairing and homeostatic cells, is a classical feature of priming.

Our results demonstrated that the memory has a spatial extension that is much larger than expected. In particular, wound-primed progenitors derived from Lrig1⁺ SCs are present up to ~7 mm away from the injury margins. Since the skin is the largest organ in mammals this spatially large adaptation involves tissue at sub-organ scale. However, in other smaller epithelia it is likely that this memory might impact the whole organ.

Several transcription factors are functional determinants in inflammatory memory[12]; however, the epigenetic regulators have not been identified. Here we demonstrate that wound distal priming of Lrig1 GL cells relies on a chromatin opening led by the reduction of a transcriptional repressive histone mark. In particular, we observed a clear correlation between the spatial distribution of decreased levels of H2AK119ub in the INFU after wound and the wide location of the wound priming. Our data conferred to the Lrig1⁺ SCs a repair-specific role. Thus, it is easy to speculate that specific cells in other epithelia might also have this unique innate potential.

mTOR and glycolysis have been identified as the metabolic basis for both epidermal response to wound healing[48] and trained immunity in monocytes[49]. This metabolic state is also relevant in epithelial cell priming and associated with a reduction in H2AK119ub. This link might be functional in other cellular contexts.

Carcinogenic alterations that lead to field cancerization can be both genetic and epigenetic[18,19]. While it is often difficult to understand which ones come first, our data highlight a sequential order in which epigenetic alterations such as a reduction of H2AK119ub, and a subsequent specific chromatin opening, are the initiators of field cancerization. Our data support the hypothesis that the reduction of H2AK119ub, a functional feature of wound-distal priming, directly promotes tumourigenesis. Our results together with recent findings in diversified cellular contexts[50,51], support the hypothesis that loss of a key histone repressive mark promotes tumourigenesis. This raises a general concern with respect to the therapeutic intervention on repressive chromatin factors that could be beneficial in regenerative medicine but detrimental in oncology.

## Online content

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

## Methods

The experiments in this manuscript are in compliance with relevant guidelines and ethical regulations.

### Mouse strains

Maintenance, care and experimental procedures have been approved by the Italian Ministry of Health, in accordance with Italian legislation (authorization no. 117/2018-PR), and the institutional review board of the Hokkaido University Graduate School of Medicine (authorization no. 22-0028). Rosa26-fl/STOP/fl-tdTomato, Lgr5-EGFP-ires-CreERT2, Gata6-EGFP-ires-CreERT2 and Lrig1-EGFP-ires-CreERT2 have been previously described[11,20,21,52]. Both sexes were used for the experiments, if not differently indicated. SKH-1 hairless mice were purchased from Charles-River Laboratories and maintained in specific pathogen free conditions.

For lineage tracing, CreERT2 strains were crossed with Rosa26-fl/STOP/fl-tdTomato strain. Genetic labelling was induced in epidermis of 6–8-week-old mice with a single topical administration of 75 μg of (Z)-4-Hydroxytamoxifen (Sigma-Aldrich) (15 mg ml$^{-1}$ in acetone).

### Full-thickness skin wound

One week post tamoxifen treatment (0w), 7–9-week-old mice were anaesthetized, and full-thickness wounds were made with a circular biopsy punch in tail (2 mm) or dorsal (5 mm) skin.

New homeostasis re-establishment after wound was assessed on the basis of the following features: (1) complete re-epithelialization, assessed through an haematoxylin and eosin (H&E) staining of skin sections; (2) re-establishment of comparable epidermal differentiated cell layers with 0w, where the differentiation marker FABP5 was used to evaluate this feature; (3) restoration of a comparable HF cycle phase to 0w; (4) immune infiltrate resolution, determined by flow cytometry.

### Epidermal and dermal whole-mount

Tail epidermal or dermal whole mounts were prepared as previously described[20]. Primary antibodies were diluted in PB buffer (0.5% skimmed milk, 0,25% fish gelatin and 1% of Triton X-100 in PBS) and incubated at the following dilutions: anti-Keratin 6A (rabbit, 1:200, BioLegend 905701), anti-Ubiquityl-Histone H2A (Lys119) (clone D27C4) (rabbit, 1:1,000, CST 8240), anti-phospho-S6 Ribosomal Protein (Ser235/236) (clone D57.2.2E) (rabbit, 1:200, CST 4858), anti-Ki67 (rabbit, 1:100, Abcam 16667), anti-E-cadherin (clone 24E10) (rabbit, 1:200, CST 3195), anti-Glut1 (SLC2A1) (EPR3915 clone) (rabbit, 1:200, Abcam ab115730) and Alexa-647 conjugated Phalloidin A (Thermo Fisher). Secondary antibodies together with DAPI (Thermo Fisher, 1 μg ml$^{-1}$) were incubated 2 h at room temperature. Tissues were mounted using Mowiol 4–88 Reagent (Sigma-Aldrich) mounting solution. MitoTracker staining (MitoTracker Deep Red FM_M22426, Thermo Fisher) was performed in whole-mount epidermis according to datasheet instructions, incubating the samples at 37 °C 10 min in 50 nM MitoTracker solution, before fixation.

### Immunostaining on sections

Frozen and paraffin sections were blocked with PB buffer for 1 h at room temperature. Primary, secondary antibodies and DAPI were incubated as above. The following primary antibodies were used on frozen sections: anti-Ubiquityl-Histone H2A (Lys119) (clone D27C4) (rabbit, 1:1,000, CST 8240), anti-Glucose Transporter Glut1 (SLC2A1) (clone EPR3915) (rabbit, 1:200, Abcam ab115730) anti-CD3 (clone 17.A2) (FITC conjugated, rat, 1:200, Miltenyi Biotec 130-119-135), anti-mFABP5 (goat, 1:500, RD System AF1476), anti-F4/80 (clone REA126) (rat, 1:50, Abcam ab6640) and anti-TNFα (clone MP6-XT22) (Brilliant violet 421 conjugated, rat, 1:50, BioLegend 506327). The following primary antibodies were used on paraffin sections: anti-Ubiquityl-Histone H2A (Lys119) (clone D27C4) (rabbit, 1:1,000, CST 8240), anti-Phospho-Histone H2A.X (Ser139) (rabbit, 1:400, CST 9718) known as γ-H2A.X and anti-SLC2A1 (Glut1) (rabbit, 1:100, Sigma-Aldrich HPA031345).

### Immunostaining on ex vivo culture

Skin explants and exiting cells were fixed and blocked. The following primary antibodies were incubated: anti-Keratin 6A (rabbit, 1:500, BioLegend 905701), anti-cytokeratin 14 (clone LL002) (mouse IgG3, 1:500, Invitrogen) and GM130 (clone 35) (mouse, 1:400, BD Pharmingen).

### Image acquisition

Whole-mount and section immunofluorescence were acquired with confocal microscopes: Leica TCS SP5 Tandem Scanner and Leica TCS SP8 Tandem Scanner equipped with 20×/40× or 63× immersion objectives (Zeiss). Z-stacks were acquired at 400 Hz with an optimal stack distance and 1,024 × 1,024 dpi resolution and projected with the LAS AF software package (Leica Microsystems) as maximum intensity projections. For stereoscopic images Leica MZ16FA stereomicroscope was used. H&Es staining were acquired using Olympus Bx41 or Leica DM6 microscope equipped with a 4× or 10× objectives.

### Image analysis

Digital images were processed and analysed using Fiji (https://imagej.nih.gov/ij/). Fluorophore intensity was measured as integrated density (IntD) in the selected ROI. The staining intensity (AU) was calculated as the ratio between the IntD of the antibody signal and the IntD of the DAPI or TOPRO3 signal in the same ROI. 3D surface plots of HFs were obtained using the Fiji function 3D surface plot, where five HF triples were overlapped and shown as pseudo-colour Fire. For phalloidin-A staining segmentation, the MorphoLibJ plugin with the Morphological Segmentation function was used. The results were displayed with the Catchment basins option.

### Keratinocyte isolation and culture

Tail skin was dissected and incubated in trypsin EDTA (Thermo Fisher, 0.25% in PBS) with the dermis side down overnight at 4 °C. The day after, the epidermis was peeled off and chopped with two scalpels for 1 min. Isolated cells were plated and cultured in keratinocyte medium (low-calcium Dulbecco's modified Eagle medium (Thermo Fisher) supplemented with 10% FBS (Sigma-Aldrich), 100 U ml$^{-1}$ penicillin–streptomycin (Thermo Fisher), HCE cocktail consisting of hydrocortisone 0.5 μg ml$^{-1}$ (Sigma-Aldrich), insulin 5 μg ml$^{-1}$ (Thermo Fisher), cholera enterotoxin $10^{-10}$ M (Sigma-Aldrich) and EGF 10 ng ml$^{-1}$ (PeproTech) and cultured at 34 °C in a humidified atmosphere, with 8% $CO_2$.

For in vitro mini-bulk RNA-seq and ATAC–seq Sca-1$^+$ tdTomato$^+$ cells were sorted and cultured for 7 days over a feeder layer of mitomycin-treated NIH/3T3 cells. At day 7, cells were collected, and tdTomato$^+$ were sorted to eliminate feeder cells and processed as indicated in mini-bulk RNA-seq and ATAC–seq paragraphs.

### In vitro adhesion and survival assay

Isolated epidermal cells were counted (tdTomato$^+$) with FACS Verse (day −1), plated on six-well plates (Corning) and treated 2 h with 4 μg ml$^{-1}$ of mitomycin C, to stop proliferation. Cells were counted again 24 h and 72 h after plating. Adhesion was calculated as the ratio of number of plated cells and cells at 24 h, while survival as the ratio of cells at 24 h and 72 h.

### In vitro time lapse migration assay

For time lapse migration assay, cells were plated in μ-Slide (Ibidi) with keratinocyte medium. Twenty-four hours after plating, dead cells were removed by PBS washes and attached cells were used for microscope acquisition. Plates were maintained in the incubator chamber of a confocal TCS SP5 microscope (Leica Microsystems) under controlled conditions (34 °C, 8% $CO_2$). Images were collected in two different positions for each well and acquired every 15 min for 16 h. Cell displacement was tracked automatically using the TrackMate plugin in Fiji with LogDetector settings. Mean track displacement was calculated for each sample

and plotted. For the normalized start position graph, we subtracted each point in the track with the coordinates of the starting point.

## Ex vivo migration assay

The ex vivo migration assay was performed as previously described[13]. Briefly, 2 mm punch biopsies were collected from tail skin. The explant was adhered to the bottom of a 24-well plate (Corning) and cultured in complete keratinocyte medium for 4, 7 and 9 days.

## Flow cytometry and fluorescence-activated cell sorting

To obtain single-cell suspensions for FACS, roughly $5 \times 5$ mm of tail skin is processed as described for keratinocyte isolation and filtered through a 70 μm cell strainer (VWR, Corning). Cells were blocked with 3% FBS in PBS for 20 min and then incubated 30 min with 0.3 μg of anti-Glut1 antibody (clone EPR3915) (Alexa Fluor 647 conjugated, Abcam ab195020) or 0.25 μg of anti-Sca-1 (clone E13-161.7) (PE/Cyanine7 conjugated, BioLegend 122514) on ice. For immune infiltrate quantification, six mice were randomly paired together to obtain the three biological replicates. For time 0w, five mice were used. Tail skin from wound bed and distal memory areas (Distal) was dissected and processed as previously described[53]. After blocking, the following antibodies were incubated 10 min at RT at the concentration of 1,5 μg ml$^{-1}$: anti-mouse CD45 (VioGreen, Miltenyi Biotec 130-110-803), CD11b (clone M1/70) (FITC, Miltenyi Biotec 130-110-803), CD3 (clone 17.A2) (FITC, Miltenyi Biotec_130130-119-135), γδTCR (clone REA633) (PE-Vio770, Miltenyi Biotec_130-123-290), F4/80 (clone REA126) (PE-Vio770, Miltenyi Biotec 130-118-320), MHC-II (APC, Miltenyi Biotec 130-102-139) and CD206 (clone C068C2) (PE, BioLegend 141706). Cells were acquired on BD-FACSVerse (BD Bioscience) and analysed with FlowJo (v10.8). CD45$^+$ cell populations analysed as previously reported[54,55]. For IL-17A staining the cells were fixed and permeabilized with the FOXP3 fix/per buffer set (BioLegend) following the manufacturer's instructions and stained with 0.4 μg anti-mouse IL-17A (PE, BioLegend 506903). Single cells were gated according to their physical parameters and acquired using the BD-FACSVerse (BD Bioscience). Cell sorting was performed with a 100 μm nozzle with a BD FACSAria II equipped with Diva software (BD Biosciences). Flow cytometry plots were generated using FlowJo (v10.8).

## Long-term memory

To evaluate the long-term maintenance of distal memory, 6–8-week-old mice were genetically labelled and, 1 week after, full-thickness wounds were made. A control group was maintained unwounded (40w). Forty weeks after the first wound (40w pw1), the 1-year-old mice were anaesthetized again, and a second overlapping wound was performed (1w pw2(40)), while a first wound was performed in the age-matched control unwounded group (1w pw1(40)). The wound closure and the HF engagement were assessed 1 week later (Fig. 6a).

## Experiments with epigenetic drugs

Five drugs targeting epigenetic factors have been selected on the basis of literature[56–60]. Drugs were dissolved in acetone at the proper concentration and applied three times on 8w pw1 tail skin. Six hours after the last treatment Lrig1 GL mice were wounded and collected 1 week after. Drug list: A-196, SUV420H1 and SUV420H2 inhibitor (Sigma-Aldrich, 3 mg ml$^{-1}$), UNC0638, EHMT1/2 inhibitor (Sigma-Aldrich, 3 mg ml$^{-1}$), SB747651A, MSK1 inhibitor (Axon Medchem, 1.5 mg ml$^{-1}$), EX-527, Sirtuins inhibitor (Sigma-Aldrich, 3 mg ml$^{-1}$) and PRT4165, Ring1a/1b inhibitor (PRC1) (Sigma-Aldrich, 3 mg ml$^{-1}$). To test the ability of PRT4165 to mimic memory, the same approach reported above was used to treat 0w mice.

## UVB irradiation and tumourigenesis

The UVB irradiation protocol was performed as previously described[61,62], with UVM-28EL (UVP Ultraviolet Product, Thermo Fisher) light source. For acute UV irradiation, tails were treated with 200 mJ cm$^{-2}$ three times, collected 2 days after the last irradiation and analysed. For in vivo tumourigenesis the DMBA-UVB two-stage-induced carcinogenesis protocol was used. Dorsal or tail skin was treated with 120 μg ml$^{-1}$ of DMBA (Sigma-Aldrich). UVB irradiation (180 mJ cm$^{-2}$) was started 10 days after and continued three times a week until the end point. As permitted by the ethics committee, tumours smaller than 1,200 mm$^3$ were collected. Tumours were classified as SCC in situ or SCC according to the tumour architecture[61,62]. For tumourigenesis, 8-week-old female SKH-1 mice were treated with 50 μl of PRT4165 (3 mg ml$^{-1}$) (Sigma-Aldrich) every other day three times and irradiated with 250 mJ cm$^{-2}$, three times a week until the end point[63].

## Human SCC

Cutaneous SCC samples with AK (UV-SCC) regions were obtained from nine patients, from both sexes. Participants' age ranged from 28 to 98 years. In addition, cutaneous SCC samples were collected from three patients with recessive dystrophic epidermolysis bullosa (EB-SCC) (five SCCs) and four patients with psoriasis (Pso-SCC). The institutional review board of the Hokkaido University Graduate School of Medicine approved the study (ID: 13-043, 14-063 and 15-029). The study was carried out according to the Declaration of Helsinki Principles, and the participants provided written informed consent. The patients with recessive dystrophic epidermolysis bullosa harboured compound heterozygous mutations in COL7A1 (NM_000094.4) (patient 1: c.5443G>A (p.Gly1815Arg) and c.5819del (p.Pro1940Argfs*65), patient 2: c.5932C>T (p.Arg1978*) and c.8029G>A (p.Gly2677Ser), patient 3: c.7723G>A (p.Gly2575Arg) and c.8569G>T (p.Glu2857*)), and the expression of type VII collagen was reduced in their skin specimens[64,65].

## Tumour analysis

For quantification of tail eSCCs each tail was subdivided in five main regions (0 to 8 mm from wound). The percentage of the scales that show eSCCs in each region were analysed (Fig. 8a). For quantification of effraction of basement membrane zone as indication of tumour invasiveness (Extended Data Fig. 10p), the length of deepest site of tumour from the normal adjacent basement membrane zone in H&E staining was evaluated by Fiji. To grade SCC in terms of differentiation, the ratio of well-structure-maintained epidermal layers and aberrant epidermal layers in H&E staining are calculated using Fiji.

## Mini-bulk RNA-seq and analysis

Mini-bulk RNA-seq of sorted epidermal cells have been performed on 150 cells. For in vivo mini-bulk RNA-seq, the tdTomato$^+$ cells were isolated and sorted from a zone that comprises both wound bed and distal memory HFs. In vitro mini-bulk RNA-seq cells was performed on sorted (and cultured) tdTomato$^+$ Sca-1$^+$ cells isolated from distal memory region or far control region. Briefly, after lysis, the biotinylated Oligo(dT) was bound to Dynabeads MyOne Streptavidin T1 beads (Thermo Fisher) and used to isolate messenger RNA. Reverse transcription, amplification and library preparation were performed as for the scRNA-seq doubling the volumes. After quality controls with FastQC v0.11.2 (https://www.bioinformatics.babraham.ac.uk/projects/fastqc), raw reads were trimmed with Trim Galore! v0.5.0 and aligned to the mouse reference genome (UCSC mm10/GRCm38) using HiSat2 v2.2.01 (ref. 66). Gene expression levels were quantified with featureCounts v1.6.12 (options: -t exon -g gene_name) using the GENCODE (https://www.gencodegenes.org) release M20 annotation[67]. Gene expression counts of each cell population were next analysed using the edgeR3 R/Bioconductor package[68]. To account for noise, the normalized counts were processed using the SVA4 R/Bioconductor package[69]. Following dispersion estimation, an analysis of variance (ANOVA)-like test was performed to identify the genes that were differentially expressed in any sample group across the time course, using 0w as the reference group in the model. Genes with |logFC| >1 in at least one timepoint and false discovery rate (FDR) <0.05 were considered

as differentially expressed. GO analysis was performed using EnrichR tool (v 3.0) (https://maayanlab.cloud/Enrichr)[70].

## scRNA-seq

For scRNA-seq at timepoints 0w, 1w pw1, 8w pw1 and 1w pw2, the tdTomato[+] cells were isolated and sorted from a zone that comprise both wound bed and distal memory HFs. For spatially resolved scRNA-seq at 8w pw1, the cells were instead isolated from wound bed (Wb), distal memory (Distal 3–7 mm from wound bed) or far control region (Ctrl >2 cm from injury) and sorted for tdTomato[+] Sca-1[+], to enrich for the upper-HF memory cells. Only female mice were used for this experiment. To minimize biological divergences, we performed two independent experiments, each of them based on four biological replicates. scRNA-seq was performed with a modified version of the Smart-Seq2 protocol[71] as in ref. [72]. The resulting complementary DNA was amplified with 25 cycles of PCR, and libraries will be prepared for sequencing with standard NexteraXT Illumina protocol (Illumina).

## scRNA-seq analysis

Reads were mapped to the *Mus musculus* transcriptome (ENSEMBL version 101) using Salmon v 0.13.1. Data processing: cells with fewer than 1,000 genes expressed or with over 50% of mitochondrial reads were filtered out, genes expressed in fewer than five cells were filtered out. Replicates were normalized, corrected for batch effects using Cluster Similarity Spectrum, Harmony or FastMNN and analysed using Seurat packages (v 4.0.1) on R (v 4.0.3) (ref. [73]). Dimensionality reduction (uniform manifold approximation and projection (UMAP)) were calculated with Seurat functions RunPCA (npcs = 100) and RunUMAP (reduction = 'css', dims = 1:8), using the top 10,000 variable genes. Cell clustering was performed with function FindNeighbors (reduction = 'css', dims = 1:8) and FindClusters (resolution = 2.1). Cluster marker genes were identified by FindAllMarkers (only.pos = T, return. thresh = 0.05) function.

The analysis of differentially expressed genes (DEGs) within the same cluster was performed using Seurat's FindMarkers() function.

Single-cell data from Lgr5 GL cells, Lrig1 GL cells aged cells and vehicle- or PRT4165-treated Lrig1 GL were projected onto the reference UMAP structure derived from Lrig1 GL young data, using SeuratIntegration Tools[74] with FindTransferAnchors and MapQuery functions.

Pseudotime analysis was carried out using Slingshot package (v. 1.8.0) (ref. [75]) on the Seurat object, using the top 10,000 variable genes. To derive the DEGs, the TestPseudotime function was used with a threshold of FDR <0.05. The genes were *z*-scored (and smoothed) and uploaded on Morpheus (https://software.broadinstitute.org/morpheus) to obtain the heat maps. GO enrichment analysis was performed using EnrichR tool (v 3.0) (ref. [70]) using GO Biological Process.

GSEA analysis were performed using the GSEA function by ClusterProfiler package (v 3.18.1) (ref. [76]).

## ATAC–seq

Epidermal cells were isolated as described above. For 0w, 1w pw1, 8w pw1 and 1w pw2, cells were isolated specifically from distal memory region (Distal), 3–7 mm from wound bed. For in vitro ATAC–seq, cells from the far away control region (Ctrl more than 2 cm away from wound bed) and from the distal memory zone (Distal 3–7 mm from wound bed) were collected and cultured. ATAC–seq relative to pre-tumour lesions was performed on cells from Wd&TS tail collecting the distal memory zone or an unwounded TS-treated epidermis (Ctrl). To enrich upper-HF cells, Lrig1-GL tdTomato[+] Sca-1[+] cells were sorted as described above. ATAC–seq was performed as previously described[77]. For each of the two independent experiments, epidermis was pooled from two mice. Briefly, 20,000–50,000 sorted cells were lysed in ATAC lysis buffer for 5 min and then transposed with TN5 transposase for 30 min at 37 °C. NexteraXT indexes (Illumina) were added with 12 cycles of PCR and samples were sequenced paired-end on a NextSeq 1000 System (Illumina).

## ATAC–seq analysis

Sequenced reads were processed with the ENCODE ATAC–seq pipeline (v1.9.0, https://github.com/ENCODE-DCC/ATAC-seq-pipeline) using the default parameters. Bowtie2 (ref. [78]) was used to align reads to the mouse reference genome UCSC mm10. After the discard of duplicated, multi-mapping and poor-quality alignments, the peak calling was performed with MAC2 (ref. [79]) generating the signal tracks as fold enrichment control. Differentially opened regions between treatment and its control samples were identified using DiffBind v3.0.3 (ref. [80]) with the following parameters: normalise = DBA_NORM_NATIVE, library = DBA_LIBSIZE_DEFAULT, background = BKGR_TRUE, AnalysisMethod = EDGER. Peaks with *p*-value <0.05 and |logFC| >0.5 were considered differentially open. Target gene annotation of each ATAC peak was obtained with GREAT[81] using the Basal plus extension association rules and the whole mouse genome as background. The enrichments of selected gene lists were performed by using Fisher's exact test in the BioConductor R package GeneOverlap (http://shenlab-sinai.github.io/shenlab-sinai/).

## Statistics and reproducibility

Statistical analysis was performed by using the GraphPad Prism 7 software (GraphPad). Data are presented as mean ± standard error of the mean (s.e.m.), mean ± standard deviation (s.d.) or median with 25th and 75th percentiles. Statistical significance was determined with the two-tailed unpaired Student's *t*-test with a 95% confidence interval under the untested assumption of normality, with Mann–Whitney test or random permutation. No statistical method was applied to pre-determine sample size, and mice were assigned at random to groups. Data distribution was assumed to be normal, but this was not formally tested. No randomization was done. Mice were categorized on the basis of genotype. No blinding was done since the same researchers performed both data acquisition and analysis. Images are representative of at least three independent experiments, and mice from both sexes were used, if not differently indicated.

## Material availability

This study did not generate new unique reagents. Material that can be shared will be released via a Material Transfer Agreement. Further information and requests for resources and reagents should be directed to and will be fulfilled by the lead contact: Giacomo Donati at giacomo. donati@unito.it.

## Reporting summary

Further information on research design is available in the Nature Portfolio Reporting Summary linked to this article.

## Data availability

Sequencing data that support the findings of this study have been deposited in the Gene Expression Omnibus (GEO) under accession code GSE197590. Source data are provided with this paper. All other data supporting the findings of this study are available from the corresponding author on reasonable request.

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

## Acknowledgements

We thank P. Porporato, C. Sebastian, K. Mulder and S. Woodhouse for their helpful suggestions. We thank all the facilities at the Molecular Biotechnology Center for their technical support and all the MBC members for making the Institute a friendly and scientifically stimulating environment. The G.D., T.H. and K.N. laboratories are supported by the Chan Zuckerberg Initiative (Single-Cell Analysis of Inflammation, Id. DAF2020-217532). The G.D. laboratory is also supported by AIRC, Associazione Italiana per la Ricerca sul Cancro (MFAG 2018 - Id. 21640) and Compagnia di San Paolo (Excellent Young PI, Id. CSTO167890). The T.H. laboratory is also supported by the Israel Science Foundation (ISF, Id. 435/20). M.W. is a recipient of JSPS overseas research fellowship. The S.O. laboratory is supported by AIRC (IG 2017 - Id. 20240).

## Author contributions

G.D. designed and supervised the study. C.L.L. and M.W. performed all the experiments with C.D., D.D., L.E., L.C. and D.B.'s assistance. Sequencing samples were prepared by V.P., sequenced by F.A. and analysed by G.P., A.L., S.K. and A.T. T.N. and K.N. provided human samples. G.D., C.L.L., M.W., V.P., T.H., K.N. and S.O. interpreted the data. G.D. and C.L.L. wrote the manuscript with input from all authors.

## Competing interests

The authors declare no competing interests.

## Additional information

**Extended data** is available for this paper at

**Correspondence and requests for materials** should be addressed to Giacomo Donati.

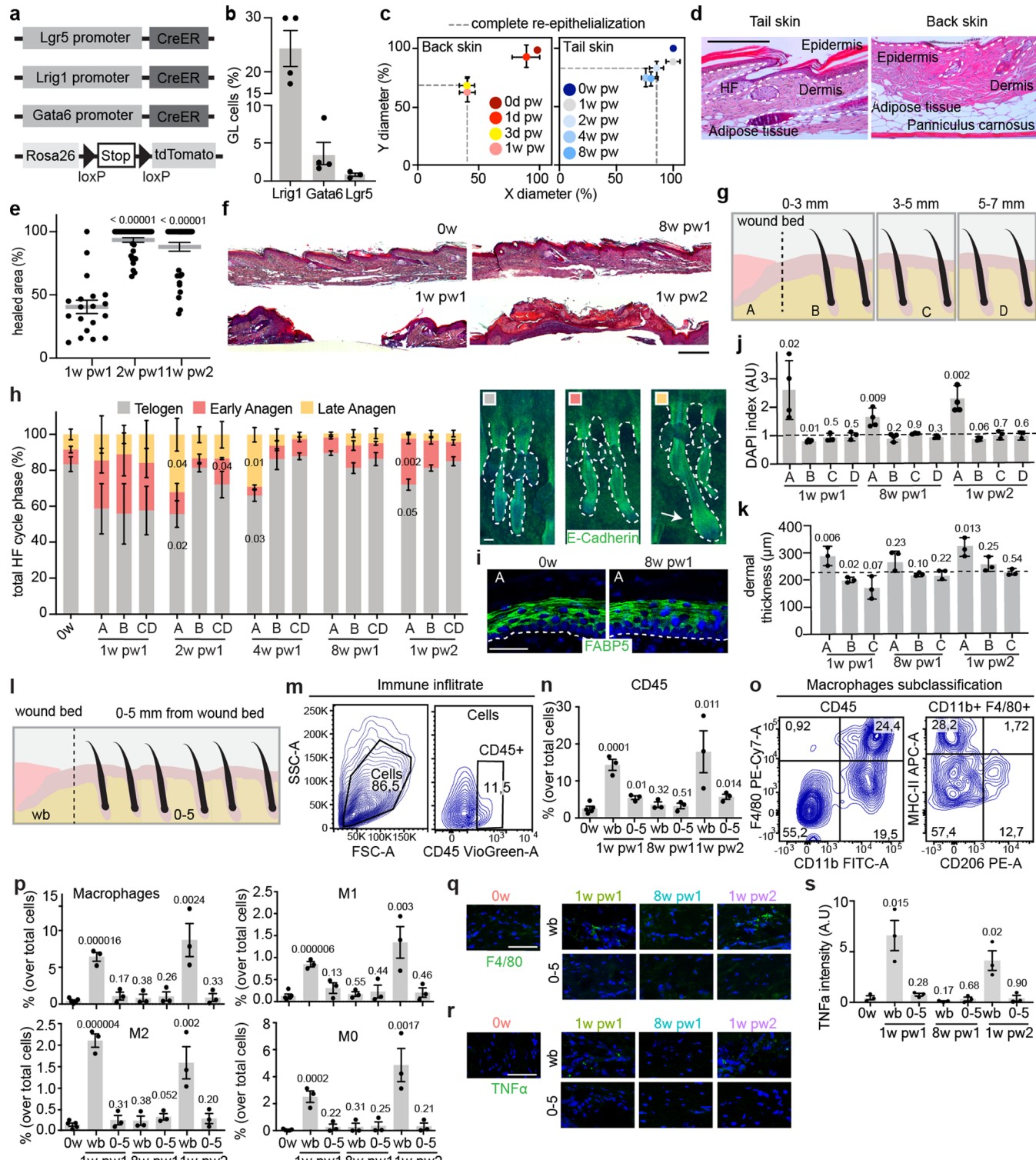

**Extended Data Fig. 1 | See next page for caption.**

**Extended Data Fig. 1 | Characterisation of the two consecutive injury model.**
**a**, Murine transgenic models. **b**, Flow cytometry quantification of GL cells in
the epidermis at 0w, plotted as mean ± SEM. n = 3 (Lgr5), n = 4 (Lrig1, Gata6).
**c**, **d**, Tail skin model reduces the issue of tissue contraction. Comparison of wound
margins contraction during healing between back and tail. n = 3 mice (**c**). H&E
staining of tail and back skin. Dashed line indicates epidermis (**d**). **e**, Percentage
of re-epithelialized area, plotted as mean ± SEM. n = 19 (1w pw1), n = 37 (1w
pw2, 2w pw1). **f**, H&E staining of skin at indicated time points. **g**, Scheme of the
sub-regions defined in injured skin. **h**, Percentage of hair follicle cycle phases
(telogen, early anagen or late anagen (white arrow)), plotted as mean ± SEM
and representative whole-mounts (right). n = 3 (2w pw1, 4w pw1), n = 4 (0w),
n = 5 (1w pw1, 8w pw1, 1w pw2). **i**, FABP5 differentiation marker in skin sections.
**j**, **k**, Dermal cell density as DAPI index (number of nuclei per mm$^2$) (**j**) and dermal
thickness (μm) as ratio to 0w (**k**). Dashed line represents 0w mean. n = 34.

**l**, The immune infiltrate is analysed in wound bed (wb) and between 0 and 5 mm
from wound bed (0–5). **m**, Gating strategy for CD45$^+$ cells. **n**, Flow cytometry
quantification (%) of CD45$^+$ cells. n = 3 (1w pw1, 8w pw1, 1w pw2), n = 5 (0w). **o–s**,
TNFα-expressing macrophages are key mediators of HFs communication after
damage[23]. They are recruited specifically in the wound bed at the two acute
phases 1w pw1 and 1w pw2, while absent at 8w pw1. Importantly they are not
recruited in distal memory areas at any tested time points. **o**, Gating strategy
for macrophages (F4/80$^+$CD11b$^+$), divided in M1 (F4/80$^+$CD11b$^+$MHC-II$^+$), M2
(F4/80$^+$CD11b$^+$CD206$^+$) and M0 (F4/80$^+$CD11b$^+$MHC-II$^-$CD206$^-$) infiltrate among
CD45$^+$ cells. **p**, Flow cytometry quantification (%) of macrophages, M1, M2 and
M0. n = 3 (1w pw1, 8w pw1, 1w pw2), n = 5 (0w). **q–s**, Skin sections stained with
F4/80 (green) (**q**) and TNFα (green) (**r**) and relative quantification (**s**). n = 3.
*P*-value (P) from a two-tailed *t*-test. Data are mean ± SD, if not differently
indicated. Scale bars: 300 μm (**d**, **f**); 50 μm (**h**, **i**, **q**, **r**).

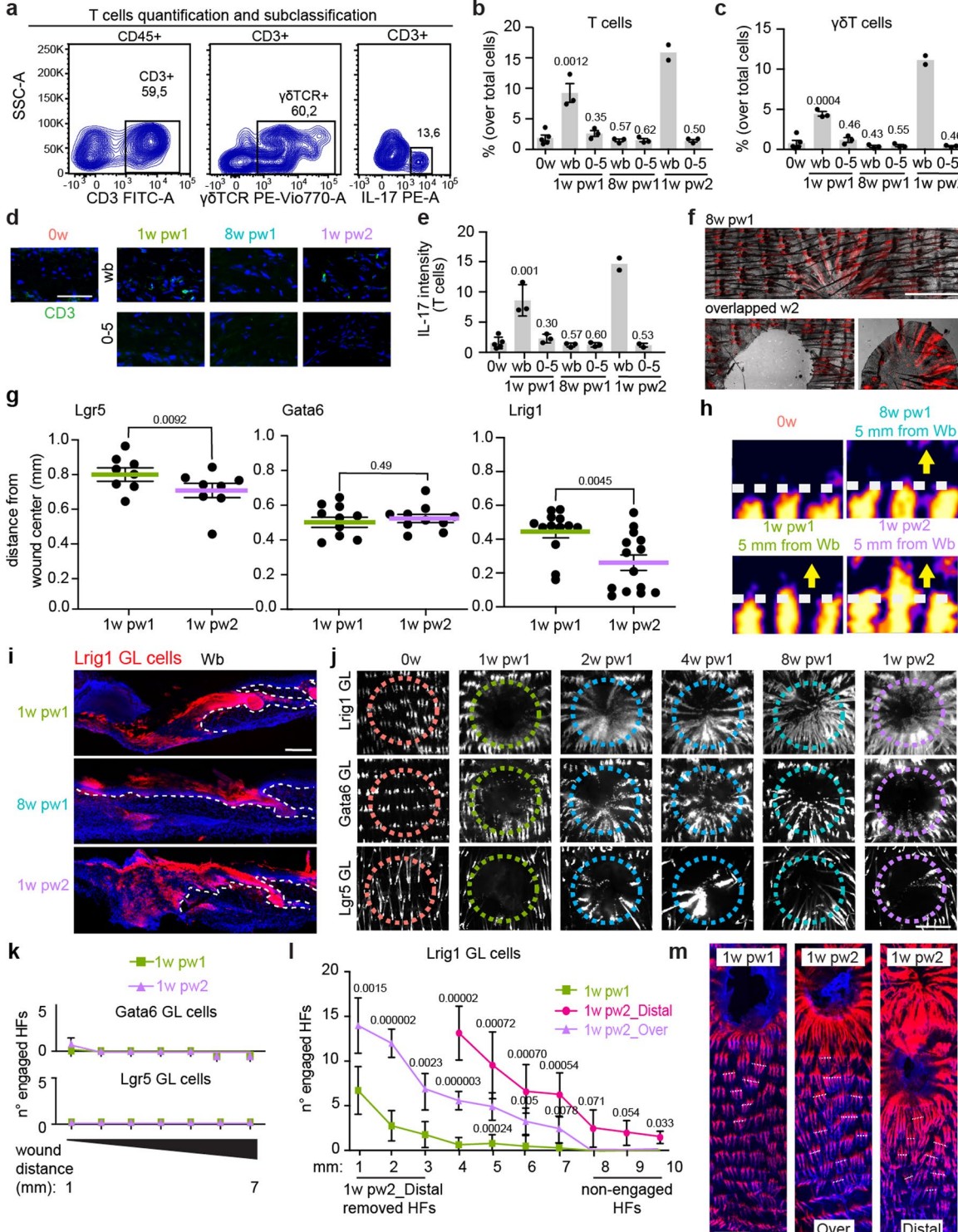

**Extended Data Fig. 2 | Distal memory is specific to the Lrig1 GL lineage. a-e,** IL-17-expressing γδ T cells, responsible for epidermal mTOR and HIF1α induction during wound healing[48], are recruited in wound, but not in distal memory areas, at the two acute healing phases 1w pw1 and 1w pw2, while absent at 8w pw1. **a,** Gating strategy for T cells (CD3⁺) and γδ T cells (CD3⁺ γδTCR⁺) among CD45⁺ cells and IL-17. **b, c,** Flow cytometry quantification (%) of T cells (**b**) and γδ T cells (**c**) following the scheme in Extended Data Fig. 1l. n = 2 (1w pw2), n = 3 (1w pw1, 8w pw1), n = 5 (0w) mice. **d,** Staining of CD3 (green) in skin sections. **e,** IL-17 intensity in T cells quantified by flow cytometry. n = 2 (1w pw2), n = 3 (1w pw1, 8w pw1), n = 5 (0w) mice. **f,** Epidermal whole-mount highlighting the removal of Lrig1 progeny-derived IFESCs (red), before and after the second overlapped wound. **g,** Quantification of first (1w pw1) and second (1w pw2) wound closure as distance from wound centre, relative to Fig. 1c. n = 8 (Lgr5), n = 10 (Gata6), n = 14 (Lrig1).

Data are mean ± SEM. **h,** 3D surface plot of Lrig1 GL cells at 0w and 1w pw1, 8w pw1 and 1w pw2, 5 mm from wound bed. Dashed line represents the IFE-HF boundary. Yellow arrows indicate wound bed (Wb). **i,** Representative confocal images of Lrig1 GL cells (red) exiting toward IFE wound bed at 1w pw1, 8w pw1 and 1w pw2. **j,** Epidermal whole-mounts (red channel) of GL tdTomato⁺ cells showing their contribution on the wound bed. Dashed circles highlight the original wound perimeter. **k,** Number of engaged HFs in Gata6 and Lgr5 GL cells. n = 10. **l, m,** HFs' engagement is shown for 1w pw1 and 1w pw2, when the second wound is performed overlapped (1w pw2_Over) or distally (1w pw2_Distal) to the first one. n = 3 (1w pw2_Distal), n = 6 (1w pw1, 1w pw2_Over) (**l**). Confocal images of engaged HFs showing Lrig1 GL cells (red). Dashed white lines highlight representative engaged HF triplets (**m**). P-value (P) from a two-tailed t-test. Data are mean ± SD, if not differently indicated. Scale bars: 100 μm (**d, h, i**); 1 mm (**f, j, m**).

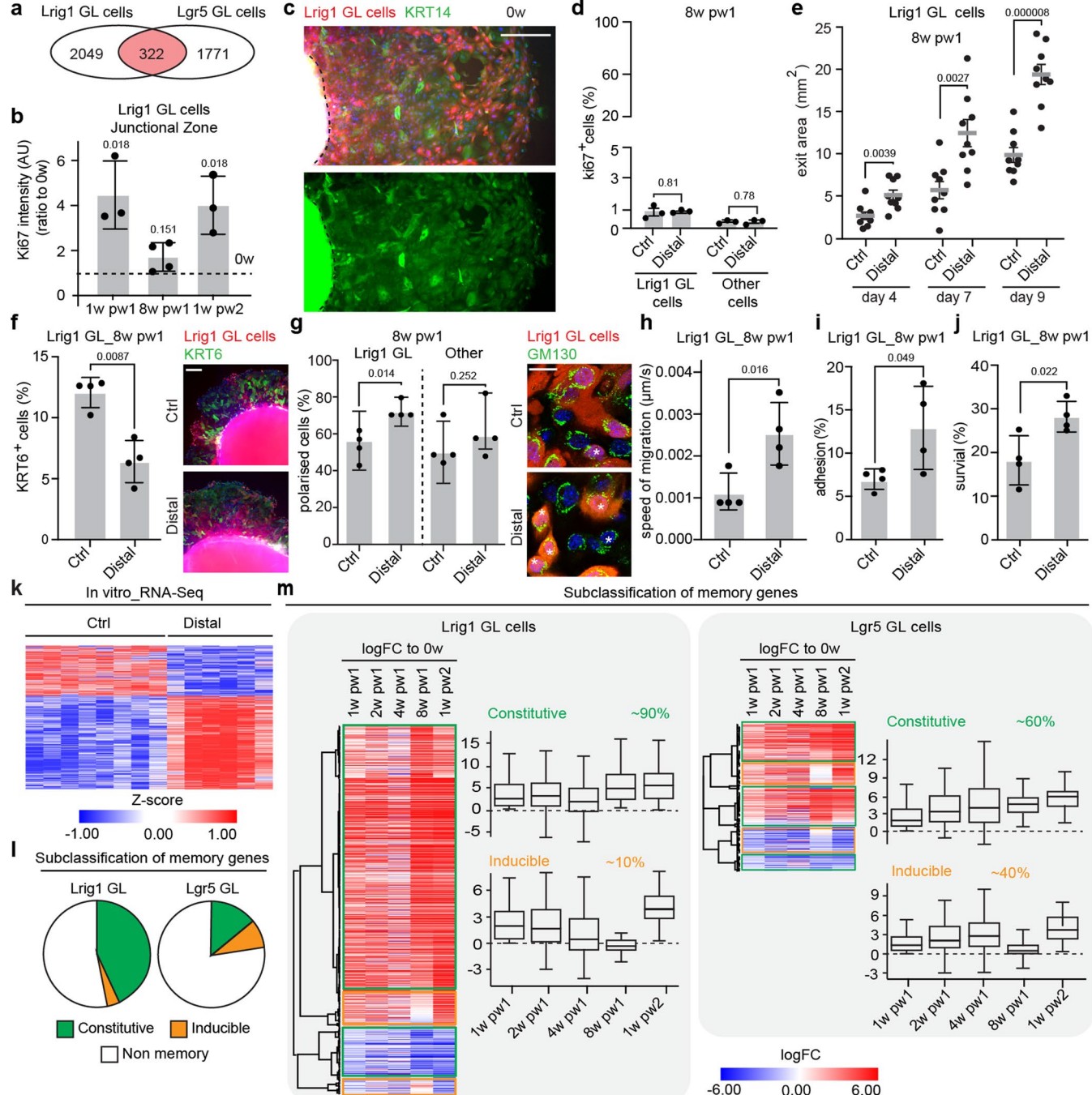

**Extended Data Fig. 3 | Ex vivo and in vitro evaluation of Lrig1 progeny fitness after wound experience. a**, Venn diagram of differentially expressed genes across the time course between Lrig1 and Lgr5 GL cells. **b**, Quantification of Ki67, as ratio to 0w. Dashed line represents 0w mean. n = 3 (0w, 1w pw1, 1w pw2), n = 4 (8w pw1). **c-j**, *Ex vivo* and *in vitro* experimental settings at 8w pw1, as in Fig. 2c: Lrig1 GL skin biopsies from wound-educated (Distal- 2 to 7 mm from wound bed) and control areas (Ctrl- > 2 cm from wound bed). **c**, *Ex vivo* assay assesses migratory ability for epidermal Krt14⁺ (green) cells. **d**, Percentage of Ki67⁺ nuclei. Epidermal cells exiting skin explants do not proliferate. Data are mean ± SEM. n = 3. **e**, Exit area (mm²) of Lrig1 GL cells from the explant, related to Fig. 2d. Data are mean ± SEM. n = 9. **f**, In cultured cells Krt6 positivity is inversely proportional to cell migration[82]. Percentage of Krt6⁺ in Lrig1 GL cells (left) and representative pictures (right). n = 4. **g**, Cis-Golgi (Gm130) distribution in Distal vs Ctrl cells.

Percentage of polarised cells (marked by asterisk) in GL tdTomato⁺ and tdTomato⁻ cells (left) and representative pictures (right). n = 8 (Ctrl), n = 10 (Distal). **h**, Speed of migration (µm/s) of Lrig1 GL cells from *in vitro* time lapse migration assay, relative to Fig. 2e, f. n = 4. **i, j**, Percentage of adhesion (**i**) and survival (**j**) of Lrig1 GL cells isolated from Distal or Ctrl HFs. n = 4 mice. **k**, Bulk RNA-seq of Lrig1 GL cells from Distal or Ctrl area after 7 days of culture. Heatmap of deregulated genes is shown. n = 8 (Ctrl), n = 6 (Distal). **l**, Pie charts illustrating the proportions of the memory genes subsets from RNA-seq of Lrig1 and Lgr5 GL cells in vivo, related to Fig. 2a. **m**, LogFC heatmap shows memory genes at each time point with respect to 0w. Constitutive (green) and inducible (yellow) memory genes, related to Fig. 2a. n = 3 mice. Data are median with 25th and 75th percentiles. *P*-value (P) from a two-tailed *t*-test. Data are mean ± SD, if not differently indicated. Scale bars: 50 µm (**b, e**); 20 µm (**g**).

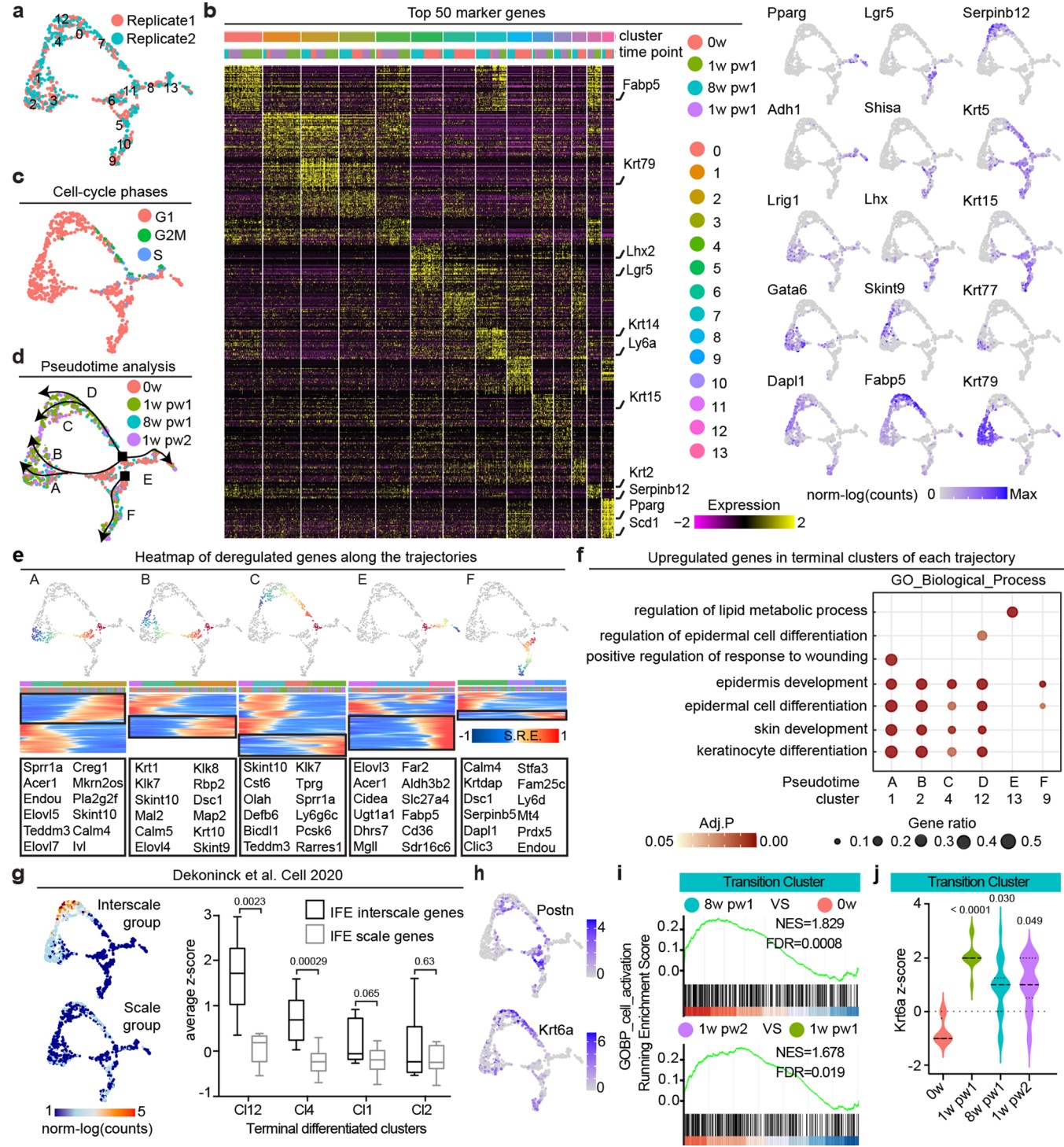

**Extended Data Fig. 4 | Single cell analysis assigned known epidermal niches to cell clusters. a**, UMAP of scRNA-seq replicates, relative to Fig. 3a–h and Fig. 4. **b**, Heatmap representing the 50 top marker genes for each cluster (left) and UMAP of cells coloured by the expression of selected genes (right). Top bars indicate the clusters and the time points. **c**, UMAP of cells coloured by assigned cell-cycle phase. **d**, Trajectories identified by pseudotime analysis. The starting points (black square) are Cluster 11 (JZ) for the upper HF and Cluster 5 (bulge) for the lower HF. **e**, Upper panel: plot of pseudotime trajectories and the associated heatmap with Smoothed Relative Expression (SRE). Lower panel: selected upregulated genes in the last cells of each trajectory are listed. Trajectory D is analysed alone in Fig. 4a, b. **f**, GO enriched for induced genes in the terminal clusters of each trajectory. Gene ratio (number of genes in the pseudotime/ number of genes in the GO term) and adjusted p-value (adjP) are plotted. **g**, UMAP of cells coloured by interscale and scale gene signature (left) and whisker plot of average z-score for scale and interscale gene groups in the epidermal terminal differentiation clusters (1, 2, 4, 12) of each trajectory (right). Gene signatures from Dekoninck et al. study[27]. n = 8 (IFE interscale), n = 9 (IFE scale) genes. **h**, Postn and Krt6a expressions are plotted on the UMAP. Scale in log(counts). **i**, GSEA ranking of the indicated gene signature using the differential gene expression estimated from the comparison 8w pw1 vs is 0w (upper panel) and 1w pw2 vs 1w pw1 (lower panel) in Transition Cluster. NES, normalized expression score. **j**, Violin plot of Krt6a expression in cells from Transition Cluster in each time point. Data are median with 25th and 75th percentiles. P-value (P) from a two-tailed t-test. scRNA-seq data (**a-j**) are the integration of two independent experiments, each of them based on 4 biological replicates.

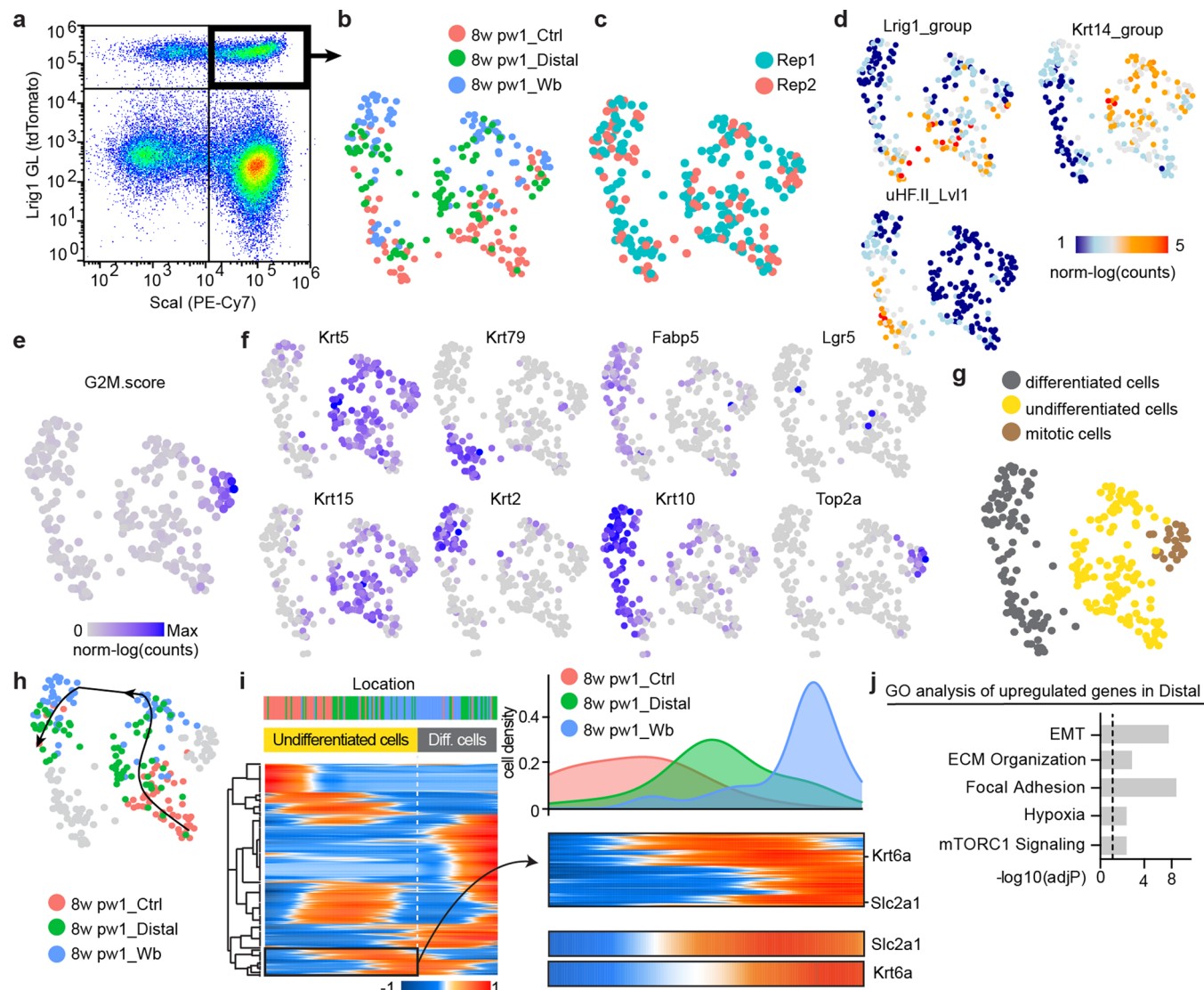

**Extended Data Fig. 5 | Spatially resolved single cell RNA-seq validates distal priming. a-j,** A scRNA-seq was performed on upper HF Lrig1 GL cells at 8w pw1 collected from distinct areas: close proximity of the wound bed (Wb); distal memory region (Distal- 3 to 7 mm from wound bed) control non-memory region (Ctrl- > 2 cm from wound bed). Data are the integration of two independent experiments, each of them based on 4 biological replicates. **a**, Gating strategy for tdTomato-Sca1 double positive cell sorting. Negative controls are used to set the gate. **b**, UMAP of the spatially resolved scRNA-seq with cells coloured by location (Wb, Distal or Ctrl). **c**, UMAP of cells coloured by replicate. **d**, Plot of gene group expression from Joost et al. study[26]. Normalised log(counts) is plotted. **e**, UMAP of cells coloured by G2-M cell cycle phase score. **f**, UMAP of cells coloured by the expression of selected genes. Normalised log(counts) is plotted. **g**, Differentiation state of Lrig1 GL cells. **h, i**, Pseudotime analysis from

undifferentiated upper HF cells to differentiated upper HF cells is performed, as in the case of the Trajectory D in Fig. 4a. **h**, Trajectory identified by pseudotime analysis. **i**, Smoothed Relative Expression (SRE) of deregulated genes along the pseudotime trajectory. Cell location respect to wound bed (as Ctrl in red, Distal in green and Wb in blue) is indicated above. The analysis of undifferentiated cells highlights the existence of a gene set expressed in Distal and Wb cells but not in Ctrl (black rectangle magnified in right panel). Importantly, Glut1 (Slc2a1) and Krt6 are included in the gene set (right). **j**, Enriched GO terms of the gene set expressed in Distal and Wb cells but not in Ctrl, identified in (**i**) are plotted as -log10 of the adjusted *p*-value (AdjP). Most of GO terms shown in Fig. 4b, were identified here, confirming that in Distal region, upper HF cells are wound primed. The dashed line underlines significance.

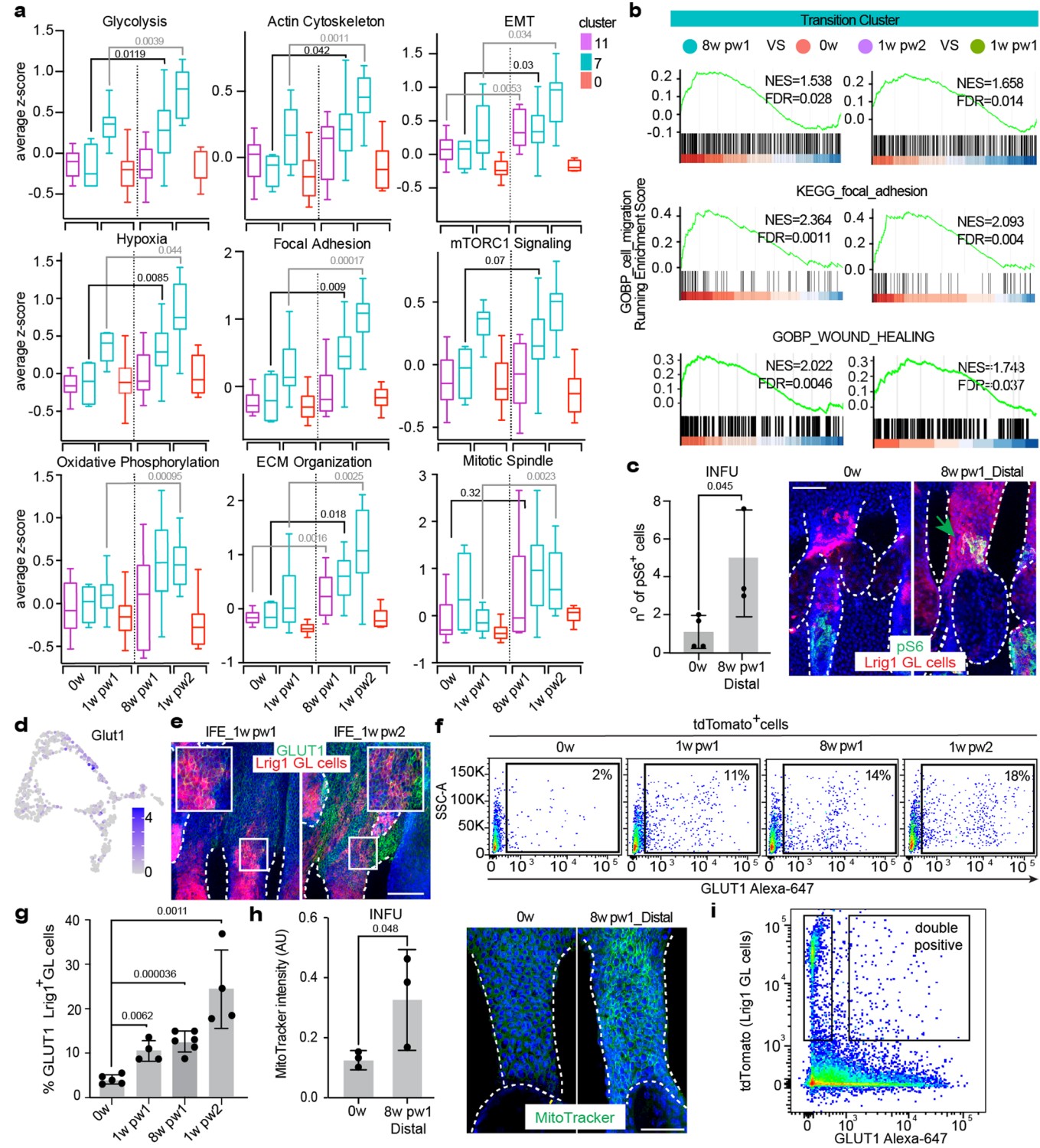

**Extended Data Fig. 6 | Characterisation of primed-memory cell state. a, b**, Analysis of scRNA-seq data, relative to Fig. 3a–h and Fig. 4. **a**, Cells in pseudotime trajectory D are divided by time points and clusters and the average z-score of the indicated GO term is plotted in the whisker plot. Data are median with 25th and 75th percentiles. n = 50 (0w_cl11), n = 4 (0w_cl7), n = 13 (1w pw1_cl7), n = 50 (1w pw1_cl11), n = 9 (8w pw1_cl11), n = 26 (8w pw1_cl7), n = 13 (1w pw2_cl7), n = 8 (1w pw2_cl11) cells. **b**, GSEA ranking of the indicated gene signature using the differential gene expression estimated from the comparison 8w pw1 vs 0w (left panel) and 1w pw2 vs 1w pw1 (right panel) in Transition Cluster (cluster 7). NES, normalized expression score. **c**, Number of pS6 (Ser235-236) positive cells among Lrig1 GL cells in infundibulum (INFU) (left) and HF images (right) at 0w and 8w pw1_Distal,

5 mm from wound site (right). n = 3 (8w pw1_Distal), n = 4 (0w) wounds. **d**, Glut1 expression levels are plotted on the UMAP. Scale in log(counts). **e**, Whole-mount staining of GLUT1 in 1w pw1 and 1w pw2. **f, g**, Gating strategy of Glut1+ tdTomato+ cells (**f**) and quantification (**g**) at the indicated time points. n = 4 (1w pw1, 1w pw2), n = 5 (0w), n = 6 (8w pw1) wounds. **h**, Quantification of MitoTracker index in INFU (left) and HF images (right) at 0w and 8w pw1_Distal, 5 mm from wound site. n = 3 mice. **i**, Representative gating to sort tdTomato-Glut1 double positive cells, relative to Fig. 4g. P-value (P) from a two-tailed t-test. Data are mean ± SD, if not differently indicated. Scale bars: 50 μm (**c, e, h**). scRNA-seq data (**a, b**) are the integration of two independent experiments, each of them based on 4 biological replicates.

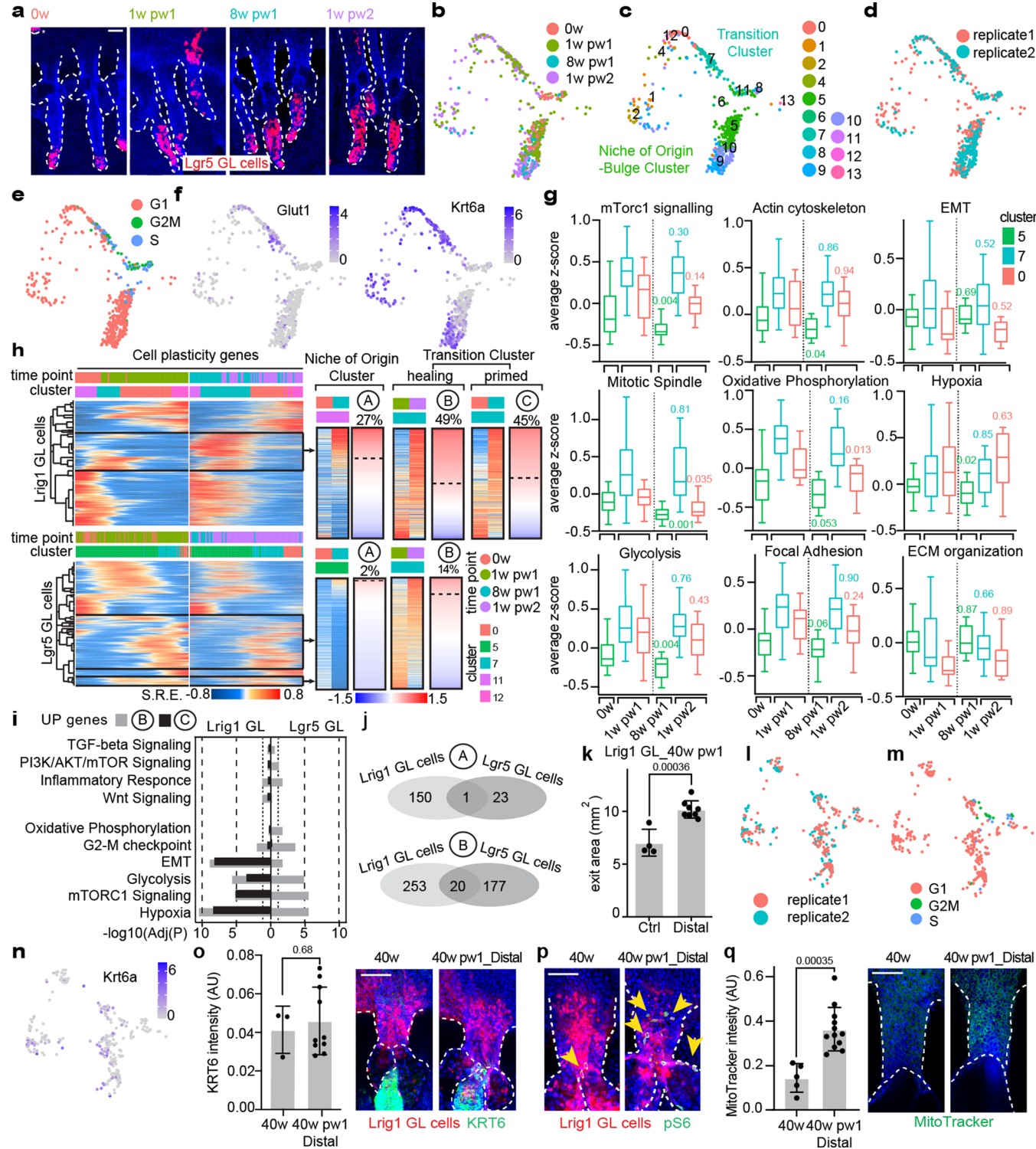

**Extended Data Fig. 7 | See next page for caption.**

**Extended Data Fig. 7 | Primed cell-state is peculiar of infundibular Lrig1 GL progeny and maintained in ageing. a**, Whole-mounts of Lgr5 GL HF triplets at the indicated time point. **b-f**, Single cell data from Lgr5 GL cells. UMAP of cells coloured by time points (**b**), clusters (**c**), replicates (**d**) or cell-cycle (**e**). **f**, Glut1 and Krt6a expressions plotted on UMAP. Scale in log(counts). **g**, Whisker plots of the average expression of each GO term enriched in Fig. 4b. Data are median with 25th and 75th percentiles. n = 46 (0w_cl5), n = 39 (1w pw1_cl7), n = 17 (1w pw1_cl0), n = 16 (8w pw1_cl5), n = 26 (1w pw2_cl7), n = 16 (1w pw2_cl0) cells. **h**, Left panel: Smoothed Relative Expression (SRE) of deregulated genes in the indicated cluster for Lrig1 GL (up) and Lgr5 GL (down) cells. Cell plasticity genes in the black rectangle. Right panel: percentage of cell plasticity genes are shown for 'A', 'B' or 'C' cells. Wound-primed cells at 8w pw1 in Transition Cluster ('C') only exists in Lrig1 GL cells. **i**, GO analysis for the deregulated genes in cell subsets 'B' and 'C', as -log10 of adjusted p-value (-log10(AdjP)). Dotted lines represent significance.

**j**, Venn diagram of the genes in 'A' and 'B' for Lgr5 and Lrig1 GL populations. **k**, *Ex vivo* migration assay. Exit area (mm²) of Lrig1 GL cells from explants collected at 40w pw1 from memory (Distal- 2 to 7 mm from wound bed) and control areas (Ctrl- > 2 cm from wound bed), related to Fig. 6e. n = 4 (Ctrl), n = 8 (Distal). **l, m**, scRNA-seq of Lrig1 GL cells at the two aged homeostasis: 40w and 40w pw1. UMAP of cells coloured by replicate (**l**) or by cell cycle phase (**m**). **n**, Krt6a expression levels. Scale in log(counts). **o**, Quantification of Krt6 (left) in the infundibulum from epidermal whole-mount (right). n = 3 (40w), n = 10 (40w pw1_Distal). **p**, Epidermal whole-mount of pS6 staining (green) at 40w or 40w pw1_Distal, related to Fig. 6k. **q**, MitoTracker staining in whole-mount infundibulum and quantification. n = 5 (40w), n = 12 (40w pw1_Distal) wounds. P-value (P) from a two-tailed t-test. Data are mean ± SD, if not differently indicated. Scale bars: 50 μm (**a, o, p, q**). scRNA-seq data (**b-j**) are the integration two independent experiments, each of them based on 4 biological replicates.

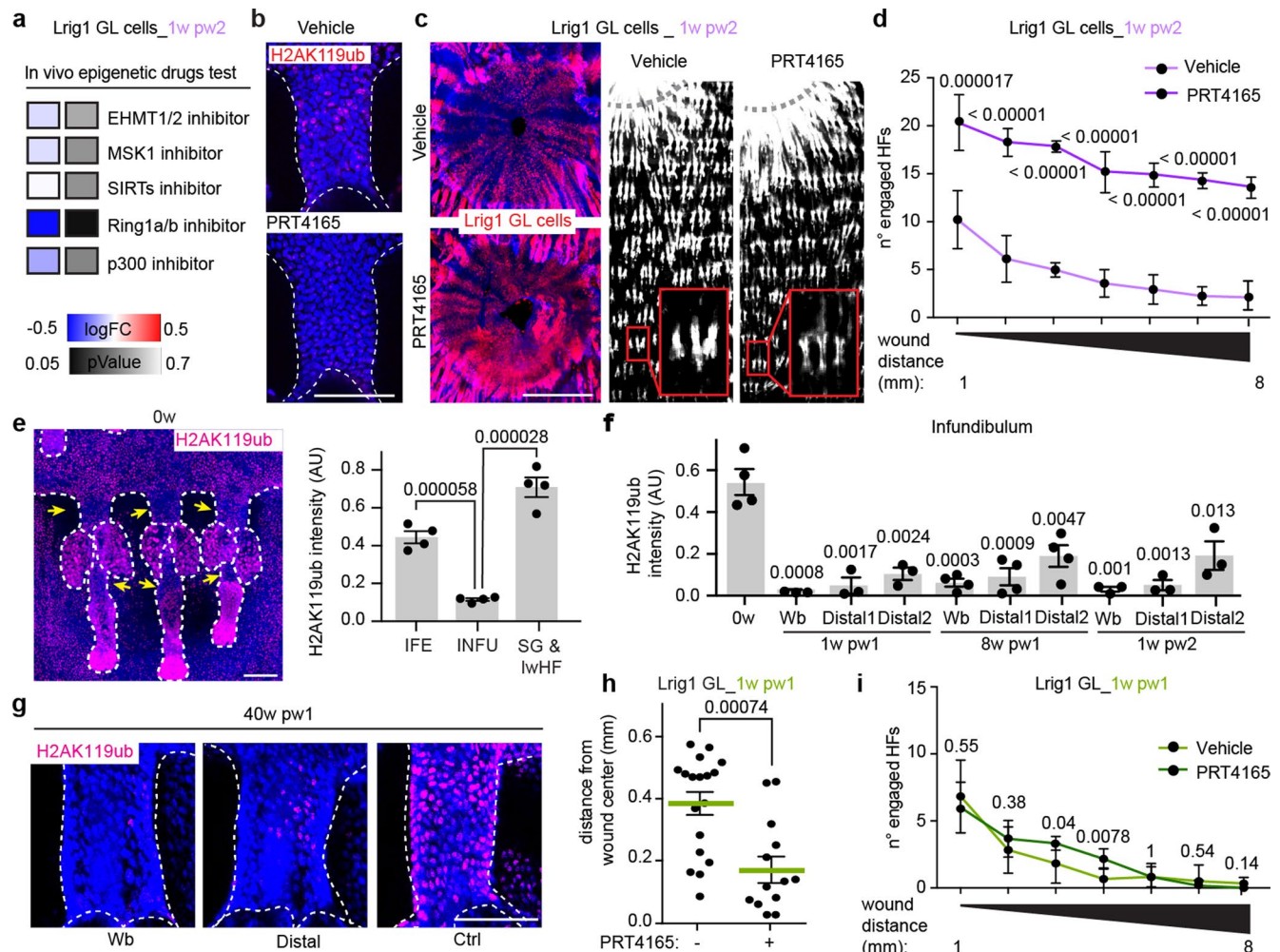

**Extended Data Fig. 8 | Physiological reduction of H2AK119Ub after wound healing in distal HFs. a**, Effect of 5 epigenetic drugs on wound closure at 1w pw2. 8w pw1 mice are treated 3 times with the reported inhibitors before wound induction. Wound closure (as the distance of cells from wound centre at 1w pw2) is quantified and plotted as logFC to vehicle-treated mice. n = 4 mice. **b**, Epidermal whole-mounts showing that PRT4165 reduces H2AK119ub in upper HF. **c**, Representative HF images of the effect of PRT4165 on wound closure (left) and distal HFs engagement in Lrig1 GL tdTomato⁺ mice at 1w pw2 (right). **d**, Number of engaged HFs at 1w pw2 after vehicle or PRT4165 treatment. n = 7 (PRT4165), n = 8 (vehicle) wounds. **e**, Homeostatic levels of H2AK119ub in whole-mount HFs and relative quantification in interfollicular epidermis (IFE), infundibulum (INFU), sebaceous gland (SG) and lower HF (lwHF). n = 4 mice. **f**, Quantification of

H2AK119ub in infundibulum (INFU), relative to Fig. 7a. Wound bed (Wb) and two different distal memory regions (Distal1- 3 to 5 mm and Distal2- 5 to 7 mm apart from wound bed) are compared to 0w. n = 3 wounds. **g**, Epidermal whole-mount H2AK119ub levels at 40 weeks post wound (40w pw1) homeostasis in wound bed (Wb), distal memory region (Distal- 3 to 7 mm from wound bed) and control area (Ctrl- >2 cm from wound bed). **h**, Quantification of distance from the wound centre in vehicle- vs PRT4165- treated skin at 1w pw1, relative to Fig. 7c. Data are mean ± SEM. n = 13 (vehicle), n = 18 (PRT4165) wounds. **i**, Number of engaged HFs at 1w pw1 after vehicle/PRT4165 treatment. n = 6 wounds. *P*-value (P) from a two-tailed *t*-test. Data are mean ± SD, if not differently indicated. Scale bars: 100 µm (**b**, **e**, **g**); 1 mm (**c**).

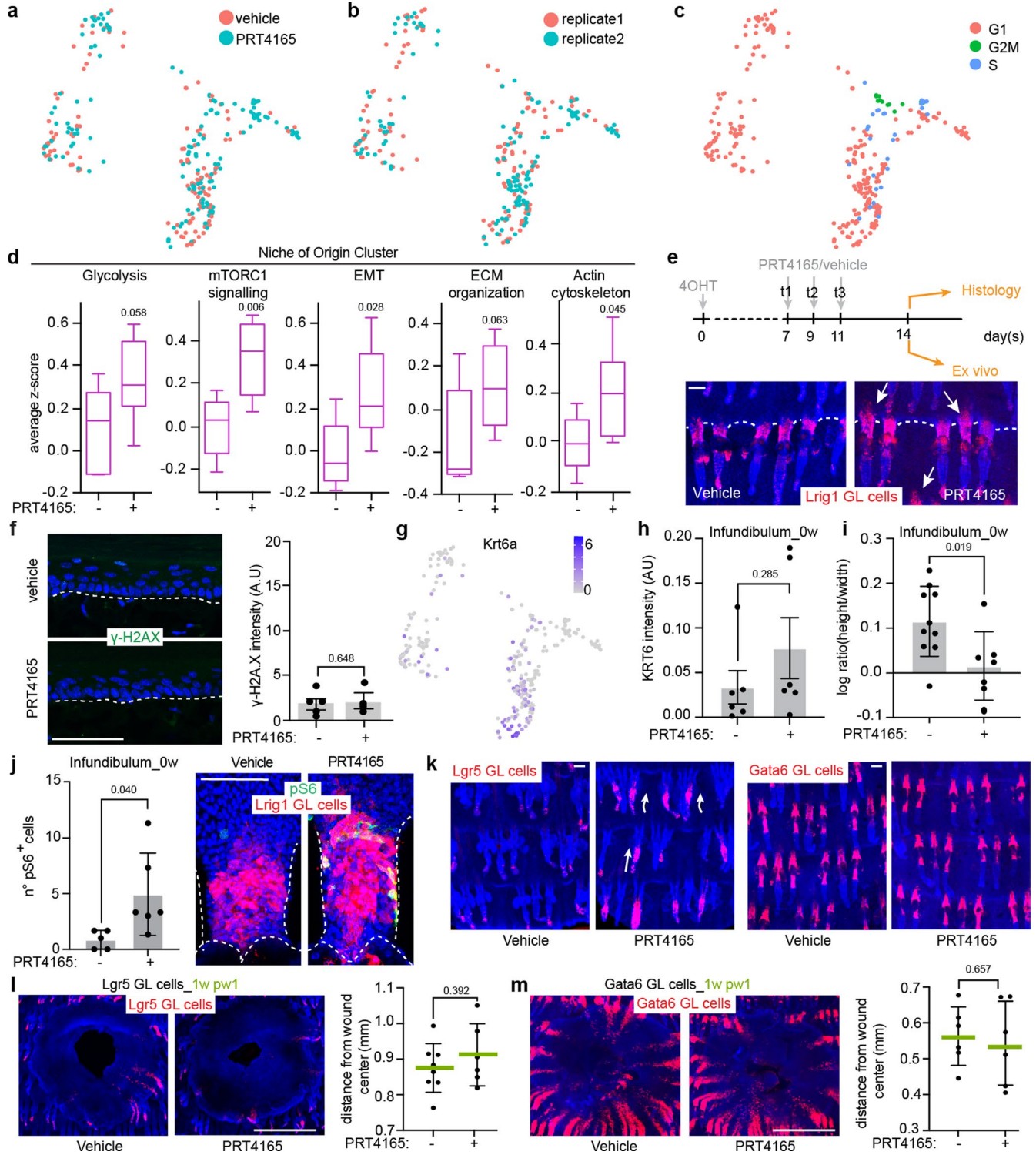

**Extended Data Fig. 9 | PRT4165 treatment mimics wound priming specifically in Lrig1 epidermal lineage. a–c**, scRNA-seq data from Lrig1 GL skin treated with vehicle/PRT4165. UMAP of cells coloured by treatment (**a**), replicate (**b**) or cell-cycle phases (**c**). **d**, Whisker plots of the average expression for each GO term enriched in Lrig1 GL single cells. Data are median with 25th and 75th percentiles. n = 5 (vehicle), n = 8 (PRT4165) cells. **e**, Upper panel: Setting for PRT4165 treatment in Lrig1 GL mice: *day 0*, skin is genetically labelled; *day 7-9-11*, PRT4165/vehicle treatment every other day for 3 times; *day 14*, tissue collection for histology or ex vivo assay. Lower panel: confocal pictures at day 14 of Lrig1 GL tdTomato+ cells, showing Lrig1 GL HF exit upon PRT4165 topical application. **f**, Representative pictures (left) and quantification (right) of γ-H2A.X (a DNA damage marker[42]) in vehicle versus PRT4165 acute treatment. Scale bar: 20 µm. n = 4 (PRT4165), n = 5 (vehicle) mice. **g**, UMAP of Krt6a expression levels. Scale in

log(counts). **h**, Quantification of Krt6 staining in the infundibulum in vehicle- and PRT4165- treated epidermis. n = 6 mice. **i**, Cell shape is reported as ratio between height and width. n = 7 (PRT4165), n = 10 (vehicle) wound. **j**, Number of pS6+ cells is calculated (left) from whole-mount staining of pS6 (right) in vehicle or PRT4165 treated skin. n = 6 (PRT4165), n = 5 (vehicle) mice. **k**, Epidermal whole-mount images of Lgr5 or Gata6 GL skin treated with PRT4165 or vehicle. **l**, **m**, Epidermal whole-mount showing GL tdTomato+ cells at wound site (left) and quantification of distance from the wound center (right) in vehicle vs PRT4165 treated skin at 1w pw1 in Lgr5 (**l**) or Gata6 (**m**) GL epidermis. n = 6 mice. *P*-value (P) from a two-tailed *t*-test. Data are mean ± SD, if not differently indicated. Scale bars: 1 mm (**l**, **m**); 100 µm (**e**, **f**, **j**, **k**). scRNA-seq data (**a-d**) are the integration of two independent experiments, each of them based on 4 biological replicates.

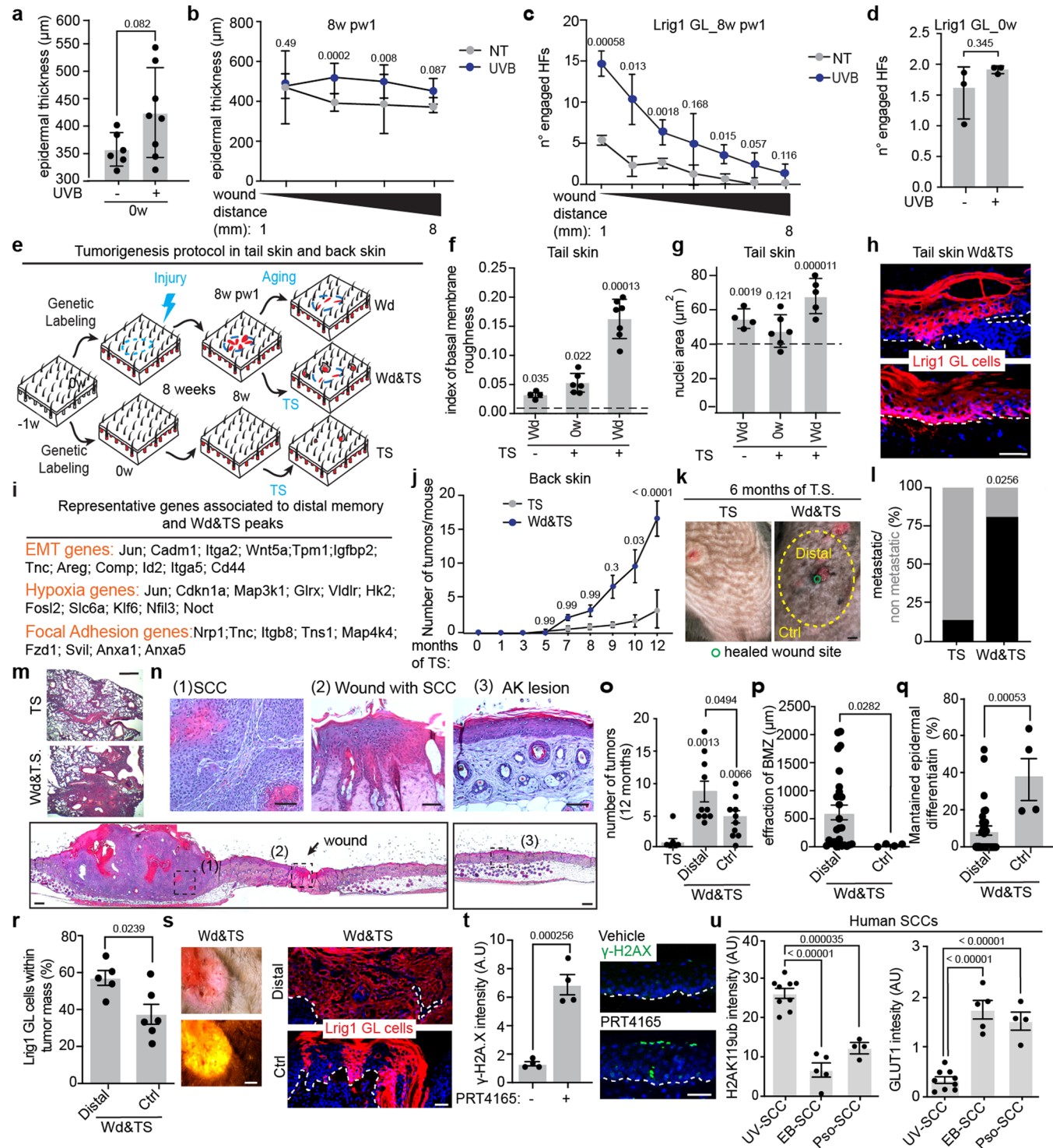

**Extended Data Fig. 10 | Healed skin is prone to SCCs onset. a–d**, Effect of acute UVB irradiation on epidermal thickening at 0w [n = 6 (NT), n = 8 (UVB) mice] (**a**) or 8w pw1 [n = 11 (NT), n = 9 (UVB) mice] (**b**), and on Lrig1 GL HF engagement at 8w pw1 (**c**) or at 0w (**d**) [n = 3 mice]. NT = not treated. **e**, Tumorigenesis protocol: (−1w) genetic labelling; (0w) injury or not; (8w) 8w weeks after, tumorigenic stimuli (TS) started. Samples are wounded (Wd), wounded with TS (Wd&TS) or just TS-treated (TS). **f**, **g**, Characterisation of early SCCs in tail skin. Quantification of membrane roughness (**f**) or nuclei area (**g**). Dashed lines represent 0w mean. n = 4 (Wd), n = 6 (TS), n = 7 (Wd&TS), n = 9 (0w). **h**, Skin sections showing Lrig1 GL cells within eSCCs. **i**, Genes associated with gained peaks in Distal vs Ctrl and with memory peaks and relative GO terms. **j**, Number of tumours in Wd&TS or TS mice, shown as a time course. n = 3 (TS), n = 9 (Wd&TS). Statistics: 2-way ANOVA. **k**, Macroscopic images of Wd&TS or TS back skin. Yellow dashed circle delimits the memory zone (Distal), while outside is the naïve zone (Ctrl). **l**, **m**, Percentage

of mice with metastasis (**l**). n = 6 (TS), n = 8 (Wd&TS). Statistics: Chi-square test. H&E staining of lung metastasis (**m**). **n**, Tiling (down) and magnified images (up) of a Wd&TS skin. AK = Actinic keratosis. **o**, Number of tumours after 12 months of TS. n = 7 (TS), n = 10 (Wd&TS). **p-s**, Characterisation of Distal and Ctrl tumours. Effraction of the basal membrane zone (BMZ). Statistics: Mann-Whitney t-test. n = 4 (Ctrl), n = 27 (Distal) (**p**); percentage of maintained epidermal differentiation over the total tumour area. n = 4 (Ctrl), n = 30 (Distal) (**q**). Percentage (**r**) and skin sections (**s**) showing Lrig1 GL cells within the tumours. n = 5 (Ctrl), n = 6 (Distal). **t**, Quantification (left) and skin sections (right) stained with γ-H2A.X in PRT4165- or vehicle-treated SKH-1 mice. n = 4 mice. **u**, Quantification of H2AK119ub and GLUT1 intensity in human samples. n = 9 (UV-SCCs), n = 5 (EB-SCCs), n = 4 (Pso-SCCs). *P*-value (P) from a two-tailed *t*-test, if not differently indicated. Data are mean ± SD. Scale bars: 2 mm (**k**), 300 μm (**m**), 50 μm (**h**, **n**, **s**, **t**).

# Reporting Summary

## Statistics

For all statistical analyses, confirm that the following items are present in the figure legend, table legend, main text, or Methods section.

| n/a | Confirmed | |
|---|---|---|
| ☐ | ☒ | The exact sample size (*n*) for each experimental group/condition, given as a discrete number and unit of measurement |
| ☐ | ☒ | A statement on whether measurements were taken from distinct samples or whether the same sample was measured repeatedly |
| ☐ | ☒ | The statistical test(s) used AND whether they are one- or two-sided *Only common tests should be described solely by name; describe more complex techniques in the Methods section.* |
| ☐ | ☒ | A description of all covariates tested |
| ☐ | ☒ | A description of any assumptions or corrections, such as tests of normality and adjustment for multiple comparisons |
| ☐ | ☒ | A full description of the statistical parameters including central tendency (e.g. means) or other basic estimates (e.g. regression coefficient) AND variation (e.g. standard deviation) or associated estimates of uncertainty (e.g. confidence intervals) |
| ☐ | ☒ | For null hypothesis testing, the test statistic (e.g. *F*, *t*, *r*) with confidence intervals, effect sizes, degrees of freedom and *P* value noted *Give P values as exact values whenever suitable.* |
| ☒ | ☐ | For Bayesian analysis, information on the choice of priors and Markov chain Monte Carlo settings |
| ☒ | ☐ | For hierarchical and complex designs, identification of the appropriate level for tests and full reporting of outcomes |
| ☐ | ☒ | Estimates of effect sizes (e.g. Cohen's *d*, Pearson's *r*), indicating how they were calculated |

*Our web collection on statistics for biologists contains articles on many of the points above.*

## Software and code

Policy information about availability of computer code

| Data collection | Leica Application Suite X (confocal microscopy) BD FACS Verse, FACSAria II |
|---|---|
| Data analysis | Fiji ( https://imagej.nih.gov/ij/) Graphad prism software (GraphPad, version 7) R studio Seurat Flow Jo |

For manuscripts utilizing custom algorithms or software that are central to the research but not yet described in published literature, software must be made available to editors and reviewers. We strongly encourage code deposition in a community repository (e.g. GitHub). See the Nature Portfolio guidelines for submitting code & software for further information.

## Data

Policy information about availability of data

All manuscripts must include a data availability statement. This statement should provide the following information, where applicable:
- Accession codes, unique identifiers, or web links for publicly available datasets
- A description of any restrictions on data availability
- For clinical datasets or third party data, please ensure that the statement adheres to our policy

Data reported in this paper will be shared by the lead contact upon request. This paper does not report original code. Any additional information required to reanalyse the data reported in this paper is available from the lead contact upon request. scRNA-seq, ATAC-seq and bulk RNA-seq data that support the findings of this study have been deposited in the Gene Expression Omnibus (GEO) under accession code GSE197590.

## Human research participants

Policy information about studies involving human research participants and Sex and Gender in Research.

| | |
|---|---|
| Reporting on sex and gender | Both sex were included in the analysis. No variance in different sexes would be assumed in this study. |
| Population characteristics | Cutaneous SCC samples with AK (UV-SCC) regions were obtained from 9 patients. In addition, cutaneous SCC samples were collected from 3 Recessive Dystrophic Epidermolysis Bullosa (RDEB) (EB-SCC) and 4 psoriasis patients (Pso-SCC), partecipant'ages ranged from 28 to 98.The RDEB patients harboured compound heterozygous mutations in COL7A1 (NM_000094.4) (patient 1: c.5443G>A (p.Gly1815Arg) and c.5819del (p.Pro1940Argfs*65), patient 2: c.5932C>T (p.Arg1978*) and c.8029G>A (p.Gly2677Ser), patient 3: c.7723G>A (p.Gly2575Arg) and c.8569G>T (p.Glu2857*)) |
| Recruitment | Samples were prospectively collected from patients who agreed to participate in the study. |
| Ethics oversight | The institutional review board of the Hokkaido University Graduate School of Medicine approved the human study described (ID: 13-043, 14-063, and 15-029). The study was carried out according to the Declaration of Helsinki Principles. The participants provided written informed consent. |

Note that full information on the approval of the study protocol must also be provided in the manuscript.

# Field-specific reporting

Please select the one below that is the best fit for your research. If you are not sure, read the appropriate sections before making your selection.

☒ Life sciences ☐ Behavioural & social sciences ☐ Ecological, evolutionary & environmental sciences

For a reference copy of the document with all sections, see nature.com/documents/nr-reporting-summary-flat.pdf

# Life sciences study design

All studies must disclose on these points even when the disclosure is negative.

| | |
|---|---|
| Sample size | No statical method was used to determine the sample size prior to the study. |
| Data exclusions | No data were excluded. |
| Replication | Number of each replicates for each experiments is written in corresponding figures and figure legends. |
| Randomization | No randomization was done. Mice were categorised based on genotype. |
| Blinding | No blinding is done since the same researchers performed both data acquisition and analysis. |

# Reporting for specific materials, systems and methods

We require information from authors about some types of materials, experimental systems and methods used in many studies. Here, indicate whether each material, system or method listed is relevant to your study. If you are not sure if a list item applies to your research, read the appropriate section before selecting a response.

## Materials & experimental systems

| n/a | Involved in the study |
|-----|------------------------|
| ☐ | ☒ Antibodies |
| ☒ | ☐ Eukaryotic cell lines |
| ☒ | ☐ Palaeontology and archaeology |
| ☐ | ☒ Animals and other organisms |
| ☒ | ☐ Clinical data |
| ☒ | ☐ Dual use research of concern |

## Methods

| n/a | Involved in the study |
|-----|------------------------|
| ☒ | ☐ ChIP-seq |
| ☐ | ☒ Flow cytometry |
| ☒ | ☐ MRI-based neuroimaging |

# Antibodies

| | |
|---|---|
| Antibodies used | Sca-1 (clone E13-161.7) (PE/Cyanine7 conjugated, Biolegend_122514), Glut1 (clone EPR3915) (Alexa Fluor 647 conjugated, Abcam_ab195020), CD45 (VioGreen, Miltenyi Biotec_130-110- 803), CD11b (clone M1/70) (FITC, Miltenyi Biotec_130-110-803), CD3 (clone 17.A2) (FITC, Miltenyi Biotec_130130-119-135), ydTCR (clone REA633) (PE-Vio770, Miltenyi Biotec_130-123-290), F4/80 (clone REA126) (PE-Vio770, Miltenyi Biotec_130-118-320), MHC-II (APC, Miltenyi Biotec_130-102-139), CD206 (clone C068C2) (PE, Biolegend_141706), IL-17A (PE, Biolegend_506903), cytokeratin 14 (clone LL002, Invitrogen), GM130 (clone 35, BD pharmingen), F4/80 (clone REA126, abcam_ab6640), anti-TNFa (clone MP6-XT22) (Brilliant violet 421 conjugated, biolegend_506327), Phospho-Histone H2A.X ((Ser139)CST_9718), SLC2A1 (GLUT1) (Sigma-Aldrich_ HPA031345),  SLC2A1 (GLUT1) (clone EPR3915, abcam_ab115730), Ubiquityl-Histone H2A (Lys119) (clone D27C4, CST_8240), Keratin 6A (Biolegend_905701), anti-phospho-S6 Ribosomal Protein (Ser235/236) (clone D57.2.2E, CST_4858), anti-Ki67 (Abcam_16667), anti-E-cadherin (clone 24E10, CST_3195), anti-mFABP5 (RD System_AF1476). |
| Validation | All antibodies are well characterized and were applied according to data sheet instructions or previously published protocols. |

# Animals and other research organisms

Policy information about studies involving animals; ARRIVE guidelines recommended for reporting animal research, and Sex and Gender in Research

| | |
|---|---|
| Laboratory animals | Rosa26-fl/STOP/fl-tdTomato (Madisen et al., 2010), Lgr5-EGFP-ires-CreERT2 (Jaks et al., 2008), Gata6-EGFP-ires-CreERT2 (Donati et al., 2017), Lrig1-EGFP-ires-CreERT2 (Page et al., 2013) and SKH-1 hairless mice were used. |
| Wild animals | Not applicable |
| Reporting on sex | Both sexes were used in this study, untless otherwise specified. For the scRNA-seq, female mice were selcted to avoid wounds from fighting in the cage. |
| Field-collected samples | Not applicable |
| Ethics oversight | Maintenance, care and experimental procedures have been approved by the Italian Ministry of Health, in accordance with Italian legislation (authorization no. 117/2018-PR), and the institutional review board of the Hokkaido University Graduate School of Medicine (authorization no. 22-0028). |

Note that full information on the approval of the study protocol must also be provided in the manuscript.

# Flow Cytometry

## Plots

Confirm that:

☒ The axis labels state the marker and fluorochrome used (e.g. CD4-FITC).

☒ The axis scales are clearly visible. Include numbers along axes only for bottom left plot of group (a 'group' is an analysis of identical markers).

☒ All plots are contour plots with outliers or pseudocolor plots.

☒ A numerical value for number of cells or percentage (with statistics) is provided.

## Methodology

| | |
|---|---|
| Sample preparation | Whole skin: murine tail skin was collected, minced and incubated overnight  in a collagenase D-pronase mixture to obtain single cell suspension. The suspension was then stained blocked with 3% FBS in PBS and then stained 30 min with the indicated dose of fluorophore-conjugated antibody.<br>Epidermal cells: murine tail skin was collected and epidermis isolated through overnight incubation with trypsin EDTA (0,25%). The epidermis was then minced and single cell suspension obtained for staining. |
| Instrument | FACS Verse and FACSAria II were used. |

| Software | FACS Diva and FlowJo (FlowJo™ v10.8) were used. |

| Cell population abundance | Final sorted populations made up of approximately 25% (Lrig1 GL), 4% (Gata6 GL) of 1% (Lgr5 GL) of tdTomato+, 5-10% of Lrig1 GL tdTomato+Sca1+, or 1-3% of Lrig1 GL tdTomato+Glut1+ of all cells. sc-RNAseq  and bulk RNA-seq confirmed the identity of sorted cells. |

| Gating strategy | Gates were set to exclude debris and doublets: live cells (TOPRO-3/FSC-A), cell size (SSC-A/FSC-A), singlets (FSC-H/FSC-A). Gates on stained population were set on negative unstained controls and compensation was applied based on single-stained controls. |

☒ Tick this box to confirm that a figure exemplifying the gating strategy is provided in the Supplementary Information.

