## [Peer Review File · Nature Cell Biology]

Peer Review Information

Journal: Nature Cell Biology

Manuscript Title: Lifelong tissue memory relies on spatially organised wound-distal progenitors

Corresponding author name(s): Professor Giacomo Donati

Editorial Notes:

Reviewer Comments & Decisions:

Decision Letter, initial version:
--

Dear Professor Donati,

Your manuscript "Lifelong tissue memory relies on spatially organized dedicated progenitors located distally from the injury", has now been seen by 3 referees, who are experts in skin wound repair, lineage tracing, single cell RNA-seq (referee 1); skin stem cells, wound repair, cell memory (referee 2); and single cell transcriptomics, epigenetics, skin, priming (referee 3), and whose comments are pasted below. In light of their advice, we regret that we cannot offer to publish the study in Nature Cell Biology.

As you will see, although the reviewers find this work interesting, they raise serious concerns that question the strength of the data and of the novel conclusions that can be drawn at this stage, including concerns with the wound-inducing protocol and claims regarding "distal" and "cell-autonomous" memory, but also cell priming.

We would be open to the possibility of considering a revised manuscript that would fully address the referee concerns, which you would have to submit as an appeal. However, any decision to re-review such a revised study would depend on the strength of the revisions and the published literature at the time of resubmission.

We are very sorry that we could not be more positive on this occasion, but we thank you for the opportunity to consider this work.

With kind regards,
Stelios

Stylianos Lefkopoulos, PhD
He/him/his
Associate Editor
Nature Cell Biology
Springer Nature
Heidelberger Platz 3, 14197 Berlin, Germany

E-mail: stylianos.lefkopoulos@springernature.com
Twitter: @s_lefkopoulos

Reviewers' comments:

Reviewer #1 (Remarks to the Author):

Levon et. al., discovered that genetically labeled Lrig1+ cells located at long distance from the wound site acquired a primed memory after the first injury, a 2mm wound in the tail skin. Specifically, these cells delocalized in the infundibulum region post-injury induction and constitutively expressed plasticity genes involved in epithelial to mesenchymal transition, focal adhesion, and glycolysis, among others. Some of the same gene signature expressed in trained cells close to the wound (Gonzalez et al. 2021).

This transcriptional pre-activation resulted in an enhanced capacity of the cells to respond to a second insult. Furthermore, they have confirmed previous discoveries in Gonzalez et al., 2021 that hair follicle stem cells localized in the bulge (Lgr5+) have a trained adaptation post-wound induction. These cells have increased expression of the plasticity genes during wound repair. However, after injury, they locate in their own niche and shut down the expression of those genes while still able to have an enhanced response to wound induction. The authors further tested the role of the physiological reduction of H2AK119ub as a key event on the de-repression of the primed genes after injury repair. Finally, they identify the wound-distal primed cells as a possible source of tumor formation induced by UVB-treatment.

Originality and significance: These studies build on the previous knowledge about the presence of trained epithelial stem cells with the capacity to enhance the repair of a second injury (Gonzalez et al. 2021). They discovered that a single injury changes long term the transcriptional activity of a specific stem cells in the junctional zone of the hair follicle (Lrig1+) distant for the site of injury. These changes in transcription profile in one hand facilitate a second wound-repair but on the other hand increase the skin vulnerably to tumor formation. These discoveries especially the one related to long-term memory are really interesting and important for the field.

Data and methodologies: validity of the approach quality of data, quality of presentation– focus on the functional characterization of the primed wound cells – They performed an overall good spatial and temporal characterization of the Lrig1+ primed memory stem cells.

Appropriate use of statistics: The statistics appear appropriate for the kind of the analyses they performed.

Conclusion: robustness, validity, reliability. Their conclusions appear reliable and robust.

Suggested improvements:

The manuscript would benefit of a more precise wound induction protocol. At which age were the wounds performed? More details and images for the hair follicle cycle stages at the analyzed time points and at the different distances from the wound can help complete and better interpret the characterization in Ext Fig 1g. This will be helpful as to understand how the various stages (i.e. rest, growth etc) could influence the performed analysis in the comparisons with pre-injury mice.

It seems that when the second wound is applied at 8w PW1, the first wound is not completely resolved. Specifically, the thickness and the cell density are still significantly elevated within 1 mm away from the wound (Ext Fig 1i h). The authors should comment on this and explain what features do they use in order to say that there is a newly established homeostasis. For instance, the expression of K6a marker could be an indication of injury-induced stress. Indeed, it was downregulated after 40 weeks post-wound induction.

Is it possible to see the localization of the Lrig1+ cells in the interfollicular epithelium at 5-6 mm away from the second wound (such as a close-up of Fig. 1e)? It is an interesting phenomenon and having this resolution will strengthen the author conclusions.

Fig. 1f data are interesting. Hair follicle stem cells far away from the first wound site have an enhanced capability to close the wound similarly to what observed for cells close to the injury. Can the authors clarify what distance these results refer to: 1-3 mm / 3-5 mm / 5-7 mm from the pre-made wound or all of them? Essentially it would be important to clarify how far from the wound can the authors see this behavior and the primed memory cells.

From the Materials and Methods section, it is not clear if the authors performed scRNA-seq experiments only in the distal part of the wound. Have the authors excluded tissue from the first 2mm from the wound that contain the previously identified memory cells by Gonzalez et al., 2021). Would be helpful to specify it.

How long the primed memory cells are maintained in the young or adult mice (first wound protocol

used)? It is not clear if to assess the primed memory genes the same mice analyzed at 8 weeks after the first injury are then analyzed again at 40 weeks to look at the genetic signature is maintained. More details in the protocol are needed to better orient the readers and interpret the data.

Statistical analysis is missing in Extended Data Fig. 5g.

The figures are extremely dense and difficult to navigate. Moving some material to the supplementary can increase the capacity of readers to extract all the important points from this paper.

Reviewer #2 (Remarks to the Author):

In their manuscript entitled "Lifelong tissue memory relies on spatially organized dedicated progenitors located distally from the injury", Levron and colleagues examine the spatial extent of wound memory and the spectrum of the adaptive responses of epithelial cells. To do so, they adapted lineage tracing and single-cell RNA-seq to identify wound distal-memory cells. They uncover that sub-organ scale adaptation of an injury relies on spatially organized and memory-dedicated progenitors, which were determined by an epigenetic actionable cell, that can lead to tumor onset.

I have significant concerns about the on the key claims of the study- namely "distal memory". The authors define distal based on a 5mm distance from initial wound as a differentiator from prior studies, which have defined distal wound as contralateral skin (presumably outside their memory zone). Based on differences in 5mm distance the authors then claim that their findings are entirely distinct, when in fact these findings are entirely consistent with previous works indicating that >2cm do not have memory (Naik et al Nature 2017) and that damaged follicles communicate with adjacent HFs during repair via TNFa (Chen et al Cell 2015). This conclusion should be toned down significantly. Below I have outlined additional concerns:

Major comments:

- The claim of cell autonomous memory is exaggerated as no cell transfer experiments or in vitro studies that address the role of the microenvironment as noted. Additionally, very recently IL-17 signaling from wound immune cells was shown to induce mTOR and HIF1a. Thus, the "memory" seen here could also be dependent on changes to the microenvironment. In line 147 the authors dissect if distal memory is "cell-autonomous" using skin biopsies and ex vivo migration assay. However, it is not clear if the results presented in Fig.2F and G are from ex vivo skin biopsy or keratinocytes culture. If it is only an ex vivo experiment the authors should not claim that it is cell-autonomous because skin biopsy contains not only keratinocytes but also other immune cells. In this context, they should perform an experiment using a pure population of keratinocytes.
- In line 108 the authors claimed that in their experiments, the second wound perfectly overlaps the first one, allowing the removal of the HF-derived IFESCs. However, there are no data either by imaging or flow cytometry to support this statement. The faster migration of Lrig1+ lineages after the second wound could still be contributed by the IFESCs derived during the first healing.
- For figures 2A and B what is the strategy of sampling for RNA sequencing? Such as location relative to wound margins and sorting strategy. Are the IFESCs included in the sample for sequencing? The clarification of strategy is important for a proper understanding of the results.
- Line 152 statement about Fig.2h and Fig. 2I is not clear. Does Fig.2H represent the quantification for Fig.2I? If so, it is a percentage of polarized cells not the expression like the authors mentioned in the

text. The text does not correspond to the figure legend.

- Please clarify the sampling location for single-cell RNA-seq in Fig.3. Based on UMAP plots it looks like cells from 8w pw1 sample also contribute to differentiated IFE clusters. So, is the sampling also include IFESCs derived from first healing?
- Based on UMAP plots in Fig.3A-3C in cluster 7 the samples 1w pw1 and 1w pw2 contribute only to the IFE epidermis. If so, in Fig.3I and Fig.3J authors are just comparing the transcript among cells with different locations. It is better to show the staining for INFU memory population markers defined by the authors in different time points between 0w and 8w pw1 to see if those cells are continually located in INFU or if those cells are retracted from IFE after healing.
- Which genes were used to define the average z-score expression of cell plasticity genes presented in Fig.4B
- It is unclear how the authors determined the inhibitors for epigenetic drugs screening

Minor comments:

- In general, figures are very condensed which sometimes is difficult to read. Some data/schematics are redundant and can be moved to supplementary material. Making these figures more accessible will greatly aid the reader and more effectively convey the findings
- What does mean the red dashes in Fig.1E?
- What is the distance of the image presented in Fig.3K from the wound site?
- Line 157 and 163 Fig.2J should be Fig2K.
- Line 262 Extended data Fig.2B should be Extended data Fig.2H?
- Fig.6 there is no L.
- For Figure 2A and Extended figure 2A, please clarify based on the comparison of which groups the DEG is calculated?
- Line 311 and 312 mistakes in references

Reviewer #3 (Remarks to the Author):

A. Summary of the key results

The authors characterize a mouse wound healing model that is based on 2 subsequent injuries at the same site. They identify a subpopulation of distal cells that becomes primed during the first injury. It is then hypothesized that this state differs from previously described states of wound memory. It is further shown that a chromatin modulatory drug (PRT4165) can modulate wound memory and preliminary evidence is supported to suggest that wound memory promotes tumor (squamous cell carcinoma) formation.

B. Originality and significance: if not novel, please include reference

Wound memory is not a novel concept and a recent publication (ref. 15) has shown that differential chromatic accessibility underpins the phenomenon. The authors make a valid point in stating that the precise lineage identity and spatial distribution remain to be fully characterized. They also show that their memory cells are distantly located to the site of injury, which appears novel. Finally, the link between wound priming and H2AK119ub as well as the connection between wound priming and skin cancer formation are novel and potentially exciting, albeit poorly substantiated.

C. Data & methodology: validity of approach, quality of data, quality of presentation

Overall, the approach is characterized by a combination of lineage tracing, transcriptomics and histology. The results are comprehensive and carefully presented. However, additional experiments

using different methods are required to substantiate key claims (see below).

D. Appropriate use of statistics and treatment of uncertainties

I'm not an expert, but I did not notice any major deficiencies in statistics.

E. Conclusions: robustness, validity, reliability

The authors do not provide any mechanistic insights into wound priming, which I consider a major deficiency, as (1) ref. 15 has already provided ATAC-seq data to characterize the epigenetic states associated with wound memory and (2) it would have been straightforward to use the Chromium single-cell multiome approach to provide integrated scRNA-seq and scATAC-seq profiles.

I also have reservations about use of PRT4165 for the functional experiments (Fig. 6). The rationale for the use of the drug is very weak (a "screen" consisting of 5 (!) epigenetic drugs, EFig. 6a). It is also not clear whether the observed effects are due to changes in epigenetic memory or rather due to side effects, e.g. on DSB repair. ATAC-seq and/or H2AK119ub ChIP-seq analyses are required to substantiate the initial findings.

Finally, I have reservations about the claim that wound priming is relevant for the formation of mouse and human squamous cell carcinoma (Fig. 7). The hypothesis is certainly interesting and the underlying mechanisms might be similar to the "epigenetic field effect" that has been proposed to support tumor formation. However, my enthusiasm is dampened by the superficiality of the analysis in Fig. 7 and by the lack of mechanistic/epigenetic insight, as outlined above.

F. Suggested improvements: experiments, data for possible revision

See point 6 above. We need a clearer picture about the epigenetic landscape that defines the wound priming state. Additionally, the mode of action of PRT4165 needs to be better analyzed and controlled. Finally, the mechanism that links wound priming to tumor formation needs to be defined. In the absence of these data, the conceptual advance over ref. 15 appears highly limited.

G. References: appropriate credit to previous work?

Yes (but I'm not a specialist in the field of wound healing).

H. Clarity and context: lucidity of abstract/summary, appropriateness of abstract, introduction and conclusions.

Fine overall, but the level of clarity and accessibility could be improved.

**Although we cannot publish your paper, it may be appropriate for another journal in the Nature Portfolio. If you wish to explore the journals and transfer your manuscript please use our manuscript transfer portal. You will not have to re-supply manuscript metadata and files, but please note that this link can only be used once and remains active until used. For more information, please see our manuscript transfer FAQ page.

Note that any decision to opt in to In Review at the original journal is not sent to the receiving journal

on transfer. You can opt in to In Review at receiving journals that support this service by choosing to modify your manuscript on transfer. In Review is available for primary research manuscript types only.

Author Rebuttal to Initial comments

(B) Point-by-point Response to Reviewers' comments

Authors' comments:

- We thank all the reviewers for their thorough analysis of our manuscript and for the constructive suggestions.
- For an easier assessment by the editor and the reviewers, the major changes in the main text with respect to the first submission are highlighted in blue, as well as the letters of the newly inserted panels and the corresponding figure legends.

Reviewer#1:

Levon et. al., discovered that genetically labeled Lrig1+ cells located at long distance from the wound site acquired a primed memory after the first injury, a 2mm wound in the tail skin. Specifically, these cells delocalized in the infundibulum region post-injury induction and constitutively expressed plasticity genes involved in epithelial to mesenchymal transition, focal adhesion, and glycolysis, among others. Some of the same gene signature expressed in trained cells close to the wound (Gonzalez et al. 2021).

This transcriptional pre-activation resulted in an enhanced capacity of the cells to respond to a second insult. Furthermore, they have confirmed previous discoveries in Gonzalez et al., 2021 that hair follicle stem cells localized in the bulge (Lgr5+) have a trained adaptation post-wound induction. These cells have increased expression of the plasticity genes during wound repair. However, after injury, they locate in their own niche and shut down the expression of those genes while still able to have an enhanced response to wound induction. The authors further tested the role of the physiological reduction of H2AK119ub as a key

event on the de-repression of the primed genes after injury repair. Finally, they identify the wound-distal primed cells as a possible source of tumor formation induced by UVB-treatment.

Originality and significance: These studies build on the previous knowledge about the presence of trained epithelial stem cells with the capacity to enhance the repair of a second injury (Gonzalez et al. 2021). They discovered that a single injury changes long term the transcriptional activity of a specific stem cells in the junctional zone of the hair follicle (Lrig1+) distant for the site of injury. These changes in transcription profile in one hand facilitate a second wound-repair but on the other hand increase the skin vulnerably to tumor formation. These discoveries especially the one related to long-term memory are really interesting and important for the field.

We thank the Reviewer for recognising that, starting from the finding of Gonzales et al 2021, we moved forward providing new results that are “really interesting and important for the field”.

Data and methodologies: validity of the approach quality of data, quality of presentation– focus on the functional characterization of the primed wound cells – They performed an overall good spatial and temporal characterization of the Lrig1+ primed memory stem cells.

Appropriate use of statistics: The statistics appear appropriate for the kind of the analyses they performed.

Conclusion: robustness, validity, reliability. Their conclusions appear reliable and robust.

We thank the Reviewer for highlighting the reliability and robustness of our experimental settings and conclusions.

Suggested improvements:

1) The manuscript would benefit of a more precise wound induction protocol. At which age were the wounds performed? More details and images for the hair follicle cycle stages at the analyzed time points

and at the different distances from the wound can help complete and better interpret the characterization in Ext Fig 1g. This will be helpful as to understand how the various stages (i.e. rest, growth etc) could influence the performed analysis in the comparisons with pre-injury mice.

We recognise the lack of details that we have now included in the revised Methods section “full-thickness skin wound”. In particular, the wound (that was always performed 1 week after tamoxifen administration) was carried out in between 7- and 9-weeks post birth when most of the hair follicles are in the resting phase (telogen).

Concerning hair follicle cycle stages, as the Reviewer suggested we now provided an improved characterization at all the time points and in a distance-dependent manner from the wound site through immunofluorescence images and quantitative analysis (Extended Data Fig.1h).

2) It seems that when the second wound is applied at 8w PW1, the first wound is not completely resolved. Specifically, the thickness and the cell density are still significantly elevated within 1 mm away from the wound (Ext Fig 1i h). The authors should comment on this and explain what features do they use in order to say that there is a newly established homeostasis. For instance, the expression of K6a marker could be an indication of injury-induced stress. Indeed, it was downregulated after 40 weeks post-wound induction.

Thank you for bringing up this point. The characterisation of the new homeostasis (8w pw1) has been updated in the manuscript with new data (Extended Data Fig.1h,i and Extended Data Fig.2a-n). Accordingly, we amended the text and the Method section “full-thickness skin wound”. In summary, the features that we used to define 8w pw1 as a new the homeostasis are: (1) Complete re-epithelialisation. Although two weeks after wounding are sufficient for a complete re-epithelialisation, we waited 6 additional weeks post wounding until 8w pw1 (Extended Data Fig.1f); (2) Re-establishment of a homeostatic epidermal differentiation program. Comparable FABP5 differentiation marker distribution is observed at 8w pw1 in the wounded area when compared with original homeostasis (0w) (Extended Data Fig.1i); (3) Comparable hair follicle cycle phase. The ratio of HFs in early anagen, late anagen and telogen at 0w and at 8w pw1 is equal (Extended Data Fig.1h). (4) Resolution of wound-associated inflammation. The increase of immune infiltrate that characterises the healing phases (1w pw1 and 1w pw2) is completely absent at 8w pw1, as shown by the detailed flow cytometry analysis and immunofluorescence of immune cells, that we now provide at all the time points and in a distance-dependent manner (Extended Data Fig.2a-n).

Because of the above reasons, since the slight increase in dermal cell density and thickness observed at 8wpw1 did not resolve even at longer time points, 8w pw1 was pointed as a new homeostasis. Importantly, these dermal features are completely restricted to the wound bed (the zone “A” in the manuscript) and they are absent from 0 to 7 mm apart, where the distal memory cells have been identified (Extended Data Fig.1j, k). In addition, the zone “A” is completely removed with the second injury.

3) Is it possible to see the localization of the Lrig1+ cells in the interfollicular epithelium at 5-6 mm away from the second wound (such as a close-up of Fig. 1e)? It is an interesting phenomenon and having this resolution will strengthen the author conclusions.

We thank the Reviewer for this helpful suggestion. In response, we integrated our whole-mount data at 1w pw1 and 1w pw2 (Fig.1f) with new representative horizontal whole-mount images of the wound bed and the distal memory region at 1w pw1, 8w pw1 and 1w pw2 (Extended Data Fig.3d and Fig.1g). The results show that Lrig1 GL distal memory cells occupy the infundibulum at 1w pw1 and 8w pw1, while they exit the HF toward the IFE as basal and suprabasal after the second injury, at 1w pw2.

4) Fig. 1f data are interesting. Hair follicle stem cells far away from the first wound site have an enhanced capability to close the wound similarly to what observed for cells close to the injury. Can the authors clarify what distance these results refer to: 1-3 mm / 3-5 mm / 5-7 mm from the premade wound or all of them?

As we now specify in Fig.1h Legend the distance at which we performed the second wound is 1 to 3 mm from the first wound (1w pw2_Distal), enabling us to evaluate the repair ability of HF cells localised from 4 mm from the first wound.

Essentially it would be important to clarify how far from the wound can the authors see this behavior and the primed memory cells.

To answer to this point, we performed an additional analysis (added to Extended Data Fig. 3f,g) complementing the results reported in Figure 1h. As highlighted in Extended Data Fig.3f, the HFs are engaged from 4 to 7 mm from the wound.

5) From the Materials and Methods section, it is not clear if the authors performed scRNA-seq experiments only in the distal part of the wound. Have the authors excluded tissue from the first 2mm from the wound that contain the previously identified memory cells by Gonzalez et al., 2021. Would be helpful to specify it.

We now clarified the experimental setting in the Method section “scRNA-seq”. The scRNA-seq experiment reported in Fig.3a-g, Fig.4a, b, e and Extended Data Fig.5a-j and Extended Data Fig.7a-b was performed to include both wound bed and distal area. However, the cellular features (i.e. cell shape) and the molecular markers (i.e. KRT6 and GLUT1) of memory, indicated by this scRNA-seq, were validated by histology in the original submission. To strengthen this point, we added a new scRNA-seq that analyses individually the wound bed, the distal memory region, and a faraway non-memory control area (Fig.3i and Extended Data Fig.6). Importantly, these new data confirm our previous statement, identifying the transcriptional priming (memory) also in distal HFs, up to 7 mm from the wound bed.

6) How long the primed memory cells are maintained in the young or adult mice (first wound protocol used)? It is not clear if to assess the primed memory genes the same mice analyzed at 8 weeks after the first injury are then analyzed again at 40 weeks to look at the genetic signature is maintained. More details in the protocol are needed to better orient the readers and interpret the data.

We apologise for the lack of clarity. A new paragraph describing the procedure is included in the Method section “Evaluation of long term memory “.

7) Statistical analysis is missing in Extended Data Fig. 5g.

The missing information has been added in the figure panel (Now Extended Data Fig.8g) -Thank you for spotting this.

8) The figures are extremely dense and difficult to navigate. Moving some material to the supplementary can increase the capacity of readers to extract all the important points from this paper.

Following this important suggestion of the Reviewer (as well as of Reviewer 2), several figure panels have been moved to Extended Data Figures, thus improving the clarity of the key results.

Reviewer #2 (Remarks to the Author):

1) In their manuscript entitled “Lifelong tissue memory relies on spatially organized dedicated progenitors located distally from the injury”, Levron and colleagues examine the spatial extent of wound memory and the spectrum of the adaptive responses of epithelial cells. To do so, they adapted lineage tracing and single-cell RNA-seq to identify wound distal-memory cells. They uncover that sub-organ scale adaptation of an injury relies on spatially organized and memory-dedicated progenitors, which were determined by an epigenetic actionable cell, that can lead to tumor onset.

I have significant concerns about the on the key claims of the study- namely “distal memory”. The authors define distal based on a 5mm distance from initial wound as a differentiator from prior studies, which have defined distal wound as contralateral skin (presumably outside their memory zone). Based on differences in 5mm distance the authors then claim that their findings are entirely distinct, when in fact these findings are entirely consistent with previous works indicating that >2cm do not have memory (Naik et al Nature 2017) and that damaged follicles communicate with adjacent HFs during repair via TNFa (Chen et al Cell 2015). This conclusion should be toned down significantly. Below I have outlined additional concerns:

We thank this Reviewer for his/her comment. Specifically, to address the concern of the novelty on “distal memory”, we would like to clarify the distinction between previous work and our work, as well as the main novelty in our manuscript.

In the excellent work from Naik S. et al. Nature 2017, the wound memory protocol is based on a 2 cm² abrasion (1st wound) followed by a 2nd injury of a 6 mm² punch biopsy, within the previously wounded area. Since the epidermis surrounding the 6 mm² biopsy has already been wounded by a space-wide abrasion, distal memory has not been assessed in that work. Indeed, Naik et colleagues deeply focus on the memory of epithelial cells residing in the first wound bed.

In Chen C. et al. Cell 2015, now cited in the manuscript, the authors elegantly demonstrated that, in hair follicle plucking context, when HF's are plucked within a 5 mm circular area, anagen induction spread to not plucked HF's within an area of ~1 mm surrounding the plucked area. A similar spatial response occurs also in wound context where the cell contribution to skin full thickness wound repair is spatially restricted to 1 mm away from the injury (Park S. et al. Nature Cell Biology 2017).

Unlike previous work, here we show that, a 2 mm² lesion educates HF cells residing up to 7 mm from the wound bed, with a wide spatial extent that has never been seen before. Therefore, as pointed out by the other two Reviewers, we believe that distal memory, as a sub-organ scale adaptation of an injury, is a major novelty in for the skin cell biology field, as well as for epithelia-related studies, in general. To our knowledge there are not any reports that focus on the spatial extent of the wound consequences on epithelial cells (i.e. wound memory) and the main consensus in the field is that they are restricted to wound close proximity (i.e. Park S. et al. Nature Cell Biology 2017; Aragona M. et al. Nat Commun 2017).

Additionally, Chen C. et al. Cell 2015 nicely described the ability of damaged follicles to communicate with adjacent HF's via a macrophage-released TNF α in the context of HF regeneration. We agree about the importance of immune system in our experimental setting. Therefore, we complemented the manuscript with the characterization of immune infiltrate, in particular macrophages and T cells, as suggested (see our response for this Reviewer's point 2, regarding T cells). Through a detailed flow cytometry and immunostaining data (Extended Data Fig.2d-i) we now show that, although TNF α -expressing macrophages are massively recruited during acute phase of wound healing (1w pw1 and 1w pw2, only in the wound bed) they are absent at 8w pw1, both in the wound bed and in the distal memory areas. Our new data are consistent with a general scenario in which TNF α -expressing macrophages are recruited specifically during an acute phase of repair but dissipated around 7 days after the damage as in Chen C. et al. Cell 2015. These data suggest that macrophage-released TNF α is not involved at 8w pw1 in the maintenance of priming in distal memory Lrig1GL cells.

2) Major comments:

- The claim of cell autonomous memory is exaggerated as no cell transfer experiments or in vitro studies that address the role of the microenvironment as noted. Additionally, very recently IL-17 signaling from wound immune cells was shown to induce mTOR and HIF1a. Thus, the "memory" seen here could also be dependent on changes to the microenvironment. In line 147 the authors dissect if distal memory is "cell-autonomous" using skin biopsies and ex vivo migration assay. However, it is not clear if the results presented in Fig.2F and G are from ex vivo skin biopsy or keratinocytes culture. If it is only an ex vivo experiment the authors should not claim that it is cell-autonomous because skin biopsy contains not only keratinocytes but also other immune cells. In this context, they should perform an experiment using a pure population of keratinocytes.

We thank the Reviewer for raising this important point. We added new experiments and we amended the manuscript accordingly to specify that our claim of the cell-intrinsic nature of priming is only referred to the "maintenance" of priming. We proved this claim through multiple in vitro cellular and omic assays (see below).

The experiments reported in the previous submitted Fig.2f and 2g (now Fig.2e and 2f) as well as the experiments in Extended Data Fig.4h-j were indeed performed on isolated keratinocytes in culture and they sustained that, once established, priming is overall maintained *in vitro* in the absence of their *in vivo* physiological microenvironment (which is imperative for our conclusions, as pointed by the Reviewer). Indeed, these experiments show that even in culture conditions 8w pw1 distal memory cells have enhanced migratory ability (Fig.2e, f) as well as increased adhesion and survival features (Extended Data Fig.4i, j). To improve the clarity, we revised the main text and added detailed information in the related Figure Legends.

To further support that priming is overall maintained in the absence the *in vivo* physiological microenvironment, we performed an additional set of experiments where we profiled the gene expression and the chromatin landscape of cultured distal memory cells.

The new RNA-seq of Lrig1 GL keratinocytes isolated from Distal memory area or Ctrl area, after 7 days of culture, highlights an enhanced migratory profile of distal memory cells when compared with Ctrl cells (Fig.2g and Extended Data Fig.4k). This nicely fits with the result of the migration assays that we previously performed (Fig.2e, f).

The differences in chromatin accessibility between cultured Lrig1 GL cells from memory Distal and Ctrl area using ATAC-seq (Fig. 5g-j) were compared with the chromatin accessibility landscape of freshly sorted distal memory cells (Fig. 5a-f). The new analysis highlights a significantly consistent overlap between in vivo and cultured memory cells in terms of gained peaks associated genes (Fig.5h) and GO categories (Fig.5i).

Our in vitro data indicate that the chromatin remodelling and the transcriptional/phenotypic consequences of the distal memory cells, once acquired, are overall maintained in the absence of microenvironmental stimuli.

As the Reviewer suggested, we experimentally addressed the involvement of IL-17 signalling in distal priming. Indeed, as nicely shown in Konieczny et al. Science 2022, the IL-17 released by the $\gamma\delta$ T cells drives re-epithelization, inducing a glycolytic metabolism which is dependent upon mTOR and HIF1 α . Through new histological and flow cytometry experiments, we quantified lymphocytes, as major skin source of IL-17 (Konieczny P. et al. Science 2022), and the relative IL-17 production (Extended Fig.2.j-n). The results show an increase of T cells and IL-17 production during the acute wound healing phase (as described in Konieczny et al. Science 2022), but similarly to macrophages and TNF α , neither lymphocytes nor IL-17 production are upregulated in 8w pw1 skin, in the wound bed or in the distal memory area.

These data indicate that distal priming is maintained even in the absence of the induction of TNF α and IL-17 or the infiltrate of macrophages and T cells.

Overall, we recognise that immune cells might play a role in the establishment of epithelial Lrig1 GL cell priming. However, we clearly demonstrate that distal priming can be maintained in vitro. Therefore, we now specify in the manuscript that priming is intrinsic to memory cells as it is maintained without in vivo external/microenvironmental signals.

3) • In line 108 the authors claimed that in their experiments, the second wound perfectly overlaps the first one, allowing the removal of the HF-derived IFESCs. However, there are no data either by imaging or flow cytometry to support this statement. The faster migration of Lrig1+ lineages after the second wound could still be contributed by the IFESCs derived during the first healing.

To address this question, we have added a representative image of the first wound bed removal (Extended Data Fig.3a).

4) • For figures 2A and B what is the strategy of sampling for RNA sequencing? Such as location relative to wound margins and sorting strategy. Are the IFESCs included in the sample for sequencing? The clarification of strategy is important for a proper understanding of the results.

We thank the Reviewer and we apologise for the lack of clarity describing this experimental setting. This mini bulk RNA-seq was performed from sorted (Tomato+) Lrig1 GL cells from both wound bed and distal areas (thus the IFESCs are included) as we now report in Method section “Mini-bulk RNA-seq and analysis”. The RNA-seq data were used to identify putative adaptation behaviours of memory cells, as the GO analysis highlighted an enhanced migration as memory feature (Fig.2b). The validation of this feature was performed on wound-educated cells located distally from the wound through ex vivo and in vitro assays (Fig.2c-f).

5)• Line 152 statement about Fig.2h and Fig. 2I is not clear. Does Fig.2H represent the quantification for Fig.2I? If so, it is a percentage of polarized cells not the expression like the authors mentioned in the text. The text does not correspond to the figure legend.

We apologise for this mistake – thank you for noticing this.

Previously Fig.2h, now Extended Data Fig.4g, indeed shows the percentage of polarised cells. Therefore, we amended the text accordingly.

- Please clarify the sampling location for single-cell RNA-seq in Fig.3. Based on UMAP plots it looks like cells from 8w pw1 sample also contribute to differentiated IFE clusters. So, is the sampling also include IFESCs derived from first healing?

We thank the Reviewer, and we apologise for the lack of clarity in the description of this scRNA-seq strategy – this is now clarified in Method section “scRNA-seq”. Indeed, this experiment was performed to include Lrig1 GL cells from both wound bed and distal areas. Importantly, in the original manuscript we did not properly emphasise that our conclusion on distal priming was coming from the histological validation of some priming features (i.e. cell shape, molecular markers) that came from the scRNA-seq analysis (Fig.3h, Fig.4d,f).

Nevertheless, we performed an additional scRNA-seq in which wound bed, distal memory and faraway control regions were individually processed (Extended Data Fig.6 and Fig.3i). The new data confirm our original claim, providing a global view of the transcriptional program of distal memory. Kindly see also Reviewer 1, point 5.

- Based on UMAP plots in Fig.3A-3C in cluster 7 the samples 1w pw1 and 1w pw2 contribute only to the IFE epidermis. If so, in Fig.3I and Fig.3J authors are just comparing the transcript among cells with different locations. It is better to show the staining for INFU memory population markers defined by the authors in different time points between 0w and 8w pw1 to see if those cells are continually located in INFU or if those cells are retracted from IFE after healing.

This is an important point: the INFU memory population at 8w pw 1 is not retracted from IFE, since at 1w pw1 the distal memory cells are actually resident in the INFU and not in the IFE. The misunderstanding originated from our undetailed description of the sampling method of scRNA-seq in Fig.3a-c. We believe that the clarification that we added to Method section “scRNA-seq” (see also Reviewer 1, point 5), partially resolves this concern. Indeed, the presence of the IFE cells made the infundibulum less represented in the scRNA-seq. However, this population was still present in the data (Fig.3a-c). To completely address the Reviewer’s point, we showed by histology that the position of Lrig1 GL cells in distal memory area at 1w pw1 and at 8w pw1 are identical (Fig.1g). Importantly, the Lrig1 GL cells in the distal memory area reside

in the infundibulum and not in the IFE from 1w pw1. The IFE exit, instead, only occurs when cells are stimulated with another injury, at 1w pw2 (Fig.1g), as now we reported also in the main text.

- Which genes were used to define the average z-score expression of cell plasticity genes presented in Fig.4B

A table of the 'cell plasticity genes', that was used to calculate the average z-score in Fig.4b is now provided as Extended Data Table 3.

- It is unclear how the authors determined the inhibitors for epigenetic drugs screening

We searched the literature for suitable drugs selectively targeting epigenetic enzymes. We selected the following: A-196, SUV420H1 and SUV420H2 inhibitor because of the role of SUV420 in EMT (Yokoyama Y. et al., 2014); UNC0638, EHMT1/2 inhibitor for the role of EHMT1/2 in cell migration and EMT process (Liu X. et al., 2018); SB747651A, MSK1 inhibitor because the drug has been demonstrated to re-modulate cell migration (Knudsen A.M. et al., 2021); EX-527, Sirtuins inhibitor for their role in regulating epidermal wound closure (Huang X. et al., 2019); PRT4165, Ring1a/1b inhibitor because of the role of PRC1 complex in maintain epidermal tissue integrity (Cohen I. et al., 2019).

The relative references have now been added to the Method section "Experiments with epigenetic drugs"
– Thank you for raising this.

Minor comments:

- In general, figures are very condensed which sometimes is difficult to read. Some data/schematics are redundant and can be moved to supplementary material. Making these figures more accessible will greatly aid the reader and more effectively convey the findings

We thank the Reviewer for the suggestion. Several panels have been moved to the Extended Data section.

- What does mean the red dashes in Fig.1E?

The red dashed lines indicate the engaged HFs, in which upon wound stimulus Lrig1 GL cells exit from the HF toward IFE. We apologise for the lack of information that we now added to the relative Figure Legend.

- What is the distance of the image presented in Fig.3K from the wound site?

The image relative to 8w pw1_Distal refers to an HF at ~5 mm from the wound bed (now Fig.3h). This is now clarified in the relative Figure Legend.

- Line 157 and 163 Fig.2J should be Fig2K.
- Line 262 Extended data Fig.2B should be Extended data Fig.2H?
- Fig.6 there is no L.

We thank the Reviewer for detecting the above-mentioned errors. The text has been revised accordingly. We have carefully checked the manuscript to rectify these and other inaccuracies.

- For Figure 2A and Extended figure 2A, please clarify based on the comparison of which groups the DEG is calculated?

In Fig.2a and in Extended Data Fig.2a (now Extended Data Fig.4a) we identified all the genes that were significantly modulated across the time course. To do so, we implemented One-way ANOVA test by fitting

a generalised linear model to all the sample groups (1w pw1, 8w pw1 and 1w pw2) and retrieved the DEGs in any time point (Reference 68), using 0w as the reference group in the model.

This is now explained in the Method section “Mini-bulk RNA-seq and analysis” and amended in the relative Figure Legend.

- Line 311 and 312 mistakes in references

We thank the Reviewer for detecting the above-mentioned error. The text has been revised accordingly.

Reviewer #3 (Remarks to the Author):

A. Summary of the key results

The authors characterize a mouse wound healing model that is based on 2 subsequent injuries at the same site. They identify a subpopulation of distal cells that becomes primed during the first injury. It is then hypothesized that this state differs from previously described states of wound memory. It is further shown that a chromatin modulatory drug (PRT4165) can modulate wound memory and preliminary evidence is supported to suggest that wound memory promotes tumor (squamous cell carcinoma) formation.

B. Originality and significance: if not novel, please include reference

Wound memory is not a novel concept and a recent publication (ref. 15) has shown that differential chromatic accessibility underpins the phenomenon. The authors make a valid point in stating that the precise lineage identity and spatial distribution remain to be fully characterized. They also show that their memory cells are distantly located to the site of injury, which appears novel. Finally, the link between wound priming and H2AK119ub as well as the connection between wound priming and skin cancer formation are novel and potentially exciting, albeit poorly substantiated.

We thank the Reviewer for underlining the novelty of our analysis that lies in the specific lineage identity of memory, the distal location of memory, the link between wound priming and H2AK119ub and the connection between wound priming and skin cancer onset.

C. Data & methodology: validity of approach, quality of data, quality of presentation

Overall, the approach is characterized by a combination of lineage tracing, transcriptomics and histology. The results are comprehensive and carefully presented. However, additional experiments using different methods are required to substantiate key claims (see below).

We thank the Reviewer for highlighting that the results are comprehensive and carefully presented.

We believe that, thanks to the suggestions from all the Reviewers, we have strengthened our main conclusions.

D. Appropriate use of statistics and treatment of uncertainties

I'm not an expert, but I did not notice any major deficiencies in statistics.

E. Conclusions: robustness, validity, reliability

The authors do not provide any mechanistic insights into wound priming, which I consider a major deficiency, as (1) ref. 15 has already provided ATAC-seq data to characterize the epigenetic states associated with wound memory and (2) it would have been straightforward to use the Chromium single-cell multiome approach to provide integrated scRNA-seq and scATAC-seq profiles.

We divided our response to this important point into part 1 and 2 as indicated by the Reviewer:

(1): "Priming" and "trained immunity-like" transcriptional programs are two different mechanisms as recently summarised by eminent scientists of the field in Divangahi M. et al., Nature Immunology 2020. In Gonzales K.A. et al., 2021 (Reference 15), the authors focus on hair follicle bulge stem cells and identify their adaptation with a "trained immunity-like" transcriptional program. In our work we compared lower hair follicle bulge stem cells (Lgr5⁺) with upper hair follicle stem cells (Lrig1⁺). We found that only the Lrig1⁺ stem cells progeny adapts to wounding through "priming", displaying more efficient enhanced cell behaviours in comparison with stem cells from the lower hair follicle (Fig.1c). Our original data on H2AK119ub reduction

and PRC1 inhibition and the new data on chromatin accessibility (see next point) describe and functionally characterise the *priming*, an adaptation behaviour that was not observed in Gonzales K.A. et al., 2021 and, to our knowledge, was not reported before in epithelial cells.

(2): As suggested by the Reviewer, to strengthen our conclusions regarding distal priming we have investigated the chromatin landscape of distal memory cells. Since we have already identified and characterised at single cell level the memory cells residing in the infundibulum, we performed ATAC-seq to profile chromatin accessibility of this sorted population.

ATAC-seq of sorted upper-HF Lrig1 GL cells (Tomato and Sca-1 double positive cells), that drastically enriches for memory cells, was performed in the distal memory region at 1w pw1, 8w pw1 and 1w pw2 and compared to 0w (Fig.5). The ATAC-seq data complemented the results of transcriptional priming in Lrig1 GL memory cells, highlighting that distal priming is driven by chromatin accessibility changes in specific genomic that become more accessible following the first injury and are maintained since then, at 8w pw1, the newly established homeostasis, and at 1w pw2. Indeed, the genes that are associated to these 1665 genomic loci (Fig.5b) are enriched in GO terms consistent with scRNA-seq data (Fig.4b and Fig.5e).

E.1) I also have reservations about use of PRT4165 for the functional experiments (Fig. 6). The rationale for the use of the drug is very weak (a "screen" consisting of 5 (!) epigenetic drugs, EFig. 6a). It is also not clear whether the observed effects are due to changes in epigenetic memory or rather due to side effects, e.g. on DSB repair. ATAC-seq and/or H2AK119ub ChIP-seq analyses are required to substantiate the initial findings.

We agree with the Reviewer that the word "screening" was exaggerated since only 5 drugs were assessed. The word 'screen' has now been modified with "test" in the manuscript. Although we tested only 5 drugs, we found that the PRT4165 PRC1 inhibitor has a relevant role in modulating memory.

Our previous data highlighted that PRT4165 treatment induced cell phenotypes and transcriptional programs similar to the wound memory.

As the Reviewer suggested, ATAC-seq of PRT4165-treated vs vehicle-treated (control) was performed on sorted upper-HF Lrig1 GL cells to profile the effects of the PRC1 inhibitor at a chromatin level.

In agreement with our previous data, the genes near to the chromatin peaks gained in the PRT4165-condition versus vehicle are significantly consistent with the genes associated to chromatin peaks gained at 1wpw1, 8w pw1, and 1w pw2 respect 0w, characteristic of physiological priming (Fig.7k) and that they belong to the same GO categories (Fig.7i). Therefore, we concluded that the drug is able to mimic distal memory not only at transcriptional, but also at chromatin level.

To address the concern about the possible side effect of PRT4165 on DSB repair, we performed an immunofluorescent staining with the marker γ -H2A.X (Fernandez-Capetillo O. et al., 2002). Our data indicate that an acute PRT41645 treatment (3 topical applications) alone is not sufficient alone to induce DNA damage (Extended Data Fig.11f).

E.2) Finally, I have reservations about the claim that wound priming is relevant for the formation of mouse and human squamous cell carcinoma (Fig. 7). The hypothesis is certainly interesting and the underlying mechanisms might be similar to the "epigenetic field effect" that has been proposed to support tumor formation. However, my enthusiasm is dampened by the superficiality of the analysis in Fig. 7 and by the lack of mechanistic/epigenetic insight, as outlined above.

We thank the Reviewer for highlighting the concept of "epigenetic field effect" referred to our results, a link that we previously failed to underline. Following this suggestion, we performed several new experiments.

Through histology we have already found that a reduction of the repressive marker H2AK119ub occurs after a wound and it is maintained in the new homeostasis at 8w pw1, both in the wound region and in the distal memory area (Fig.7a and Extended Data Fig.10f,g). To support the hypothesis of an epigenetic field effect, we now show that H2AK119ub reduction is maintained during the UV carcinogenesis and that it anti-correlates with an enhanced DNA damage accumulation (showed through γ -H2A.X histology), a key functional event in tumor initiation, in wound bed and in wound distal areas when compared to a non-injured

skin (Fig.8b). Importantly, the spatial distribution of γ -H2A.X overlaps with an increased susceptibility to cancer onset (Fig.8a).

These results, together with the tumorigenesis experiment in the presence of PRT4165 (Fig.8m) highlight the functional correlation between H2AK199ub reduction/ distal priming and tumour onset in skin.

To gain more epigenetic insights, and to further validate the link between epigenetic field effect and distal memory, we performed ATAC-seq of td Tomato⁺ Sca-1⁺ cells from distal memory and control skin areas in the context of UV carcinogenesis (Fig.8c-h).

Consistently with Lrig1 GL primed cells at 8w pw1 (Fig.5a), long-term UV treated Lrig1 GL primed cells from distal memory areas display a global increase in chromatin accessibility with respect to not memory areas (Fig.8c), highlighting the consequences of the loss of a repressive marker such as H2AK119ub. Importantly, when compared to the chromatin state relative to distal priming, the genes associated with priming are significantly consistent with the ones observed in Lrig1 GL cells from UV-treated distal memory areas and moreover, they belong to the same GO categories (Fig.8f-h).

In conclusion, our data show the existence of an epigenetic field effect in the wound distal area, but absent far away from the wound bed, as a consequence of wound priming, that favours a faster tumour onset respect to control areas located far from the wound bed.

F. Suggested improvements: experiments, data for possible revision

See point 6 above. We need a clearer picture about the epigenetic landscape that defines the wound priming state. Additionally, the mode of action of PRT4165 needs to be better analyzed and controlled. Finally, the mechanism that links wound priming to tumor formation needs to be defined. In the absence of these data, the conceptual advance over ref. 15 appears highly limited.

Thank you for these helpful suggestions: to consolidate the conceptual advances of our study in the new revised manuscript we provided the data requested by this reviewer. In particular, we:

1) provided a clear picture about the chromatin landscape that defines the wound priming state (Fig.5) through new experiments as detailed in our response to this Reviewer's comments;

2) showed that the treatment with PRT4165 overall recapitulates the transcriptional program that characterises priming (Fig.7e), enhancing several cell phenotypes such as migration (Fig.7c). We additionally showed that PRT4165 treatment induces chromatin accessibility changes similarly to physiological priming (Fig.7i-k), validating the functional role of H2AK119ub reduction in distal memory onset;

3) proved that the chromatin rearrangement of distal priming, that occurs after an injury experience in distal areas and involves the loss of H2AK119ub repressive mark, is maintained in our UV induced skin carcinogenesis and lays the foundations for the establishment of an epigenetic field effect that boosts SCCs initiation (Fig.8).

Please also refer to our response to point E.2 of this Reviewer for additional details.

G. References: appropriate credit to previous work?

Yes (but I'm not a specialist in the field of wound healing).

H. Clarity and context: lucidity of abstract/summary, appropriateness of abstract, introduction and conclusions.

Fine overall, but the level of clarity and accessibility could be improved.

We carefully revised the manuscript in this sense.

Decision Letter, first revision:

Our ref: NCB-A48516A-Z

22nd December 2022

Dear Giacomo,

Thank you for submitting your revised manuscript "Lifelong tissue memory relies on spatially organized dedicated progenitors located distally from the injury" (NCB-A48516A-Z). It has now been seen by the original referees and their comments are below. The reviewers find that the paper has improved in revision, and therefore we'll be happy in principle to publish it in Nature Cell Biology, pending minor revisions to satisfy the referees' final requests and to comply with our editorial and formatting guidelines.

If the current version of your manuscript is in a PDF format, please email us a copy of the file in an editable format (Microsoft Word or LaTeX)-- we cannot proceed with PDFs at this stage.

We are now performing detailed checks on your paper and will send you a checklist detailing our editorial and formatting requirements. This normally requires about a week, but given the upcoming holidays, I hope you understand there will be a delay. Thank you for your patience.

Please do not upload the final materials and make any revisions until you receive this additional information from us. I would like to note that, when the time comes for you to resubmit the revised manuscript (after we send you our checklist), we expect you to also provide a full point-by-point response to the final referee comments (like the one you have shared with me for referee #1) and have your manuscript revised, not only per our editorial guidelines, but also per the final requests by referee #1.

Thank you again for your interest in Nature Cell Biology. Please do not hesitate to contact me if you have any questions.

Happy holidays!

Stelios

Stylianos Lefkopoulos, PhD
He/him/his
Associate Editor
Nature Cell Biology
Springer Nature
Heidelberger Platz 3, 14197 Berlin, Germany

E-mail: stylianos.lefkopoulos@springernature.com

Twitter: @s_lefkopoulos

Reviewer #1 (Remarks to the Author):

The authors have done extensive revisions satisfying the various requests. There are few remaining clarifications listed below that can help increase the clarity of the manuscript.

Page 6 line 110. The cell contribution to full-thickness wound being restricted to 1 mm away from the wound site is related to wounds that are 1 mm and performed in the ear (Ref 22). This spatial organization of cellular contributions could be different in wounds that are 2 mm or 5 mm and

performed in different skin sites used in this paper, such as the tail or the back. Thus, putting this sentence into the right context (size and geographic locations) will avoid confusion.

Please explain in the text what the differences are the two graphs in Fig 2d and Extended Fig 4.

Fig 4 b please add an explanation of which two conditions are compared to obtain the GO Terms.

Measurement of wound closure should be provided in Extended Date Fig. 10a-c.

Page 13 lines 342-343 - 8w pw1 and 0w should be 40w pw1 and 40w.

How was this claim made in Page 14 lines 362-364 supported?

The new data supporting that epithelial priming promotes tumor onset are really intriguing. It would be interesting to expand upon this and discuss why in the author's opinion the tumors do not form in the wound bed but only in the distal part considering that it has the same downregulation of H2AK119ub.

Page 8 lines 192-194. While scRNA seq is not my expertise, the number of cells seems quite low. It is possible that this is why the increase in number of Lrig1 in the INFU at 8w pwi1 is not as clear as the staining show.

Reviewer #2 (Remarks to the Author):

The authors have done a fantastic job of responding to all our comments. I believe this paper is exceptionally well suited for publication given the breadth of new experiments included that clarify and convincingly demonstrate memory adjacent to initial wound site. Publish without delay!

Reviewer #3 (Remarks to the Author):

The authors have responded to all my points and the manuscript has been improved considerably.

Decision Letter, final checks:

Our ref: NCB-A48516A-Z

12th January 2023

Dear Dr. Donati,

Thank you for your patience as we've prepared the guidelines for final submission of your Nature Cell Biology manuscript, "Lifelong tissue memory relies on spatially organized dedicated progenitors

located distally from the injury" (NCB-A48516A-Z). Please carefully follow the step-by-step instructions provided in the attached file, and add a response in each row of the table to indicate the changes that you have made. Please also check and comment on any additional marked-up edits we have proposed within the text. Ensuring that each point is addressed will help to ensure that your revised manuscript can be swiftly handed over to our production team.

In recognition of the time and expertise our reviewers provide to Nature Cell Biology's editorial process, we would like to formally acknowledge their contribution to the external peer review of your manuscript entitled "Lifelong tissue memory relies on spatially organized dedicated progenitors located distally from the injury". For those reviewers who give their assent, we will be publishing their names alongside the published article.

Nature Cell Biology offers a Transparent Peer Review option for new original research manuscripts submitted after December 1st, 2019. As part of this initiative, we encourage our authors to support increased transparency into the peer review process by agreeing to have the reviewer comments, author rebuttal letters, and editorial decision letters published as a Supplementary item. When you submit your final files please clearly state in your cover letter whether or not you would like to participate in this initiative. Please note that failure to state your preference will result in delays in accepting your manuscript for publication.

Cover suggestions

As you prepare your final files we encourage you to consider whether you have any images or illustrations that may be appropriate for use on the cover of Nature Cell Biology.

Nature Cell Biology has now transitioned to a unified Rights Collection system which will allow our Author Services team to quickly and easily collect the rights and permissions required to publish your work. Approximately 10 days after your paper is formally accepted, you will receive an email in providing you with a link to complete the grant of rights. If your paper is eligible for Open Access, our Author Services team will also be in touch regarding any additional information that may be required to arrange payment for your article.

Please note that *Nature Cell Biology* is a Transformative Journal (TJ). Authors may publish their research with us through the traditional subscription access route or make their paper immediately open access through payment of an article-processing charge (APC). Authors will not be required to make a final decision about access to their article until it has been accepted. Find out more about Transformative Journals

Please use the following link for uploading these materials:
[Redacted]

Best regards,

Kendra Donahue
Staff
Nature Cell Biology

On behalf of

Stylianos Lefkopoulos, PhD

He/him/his
Associate Editor
Nature Cell Biology
Springer Nature
Heidelberger Platz 3, 14197 Berlin, Germany

E-mail: stylianos.lefkopoulos@springernature.com
Twitter: @s_lefkopoulos

Reviewer #1:

Remarks to the Author:

The authors have done extensive revisions satisfying the various requests. There are few remaining clarifications listed below that can help increase the clarity of the manuscript.

Page 6 line 110. The cell contribution to full-thickness wound being restricted to 1 mm away from the wound site is related to wounds that are 1 mm and performed in the ear (Ref 22). This spatial organization of cellular contributions could be different in wounds that are 2 mm or 5 mm and performed in different skin sites used in this paper, such as the tail or the back. Thus, putting this sentence into the right context (size and geographic locations) will avoid confusion.

Please explain in the text what the differences are the two graphs in Fig 2d and Extended Fig 4.

Fig 4 b please add an explanation of which two conditions are compared to obtain the GOTerms.

Measurement of wound closure should be provided in Extended Date Fig. 10a-c.

Page 13 lines 342-343 - 8w pw1 and 0w should be 40w pw1 and 40w.

How was this claim made in Page 14 lines 362-364 supported?

The new data supporting that epithelial priming promotes tumor onset are really intriguing. It would be interesting to expand upon this and discuss why in the author's opinion the tumors do not form in the wound bed but only in the distal part considering that it has the same downregulation of H2AK119ub.

Page 8 lines 192-194. While scRNA seq is not my expertise, the number of cells seems quite low. It is possible that this is why the increase in number of Lrig1 in the INFU at 8w pwi1 is not as clear as the staining show.

Reviewer #2:

Remarks to the Author:

The authors have done a fantastic job of responding to all our comments. I believe this paper is exceptionally well suited for publication given the breadth of new experiments included that clarify and convincingly demonstrate memory adjacent to initial wound site. Publish without delay!

Reviewer #3:

Remarks to the Author:

The authors have responded to all my points and the manuscript has been improved considerably.

Author Rebuttal, First Revision:

Point-by-point Response to Reviewer #1's comments

We thank the Reviewers that appreciated the extensive revision that we did on the manuscript.

The amended text is highlighted in green in the manuscript file.

Reviewer #1

The authors have done extensive revisions satisfying the various requests. There are few remaining clarifications listed below that can help increase the clarity of the manuscript.

Page 6 line 110. The cell contribution to full-thickness wound being restricted to 1 mm away from the wound site is related to wounds that are 1 mm and performed in the ear (Ref 22). This spatial organization of cellular contributions could be different in wounds that are 2 mm or 5 mm and performed in different skin sites used in this paper, such as the tail or the back. Thus, putting this sentence into the right context (size and geographic locations) will avoid confusion.

The Reviewer is right. To give the right context to the highlighted literature data, we have now added in the manuscript the size and the location of the lesion used in the cited work [Ref 22] (page 5).

Please explain in the text what the differences are the two graphs in Fig 2d and Extended Fig 4.

The graphs reported in Fig.2d and Extended Data Fig.4e (now Extended Data Fig.3e) represent respectively the length of exit and the area of exit of Lrig1 GL cells. As the Reviewer requested, the information has been implemented to the figure legend Extended Data Fig.3e.

Fig 4 b please add an explanation of which two conditions are compared to obtain the GOTerms.

We are sorry for the lack of clarity in this point. The GO terms represented in the histogram are relative to the gene cluster highlighted in the black box and named "cell plasticity genes" in the figure. These genes have been selected on the base of their higher expression level in the second healing time points (8w

pw1 and 1w pw2) when compared to the first healing phases (0w and 1w pw1), as shown in the heatmap in Fig.4b. We have supplemented the relative figure legend with a more specific description (page 19).

Measurement of wound closure should be provided in Extended Date Fig. 10a-c.

The measurements of wound closure are already provided in Extended Data Fig.10a-c (now Extended Data Fig.8a-c) as logFC between the distance from wound center of drug treated replicates versus vehicle treated controls. This has been highlighted in the related figure legend.

Page 13 lines 342-343 - 8w pw1 and 0w should be 40w pw1 and 40w.

We thank the reviewer for detecting this that has now been rectified (Page 10, now).

How was this claim made in Page 14 lines 362-364 supported?

In the lines pointed by the reviewer we highlighted as a crucial conclusion that the downregulation of H2AK119ub repressive mark decreases in INFU during both wound healing time points (1w pw1 and 1w pw2). This is clearly evident from the histogram in Extended Data Fig.10f (now Extended Data Fig.8f). However, our second conclusion “with a major effect at 1w pw2” was indeed an error, as reviewer spotted. Therefore, we removed it from the manuscript.

The new data supporting that epithelial priming promotes tumor onset are really intriguing. It would be interesting to expand upon this and discuss why in the author's opinion the tumors do not form in the wound bed but only in the distal part considering that it has the same downregulation of H2AK119ub.

As highlighted by the quantifications provided in Fig.8a, but not properly commented in the text, the maximum incidence of eSCCs is relative to the wound bed itself (third histogram bar in Fig.8a). A sentence that clearly underlines that the spatial gradient of tumorigenesis starts from wound bed (included) and expands to distal memory regions, has been provided in the new version of the manuscript (page 12).

Page 8 lines 192-194. While scRNA seq is not my expertise, the number of cells seems quite low. It is possible that this is why the increase in number of Lrig1 in the INFU at 8w pw1 is not as clear as the staining show.

We fully agree with the Reviewer's interpretation. Indeed, even if the scRNA-seq resolution was limited by the low cell number, the histological data highly improved the resolution and gave a clear visualization of the distal memory cells.

Final Decision Letter:

Dear Giacomo,

I am pleased to inform you that your manuscript, "Tissue memory relies on stem cell priming in distal undamaged areas", has now been accepted for publication in Nature Cell Biology.

Please note that *Nature Cell Biology* is a Transformative Journal (TJ). Authors may publish their research with us through the traditional subscription access route or make their paper immediately

open access through payment of an article-processing charge (APC). Authors will not be required to make a final decision about access to their article until it has been accepted. Find out more about Transformative Journals

If you have not already done so, we strongly recommend that you upload the step-by-step protocols used in this manuscript to the Protocol Exchange (www.nature.com/protocolexchange), an open online resource established by Nature Protocols that allows researchers to share their detailed experimental know-how. All uploaded protocols are made freely available, assigned DOIs for ease of citation and are fully searchable through nature.com. Protocols and Nature Portfolio journal papers in which they are used can be linked to one another, and this link is clearly and prominently visible in the online versions of both papers. Authors who performed the specific experiments can act as primary authors for the Protocol as they will be best placed to share the methodology details, but the Corresponding Author of the present research paper should be included as one of the authors. By uploading your Protocols to Protocol Exchange, you are enabling researchers to more readily reproduce or adapt the methodology you use, as well as increasing the visibility of your protocols and papers. You can also establish a dedicated page to collect your lab Protocols. Further information can be found at www.nature.com/protocolexchange/about

With kind regards,
Stelios

Stylianos Lefkopoulos, PhD
He/him/his

Associate Editor
Nature Cell Biology
Springer Nature
Heidelberger Platz 3, 14197 Berlin, Germany